# Neural Spacetimes
# for DAG Representation Learning

**Haitz Sáez de Ocáriz Borde**
University of Oxford

**Anastasis Kratsios**
McMaster University & Vector Institute

**Marc T. Law**
NVIDIA

**Xiaowen Dong**
University of Oxford

**Michael Bronstein**
University of Oxford & AITHYRA

## Abstract

We propose a class of trainable deep learning-based geometries called Neural SpaceTimes (NSTs), which can universally represent nodes in weighted Directed Acyclic Graphs (DAGs) as events in a spacetime manifold. While most works in the literature focus on undirected graph representation learning or causality embedding separately, our differentiable geometry can encode both graph edge weights in its spatial dimensions and causality in the form of edge directionality in its temporal dimensions. We use a product manifold that combines a quasi-metric (for space) and a partial order (for time). NSTs are implemented as three neural networks trained in an end-to-end manner: an embedding network, which learns to optimize the location of nodes as events in the spacetime manifold, and two other networks that optimize the space and time geometries in parallel, which we call a neural (quasi-)metric and a neural partial order, respectively. The latter two networks leverage recent ideas at the intersection of fractal geometry and deep learning to shape the geometry of the representation space in a data-driven fashion, unlike other works in the literature that use fixed spacetime manifolds such as Minkowski space or De Sitter space to embed DAGs. Our main theoretical guarantee is a universal embedding theorem, showing that any $k$-point DAG can be embedded into an NST with $1 + \mathcal{O}(\log(k))$ distortion while exactly preserving its causal structure. The total number of parameters defining the NST is sub-cubic in $k$ and linear in the width of the DAG. If the DAG has a planar Hasse diagram, this is improved to $\mathcal{O}(\log(k) + 2)$ spatial and 2 temporal dimensions. We validate our framework computationally with synthetic weighted DAGs and real-world network embeddings; in both cases, the NSTs achieve lower embedding distortions than their counterparts using fixed spacetime geometries.

## 1 Introduction

Graphs are a ubiquitous mathematical abstraction used in practically every branch of science from the analysis of social networks (Robins et al., 2007) to economic stability (Hurd et al., 2016) and genomics (Marbach et al., 2012). The breadth of social and physical phenomena encoded by large graphs has motivated the machine learning community to seek efficient representations (or *embeddings*) of graphs in continuous spaces. Since the structure of graphs typically has properties different from Euclidean geometry (e.g., trees exhibit exponential volume growth), a significant effort has been made to identify non-Euclidean geometries which can faithfully capture the structure of general graphs. Examples include hyperbolic embeddings of tree-like graphs (Ganea et al., 2018; Sonthalia & Gilbert, 2020; Shimizu et al., 2021; Kratsios et al., 2023a;b), spherical and toroidal embeddings of graphs with cycles (Schoenberg, 1942; Guella et al., 2016; Giovanni et al., 2022b), embeddings of graphs with several of these characteristics into product Riemannian geometries (Borde et al., 2023a;b; Giovanni et al., 2022a), combinations of constant curvature Riemannian manifolds (Lu et al., 2023) and manifolds with locally controllable Ricci curvature (Giovanni et al., 2022b).

---

Email addresses: `chri6704@ox.ac.uk`, `kratsioa@mcmaster.ca`, `marcl@nvidia.com`, `xdong@robots.ox.ac.uk`, `michael.bronstein@cs.ox.ac.uk`

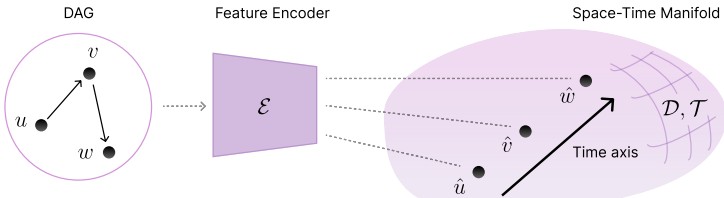

Figure 1: A *Neural Spacetime* (NST) is a learnable triplet $\mathcal{S} = (\mathcal{E}, \mathcal{D}, \mathcal{T})$, where $\mathcal{E} : \mathbb{R}^N \to \mathbb{R}^{D+T}$ is a *(feature) encoder network*, $\mathcal{D} : \mathbb{R}^{D+T} \times \mathbb{R}^{D+T} \to [0, \infty)$ is a learnable *quasi-metric* on $\mathbb{R}^D$ and $\mathcal{T} : \mathbb{R}^{D+T} \to \mathbb{R}^T$ is a learnable *partial order* on $\mathbb{R}^T$. Given an input Directed Acyclic Graph (DAG), $\mathcal{E}$ optimizes the location of the nodes $u, v, w$ as events in the spacetime manifold $\hat{u}, \hat{v}, \hat{w}$, while concurrently $\mathcal{D}$ and $\mathcal{T}$ learn the geometry of space and time themselves. The objective is to find a geometry that can faithfully represent, with minimal distortion, the metric geometry of the input DAG in space as well as its causal connectivity in time.

**The Directed Graph Embedding Problem.** The description of many real-world systems requires directed graphs, for example, gene regulatory networks (Marbach et al., 2012), flow networks (Deleu et al., 2022), stochastic processes (Backhoff-Veraguas et al., 2020), or graph metanetworks (Lim et al., 2024) to name a few. Directed graphs (or digraphs) play a significant role in causal reasoning, where they are used to study the relationship between a cause and its effect in a complex system. These problems make use of representation spaces with causal structures, modeling the directional edges of a discrete graph as causal connections in a continuous representation space (Zheng et al., 2018). However, most approaches in causal representation learning primarily focus on capturing directional information while neglecting distance information. Recent works (Sim et al., 2021; Law & Lucas, 2023) have suggested that Lorentzian *spacetimes* from general relativity are suitable representation spaces for directed graphs, as these geometries possess an *arrow of time* that provides them with causal structure (see Appendix A.5) while also being capable of encoding distance. Instead of considering a single arrow of time, in this paper we propose to turn to richer causal structures with multiple time dimensions, allowing us to model greater directional complexity.

**Problem Formulation.** Our goal is to learn a spacetime manifold representation where nodes in a DAG can be embedded as events. This should be done while capturing edge directionality in the form of causal structure (which includes modelling lack of connectivity as anti-chains), as well as the edge weights in the original discrete structure: these are represented as temporal ordering and spatial distance in the continuous embedding space.

**The Neural Spacetime Model.** We propose a trainable geometry with enough temporal and spatial dimensions to encode a broad class of weighted DAGs. In particular, the *Neural SpaceTime* (NST) model utilizes a MultiLayer Perceptron (MLP) encoder that maps graph node features to an intermediate Euclidean latent space, which is then fragmented into space and time and processed in parallel by a *neural metric* network (inspired by the neural snowflake model from Borde & Kratsios (2023)) and a *neural partial order*. These embed nodes jointly as causally connected events in the continuous representation while respecting the one-hop distance (given by the DAG weights) and directionality of the original discrete graph structure, see Figure 1. The spatial component of our model, which considers $D$ spatial dimensions, leverages *large-scale* and asymptotic embedding approaches relevant in coarse and hyperbolic geometry (Nowak, 2005; Eskenazis et al., 2019), as well as the *small-scale* embedding-based approach of Borde & Kratsios (2023) inspired by fractal-type metric embeddings. These are often used to encode undirected graphs equipped with their (global) geodesic undirected distance (Bourgain, 1986; Matoušek, 1999; Gupta, 2000; Krauthgamer et al., 2004; Neiman, 2016; Elkin & Neiman, 2021; Andoni et al., 2018; Filtser, 2020; Kratsios et al., 2023a; Abraham et al., 2022), which implies that their local distance information can also be embedded (Abraham et al., 2007; Charikar et al., 2010). Note that the opposite is not true: optimizing for the local geometry does not guarantee a faithful representation of the global geometry (Ostrovska & Ostrovskii, 2019). On the other hand, the temporal component of our trainable geometry parameterizes an order structure $\lesssim^{\mathcal{T}}$ on multiple $T$ time dimensions. Thus, causality in our representation space simply means that a point $x \in \mathbb{R}^{D+T}$ precedes another $y \in \mathbb{R}^{D+T}$ in the sense that $x \lesssim^{\mathcal{T}} y$ (which only considers order in time $T$, see Section 3). When $T = 1$, our geometry corresponds to using a Cauchy time function of the globally hyperbolic Lorentzian spacetimes to define a partial order (see Beem et al. (1996, Page 65) or Burtscher & Garcia-Heveling (2024)).

**Our contributions** are: **(1)** We propose a tractable neural network architecture for spacetimes (with multiple time dimensions), ensuring that only valid geometries are learnable; hence, the term *neural spacetimes* (NSTs). Our approach decouples the representation into a product manifold, which models (quasi-)metrics and (partial) orders independently. **(2)** Our main theoretical result, Theorem 1, provides a global embeddability guarantee showing that any (finite) poset can be embedded into a neural spacetime with a sufficient number of temporal and spatial dimensions. The embeddings are *causal* in time and *asymptotically isometric* in space. Furthermore, the neural spacetime model only requires $\mathcal{O}(k^2)$ parameters to globally embed a weighted DAG with $k$ nodes. Theorem 2 shows that a broad class of posets can be embedded into a NST with at most two temporal dimensions. **(3)** We experimentally validate our model on synthetic metric DAG datasets, as well as real-world directed graphs that involve web hyperlink connections and gene expression networks, respectively.

## 2 PRELIMINARIES: DIRECTED GRAPHS, POSETS, AND QUASI-METRICS

We now introduce the relevant notions of causal and spatial structure required to formulate our main results and our model. This section concludes by discussing the concept of *spacetime embeddings*, which represent DAGs in a *continuous space* encoding both their causal and spatial structure.

### 2.1 CAUSAL STRUCTURE

**Directed Graphs.** Consider a weighted directed graph, $G_D = (E_D, V, W_D)$, where $E_D$ represents a set of directed edges, $V$ a set of vertices, and $W_D$ the strength of the connections, respectively. We say that two vertices $u, v \in V$ are *causally connected*, denoted by $u \preccurlyeq v$, if there exists a sequence of directed edges, a path $\left((\nu_n, \nu_{n+1})\right)_{n=1}^{N-1}$ in $V$ with $\nu_1 = u$ and $\nu_N = v$. Notice that causal connectivity is a transitive relation; that is, if $u \preccurlyeq v$ and $v \preccurlyeq w$, then $u \preccurlyeq w$, for each $u, v, w \in V$. The neighborhood $\mathcal{N}(u)$ of a node $u \in V$ is the set of nodes sharing an edge with $u$; i.e. $\mathcal{N}(u) \overset{\text{def.}}{=} \{v \in V : (u, v) \in E_D \text{ or } (v, u) \in E_D\}$. Every weighted directed graph $G_D = (E_D, V, W_D)$ is naturally a graph upon forgetting edge directions, meaning that it induces a weighted undirected graph $G = (E, V, W)$ where $E \overset{\text{def.}}{=} \{\{u, v\} \in V : (u, v) \in E_D \text{ or } (v, u) \in E_D\}$ and where $W$ is the symmetrization of $W_D$, $W(u, v) \overset{\text{def.}}{=} \max\{W_D(u, v), W_D(v, u)\}$.

**Remark 1** (Feature Vectors and Nodes). *In practice, e.g. in Graph Neural Network (GNN) (Scarselli et al., 2009) use-cases, one equips the nodes in the graph $G = (E_D, V, W_D)$ with a set of feature vectors $\{x_v\}_{v \in V}$ in $\mathbb{R}^N$. We thus henceforth identify the set of nodes $V$ in any graph with a set of feature vectors in $\mathbb{R}^N$, for some positive integer $N$, via the identification $V \in v \leftrightarrow x_v \in \mathbb{R}^N$.*

**Posets.** A broad class of logical relations can be encoded as directed graphs. We consider the class of partially ordered sets (posets): the pair or tuple $(V, \precsim)$. In particular, a key characteristic of a poset is that for any two elements $u$ and $v$ in the set the following properties are satisfied: reflexivity, $\forall u \in V, u \preccurlyeq u$; antisymmetry, $u \preccurlyeq v \wedge v \preccurlyeq u \implies u = v$; and transitivity, $\forall u, v, w \in V, u \preccurlyeq v \wedge v \preccurlyeq w \implies u \preccurlyeq w$. Every poset induces a DAG structure on the vertex set $V$.

**Example 1** (From DAGs to Posets). *Given the DAG $G_D = (E_D, V, W_D)$, then we define the relation $\preccurlyeq$ on vertices $u, v \in V$ such that $u \preccurlyeq v$ if either $u = v$ or there exists a directed path from $u$ to $v$ in the graph. This relation establishes $(V, \precsim)$ as a poset.*

The antisymmetry condition is not satisfied by all digraphs, but only by certain types of digraphs such as DAGs, since it is incompatible with directed cycles of length greater than 1 (loops other than self-loops). Conversely, posets induce directed graphs. This identification is encoded through the *Hasse diagram* of a poset, see Example 2, which is a natural way to *forget all composite relations* in the poset by identifying the key *basic skeleton* relations encoding all its structure.

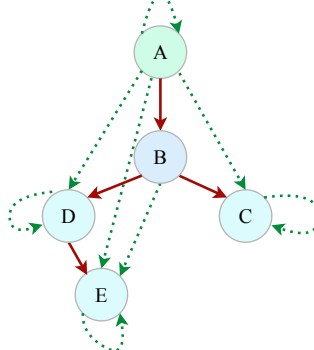

Figure 2: The red arrows illustrate the Hasse diagram (DAG) with directed edge set $\{(A, B), (B, C), (B, D), (D, E)\}$. Its corresponding poset is depicted using green arrows on the same vertex set $\{A, B, C, D, E\}$. Our *partial order* is not a *total order* as there is no red or green arrow between $C$ and $E$. Red arrows encode key DAG structure and green arrows encode all their possible compositions.

The poset can then be recovered from its Hasse diagram by adding all compositions of edges and self-loops, see Figure 2.

**Example 2** (From Posets to DAGS: Hasse Diagrams). *The Hasse diagram of a poset $(V, \lesssim)$ is a directed graph $G_D = (E_D, V)$ with $V \stackrel{\text{def.}}{=} E_D$ and $\forall u, v \in V$, $(u, v) \in E_D$ iff 1) $x_u \leq x_v$, 2) $u \neq v$ and 3) there is no $w \in V \setminus \{u, v\}$ with $x_u \leq x_w \leq x_v$ (so $x_u \leq x_v$ is a "minimal relation").*

Our theoretical results in Section 3.1 focus on embedding guarantees for posets induced by DAGs, which are of interest in causal inference (Textor et al., 2016; Oates et al., 2016), causal optimal transport on DAGs (Eckstein & Cheridito, 2023), which is particularly important in sequential decisions making (Acciaio et al., 2020; Backhoff-Veraguas et al., 2020; Xu et al., 2020; Eckstein & Pammer, 2024; Kršek & Pammer, 2024; Gunasingam & Wong, 2024) and in robust finance (Bartl et al., 2021; Acciaio et al., 2024)), in applied category theory (Fong & Spivak, 2019) since any thin category is a poset (Chandler, 2019), and in biology applications such as gene expressions (Marbach et al., 2012).

## 2.2 SPATIAL STRUCTURE

**Quasi-Metric Spaces.** Although Riemannian manifolds have been employed to formalize non-Euclidean distances between points, their additional structural characteristics, such as smoothness and infinitesimal angles, present considerable constraints. This complexity often makes it challenging to express the distance function for arbitrary Riemannian manifolds in closed-form. On the other hand, quasi-metric spaces isolate the pertinent properties of Riemannian distance functions without requiring any of their additional structures for graph embedding. A *quasi-metric space* is defined as a set $X$ equipped with a distance function $d : X \times X \to [0, \infty)$ satisfying the following conditions for every $x_u, x_v, x_w \in X$: i) $d(x_u, x_v) = 0$ if and only if $x_u = x_v$, ii) $d(x_u, x_v) = d(x_v, x_u)$, iii) $d(x_u, x_v) \leq C\big(d(x_u, x_w) + d(x_w, x_v)\big)$, for some constant $C \geq 1$. Property (iii) is called the $C$-relaxed triangle inequality. When $C = 1$, $(X, d)$ is termed a *metric space*, and the $C$-relaxed triangle inequality becomes the standard triangle inequality. Quasi-metrics arise naturally when considering local embeddings since the triangle inequality is only required to hold locally, allowing for small neighborhoods of distinct points in a point metric space to be embedded independently from one another. Furthermore, these geometries share many of the important properties of metric spaces, e.g. the Arzela-Ascoli theorem holds (Xia, 2009). Moreover, if property (i) is removed, such that several points may be indistinguishable by their distance information, then we say that $d$ is a *pseudo-quasi-metric* (Kim, 1968; Künzi, 1992). This naturally occurs when additional time dimensions are used to encode causality, but distance is ignored.

**Example 3.** *Every weighted graph $G = (E, V, W)$ induces a metric space $(V, d_G)$; e.g. using the shortest path (graph geodesic) distance $d_G$, defined for each $u, v \in V$ by*

$$d_G(u, v) \stackrel{\text{def.}}{=} \inf\left\{ \sum_{n=1}^{N-1} W(\nu_n, \nu_{n+1}) : \exists \, (\nu_1 = u, \nu_2), \dots, (\nu_{N-1}, \nu_N = v) \in E \right\}. \quad (1)$$

One could define the $\inf$ of the empty set to be $\max_{v,u \in V} W(u, v) + 1$ (instead of $\infty$, which is unsuitable for learning and embedding). However, following the spacetime representation literature, we are only interested in learning the distance between causally connected nodes (Section 3.2). We emphasize that only *simple* weighted digraphs are considered here, meaning that we do not allow for self-loops, and each ordered pair of nodes has at most one edge between them. As later discussed in Section 3.1 and Section 3.2, NSTs can model causal connectivity in one direction (but no undirected edges), as well as lack of causal connectivity between events (anti-chains).

## 2.3 SPACETIME EMBEDDINGS

We will work in a class of objects with the causal structure of DAGs/posets and the distance structure of weighted undirected graphs. We formalize this class of *causal metric spaces* as follows.

**Definition 1** (Causal (Quasi-)Metric Space). *A triple $(\mathcal{X}, d, \lesssim)$ such that $(\mathcal{X}, d)$ is a (quasi-)metric space and $(\mathcal{X}, \lesssim)$ is a poset, is called a causal (quasi-)metric space.*

Having defined a meaningful class of causal metric spaces, we now formalize what it means to approximately create a copy of a causal metric space into another. The goal is to begin with a complicated *discrete* structure, such as a DAG, and embed it into a well-behaved *continuous* structure.

**Definition 2** (Spacetime Embedding). *Let $(\mathcal{X}, d, \lesssim^x)$ and $(\mathcal{Y}, \rho, \lesssim^y)$ be causal (quasi-)metric spaces. A map $f : \mathcal{X} \to \mathcal{Y}$ is a spacetime embedding if there are constants $D \geq 1$ and $c > 0$*

*such that for each $x_1, x_2 \in \mathcal{X}$*
*(i) **Causal Embedding:** $x_1 \precsim^x x_2 \Leftrightarrow f(x_1) \precsim^y f(x_2)$*
*(ii)**Metric Embedding:** $c\,\rho(f(x_1), f(x_2)) \leq d(x_1, x_2) \leq D\,c\rho(f(x_1), f(x_2))$.*
*The constant $D$ is called the distortion, and $c$ is the scale of the spacetime embedding. A spacetime embedding $f : \mathcal{X} \to \mathcal{Y}$ for which $D = c = 1$ is called isocausal.*

**Remark 2** (Optimal Distortion). *Note that $D = 1$ is the minimal possible distortion[1]. This is because $D < 1$ is impossible and $D = 1$ yields an equality between the rescaled target metric $c\,\rho$ and the original metric $d$; i.e. $c\rho(f(x_1), f(x_2)) = d(x_1, x_2)$ for all $x_1, x_2 \in \mathcal{X}$.*

Our *spacetime embeddings* are illustrated in Figure 3. We encode causality using the notion of an order embedding from order theory (Davey & Priestley, 2002), also used in general relativity (Kronheimer & Penrose, 1967; Sorkin, 1991; Reid, 2003; Henson, 2009; Benincasa & Dowker, 2010), jointly with the notion of a metric embedding core to theoretical computation science (Gupta & Hambrusch, 1992; Gupta, 1999; Gupta et al., 2004) and representation learning (Sonthalia & Gilbert, 2020).

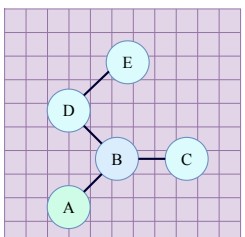 $\times$ 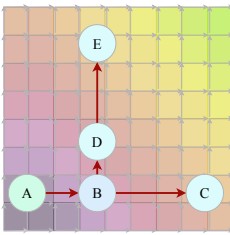

(a) *Spatial Embedding:* the objective is to replicate the distance between nodes, counted in a minimal number of hops, with the Euclidean distance on $\mathbb{R}^2$. In the neural spacetime, however, the distance will be non-Euclidean.

(b) *Temporal Embedding:* the goal of the causal embedding is to match the directions in the embedded graph (direction of red arrows) to the order in $\mathbb{R}^2$ with the product order (direction of gray arrows).

Figure 3: *Spacetime Embeddings (Definition 2)*: We illustrate a spacetime embedding of the directed graph in Figure 2 into $\mathbb{R}^4 = \mathbb{R}^2 \times \mathbb{R}^2$ with 2-space dimensions and 2 time dimensions. Notice that the spatial component of the spacetime embedding is not a causal embedding and vice versa.

## 3 NEURAL SPACETIMES

Let $N \in \mathbb{N}_+$ be the dimensionality of the feature vectors $x_u, x_v \in \mathbb{R}^N$ associated with nodes $u, v \in V$, where $V$ represents the set of nodes of the DAG $G_D = (E_D, V, W_D)$. Fix a space dimension $D \in \mathbb{N}_+$ and a time dimension $T \in \mathbb{N}_+$. A *neural spacetime* (NST) is a learnable triplet $\mathcal{S} = (\mathcal{E}, \mathcal{D}, \mathcal{T})$, where $\mathcal{E} : \mathbb{R}^N \to \mathbb{R}^{D+T}$ is an *encoder network*, e.g. an MLP, $\mathcal{D} : \mathbb{R}^{D+T} \times \mathbb{R}^{D+T} \to [0, \infty)$ is a learnable *quasi-metric* on $\mathbb{R}^D$ and $\mathcal{T} : \mathbb{R}^{D+T} \to \mathbb{R}^T$ is a learnable *partial order* on $\mathbb{R}^T$. We implement $\mathcal{D}$ and $\mathcal{T}$ as two neural networks that process the spatial and temporal dimensions of encoded feature vectors $\hat{x}_u, \hat{x}_v \overset{\text{def.}}{=} \mathcal{E}(x_u), \mathcal{E}(x_v) \in \mathbb{R}^{D+T}$ in parallel.

The encoder network, $\mathcal{E} : \mathbb{R}^N \to \mathbb{R}^{D+T}$, maps the original node feature vectors of the input graph to the dimensions of space and time used by the NST representation. We employ this mapping as an intermediate Euclidean space upon which to learn the quasi-metric and partial order. The model learns to allocate relevant information to each dimension through gradient descent, rather than attempting to manually specify which of the original node feature dimensions should correspond to space and time.

To define $\mathcal{D}$, which will be implemented using a variation of the original neural snowflake (see equation 10 from Appendix A.3). The original neural snowflake, and our upgrade thereof, employs two neural networks: the first represents the metric space as vectorial data in $\mathbb{R}^D$, and the second perturbs the Euclidean distance thereon. Together, any finite metric space may be perfectly (isometrically) embedded in this manner, whereas only one of the networks alone is not enough; e.g. expander graphs cannot be isometrically embedded into any low-dimensional Euclidean space (Kratsios et al., 2023a, Proposition 13).

---

[1]The symbol $D$ is also used in other parts of the text to denote the spatial dimensionality of the NST. We stick to $D$ here since it is standard in the metric embedding literature, but the two should not be confused.

We consider the following activation function, $\sigma_{s,l}$. The activation $\sigma_{s,l}$ depends on two trainable parameters $s, l > 0$. In the spirit of the fractalesque structure of neural snowflakes, $s$ controls whether small-scale distances should be expanded or contracted relative to the distance on $\mathbb{R}^D$. Similarly, $l$ controls the large-scale distances of points and dictates whether those should be expanded or contracted. See Appendix C.1 for an extended discussion.

**Definition 3** (Neural (Quasi-)Metric Activation). *For any $s, l > 0$, we define the neural (quasi-) metric activation to be the map $\sigma_{s,l} : \mathbb{R} \to \mathbb{R}$ given for each $x \in \mathbb{R}$ by*

$$\sigma_{s,l}(x) \overset{\text{def.}}{=} \begin{cases} \operatorname{sgn}(x) \, |x|^s & \text{if } |x| < 1 \\ \operatorname{sgn}(x) \, |x|^l & \text{if } |x| \geq 1 \end{cases} \tag{2}$$

*where the sign function* $\operatorname{sgn}$ *returns* $1$ *for* $x \geq 0$ *and* $-1$ *for* $x < 0$.

If $s = l$ then $\sigma_{s,l}(x) = \operatorname{sgn}(x) \, |x|^s$ and one recovers the key component of the snowflake activation function of Borde & Kratsios (2023) used in the majority of the proofs of its metric embedding guarantees. Note that $s$ and $l$ are not required to be coupled in any way.

A *neural (quasi-)metric* is a map $\mathcal{D} : \mathbb{R}^{D+T} \times \mathbb{R}^{D+T} \to [0, \infty)$ with iterative representation:

$$\mathcal{D}(\hat{x}_u, \hat{x}_v) \overset{\text{def.}}{=} \mathsf{W}_J \sigma_{s_J, l_J} \bullet (u_{J-1}); \; u_j \overset{\text{def.}}{=} \mathsf{W}_j \sigma_{s_j, l_j} \bullet (u_{j-1}) \text{ for } j = 1, \dots, J-1,$$
$$u_0 \overset{\text{def.}}{=} |\sigma_{s_0, l_0} \bullet (\hat{x}_u)_{1:D} - \sigma_{s_0, l_0} \bullet (\hat{x}_v)_{1:D}|, \tag{3}$$

for each $\hat{x}_u, \hat{x}_v \in \mathbb{R}^{D+T}$, where the *depth* parameter $J \in \mathbb{N}_+$, where the $s_0, l_0, \dots, s_J, l_J > 0$, and $\bullet$ denotes component-wise composition (i.e. the function is applied element-wise, which is standard in Deep Learning), weight matrices $\mathsf{W}_j \in I_D^+$ (an invertible positive matrix, see Appendix A.2) for $j < J$ and $\mathsf{W}_J \in (0, \infty)^{1 \times d}$ all of which have *positive* entries, and where the absolute value $|\cdot|$ is also applied component-wise. Note that $(\cdot)_{1:D}$ extracts the first $D$ dimensions of the input vector.

Moreover, we seek an ordering $\lesssim$ on $\mathbb{R}^{D+T}$ so that $u \preccurlyeq v$ in the poset if and only if their respective embeddings $\hat{x}_u$ and $\hat{x}_v$ are ordered in the feature space via $\lesssim$. Our class of *trainable* orders are parameterized with the following type of neural networks, which we call *neural partial orders*. Consider a map $\mathcal{T} : \mathbb{R}^{D+T} \to \mathbb{R}^T$ admitting the iterative representation

$$\mathcal{T}(\hat{x}_u) \overset{\text{def.}}{=} z_{\tilde{J}}, \qquad \forall j \in \{1, \dots, \tilde{J}\}, z_j \overset{\text{def.}}{=} \mathsf{V}_j \sigma_{\tilde{s}_j, \tilde{s}_j} \circ \mathrm{LR} \bullet (z_{j-1}) + b_j, \qquad z_0 \overset{\text{def.}}{=} (\hat{x}_u)_{D+1:D+T}, \tag{4}$$

where LR stands for LeakyReLU and $\circ$ for function composition, with the *depth* parameter $\tilde{J} \in \mathbb{N}_+$, weight matrices $\mathsf{V}_j \in I_T^+$, and bias terms $b_1, \dots, b_{\tilde{J}} \in \mathbb{R}^T$. Also, note that $\mathcal{T}(\hat{x}_u) \in \mathbb{R}^T$, whereas when applying the subscript $\mathcal{T}(\hat{x}_u)_t \in \mathbb{R}$, the operation returns the entry of the coordinate embedding at dimension $t$. At each layer of $\mathcal{T}$ we use the (single trainable parameter, $s = l$ in equation 2) activation $\tilde{s}, \dots, \tilde{s}_{\tilde{J}} > 0$. Moreover, we would like to highlight that $\mathcal{D}$ takes both $\hat{x}_u, \hat{x}_v \in \mathbb{R}^{D+T}$ as input to compute distances, whereas $\mathcal{T}$ processes inputs independently. Hence, given any such $\mathcal{T}$ we define the partial order $\lesssim^{\mathcal{T}}$ on $\mathbb{R}^{D+T}$ given, for each $\hat{x}_u, \hat{x}_v \in \mathbb{R}^{D+T}$, by

$$\hat{x}_u \lesssim^{\mathcal{T}} \hat{x}_v \iff \mathcal{T}(\hat{x}_u)_t \leq \mathcal{T}(\hat{x}_v)_t, \forall t = D+1, \dots, D+T. \tag{5}$$

The next proposition shows that $\lesssim^{\mathcal{T}}$ always defines a partial order on the *time dimensions* of $\mathbb{R}^{D+T}$, for any $\mathcal{T}$. This is formalized by noting that, for any $D \in \mathbb{N}_+$, the quotient space $\mathbb{R}^{D+T} / \sim$ under the equivalence relation $x \sim \tilde{x} \Leftrightarrow x_{1:D} = \tilde{x}_{1:D}$ is isomorphic (as a vector space) to $\mathbb{R}^T$. Thus, there is no loss in generality in assuming that $D = 0$.

**Proposition 1** (Neural Spacetimes Always Implement Partial Orders). *If $T \in \mathbb{N}_+$, $D = 0$, and $\mathcal{T} : \mathbb{R}^{D+T} \to \mathbb{R}^T$ admits a representation as equation 4, then $\lesssim^{\mathcal{T}}$ is a partial order on $\mathbb{R}^{D+T}$. See page* 21 *for proof.*

Just as the neural snowflakes of Borde & Kratsios (2023), neural (quasi-)metrics implements a quasi-metric on the *spatial dimension* $\mathbb{R}^D$ in $\mathbb{R}^{D+T}$ (ignoring the time dimensions used only to encode causality). A key difference between the two is that our formulation allows for the weighting or discovery of the importance of each spatial dimension, as well as the rescaling of each spatial dimension individually. Thus, our model in equation 3 enables greater flexibility as it departs further from the somewhat Euclidean structure of Borde & Kratsios (2023), which is based on warping a precomputed Euclidean distance.

**Remark 3** (Comparing Neural Snowflakes to Neural (Quasi-)metrics in NSTs.). *Neural (quasi-)metrics generalize the non-Euclidean metrics studied in Gozlan (2010, Proposition 1.2) and implemented as part of the neural snowflake model in Borde & Kratsios (2023).*

This can be of interest in learning theory since it was shown in Gozlan (2010) that these types of distances exhibit favourable concentration of measure properties when measured in 1-Wasserstein distance from optimal transport. The connection to learning occurs due to the technique of Amit et al. (2022); Hou et al. (2023); Benitez et al. (2023); Kratsios et al. (2024) whereby one may obtain uniform generalization bounds for Hölder learners through concentration of measure results.

**Proposition 2** (A Neural (Quasi-)Metric is a quasi-metric on the spatial dimensions). *Any $\mathcal{D}$ with representation in equation 3 is a quasi-metric on $\mathbb{R}^D$. If each $\mathsf{W}_j$ is orthogonal [2] and $0 < s_j, l_j \leq 1$ then $\mathcal{D}$ is a metric on $\mathbb{R}^D$. Furthermore, if $T \in \mathbb{N}_+$, then $\mathcal{D}$ is a pseudo-quasi-metric on $\mathbb{R}^{D+T}$. See page 21 for proof.*

## 3.1 Embedding Guarantees

We now present our main theoretical result which is the (global) embedding guarantees for NSTs.

**Definition 4** (Width). *Let $(P, \lesssim)$ be a poset. A $S \subseteq P$ is called an anti-chain if for each $u, v \in S$ neither $u \preccurlyeq v$ nor $v \preccurlyeq u$. The width of $(P, \lesssim)$ is the maximal cardinality of any anti-chain in $P$.*

Note that to model anti-chains with our framework, more than one time dimension is needed. This allows our model to represent lack of causal connectivity between nodes that do not share a directed path between them in the graph. The result below uses the notion of doubling constant and metric space dimension from fractal geometry, see Appendix A.1.

**Theorem 1** (Universal Spacetime Embeddings). *Fix $W, N \in \mathbb{N}_+$, $K > 0$, and let $(P, d, \lesssim)$ be a finite causal metric space with doubling constant $K$, width $W$, and $P \overset{\text{def.}}{=} \{x_v\}_{v \in V} \subseteq \mathbb{R}^N$. There exists a $D \in \mathcal{O}(\log(K))$, an MLP $\mathcal{E} : \mathbb{R}^N \to \mathbb{R}^{D+W}$, $\mathcal{T} : \mathbb{R}^{D+W} \to \mathbb{R}^W$ with representation equation 4, and $\mathcal{D} : \mathbb{R}^{D+W} \times \mathbb{R}^{D+W} \to [0, \infty)$ with representation equation 3 such that for each $x_u, x_v \in P$*
*(i) **Order Embedding:** $x_u \preccurlyeq x_v$ if and only if $\mathcal{E}(x_u) \lesssim^{\mathcal{T}} \mathcal{E}(x_v)$,*
*(ii) **Metric Embedding:** $d(x_u, x_v) \leq \mathcal{D}\big(\mathcal{E}(x_u), \mathcal{E}(x_v)\big) \leq \mathcal{O}\big(\log(K)^5\big) d(x_u, x_v)$.*
*Moreover, setting $D = k$, (ii) can be improved to an isocausal embedding, i.e. $d(x_u, x_v) = \mathcal{D}\big(\mathcal{E}(x_u), \mathcal{E}(x_v)\big)$. In either case, the number of non-zero parameters determining the neural spacetime triplet $\mathcal{S} = (\mathcal{E}, \mathcal{D}, \mathcal{T})$ is $\tilde{\mathcal{O}}\big(D + W + k^{5/2}D^4 N\big)$. See page 25 for proof.*

Theorem 1 guarantees that one never needs more than $W$ time dimensions[3] and $\#P$ space dimensions to have a (perfect) isocausal spacetime embedding. However, note that $W + \mathcal{O}(\log(\#P))$ spacetime dimensions are guaranteed to provide an embedding with a very small distortion, which is likely good enough for most practical problems. Theorem 1 is *extremal*, in the sense that it holds for all causal metric spaces. One may, therefore, ask: *How much can the result be improved for causal metric spaces with favourable properties?*

In analogy with the Minkowski spacetime representations in Law & Lucas (2023), which can accommodate directed line graphs, we now investigate when few (at most two) time dimensions are enough to guarantee a causal embedding for a poset. Using Baker et al. (1970), we first characterized posets which admit a spacetime embedding with two time dimensions via their *Hasse diagrams*.

**Proposition 3** (When two time dimensions are not enough). *If the Hasse diagram of a poset $(P, \lesssim)$ is not a planar graph, then there is no spacetime embedding $\mathcal{E} : P \to \mathbb{R}^{D+W}$, $\mathcal{D} : \mathbb{R}^{D+W} \times \mathbb{R}^{D+W} \to [0, \infty)$, $\mathcal{T} : \mathbb{R}^{D+W} \to \mathbb{R}^W$ with $W = 1, 2$. See page 26 for proof.*

We now improve both the temporal and spatial guarantees of the spacetime embedding in Theorem 1 for the posets characterized by the preceding proposition. To leverage the planar structure of the Hasse diagrams of these posets and thus enhance the spatial component of our spacetime embeddings, we instead equip any such poset $(P, \lesssim)$ with the metric $d_{\mathrm{H}}$, defined as the graph geodesic distance on the undirected Hasse diagram of $(P, \lesssim)$.

---

[2] Since any $\mathsf{W}_j$ must have only non-negative entries, then if $\mathsf{W}_j$ is additionally orthogonal, it must be a permutation matrix.

[3] $W$ here is used for time dimensions and not edge weigths.

**Theorem 2** (Low-Distortion Spacetime Embeddings in Time Dimensions). *Fix $k, N \in \mathbb{N}_+$, and let $(P, \lesssim, d_{\mathrm{H}})$ is $k$-point poset whose Hasse diagram for $(P, \lesssim)$ is planar, $P \subset \mathbb{R}^N$. There is a ReLU MLP $\mathcal{E} : \mathbb{R}^N \to \mathbb{R}^{\mathcal{O}(\log(k))+2}$, $\mathcal{D} : \mathbb{R}^{\mathcal{O}(\log(k))+2} \times \mathbb{R}^{\mathcal{O}(\log(k))+2} \to [0, \infty)$ with representation equation 3, and $\mathcal{T} : \mathbb{R}^{\mathcal{O}(\log(k))+2} \to \mathbb{R}^2$ as in equation 4 satisfying: for each $x_u, x_v \in P$*
*(i) **Causality**: $x_u \preccurlyeq x_v$ if and only if $\mathcal{E}(x_u) \lesssim^{\mathcal{T}} \mathcal{E}(x_v)$,*
*(ii) **Low-Distortion**: $d_{\mathrm{H}}(x_u, x_v) \leq \mathcal{D}\big(\mathcal{E}(x_u), \mathcal{E}(x_v)\big) \leq \mathcal{O}(\log(k)^2)\, d_{\mathrm{H}}(x_u, x_v)$.*
*The number of parameters in $\mathcal{S}$ is $\mathcal{O}\big(k^{5/2} D^4 N \, \log(N) \, \log\big(k^2 \, \mathrm{diam}(P, d_{\mathrm{H}})\big)\big)$. Proof on page 26.*

Our training procedure (Section 3.2) focuses on embedding the *local* structure of directed graphs and is supported by our theoretical guarantees that NSTs are expressive enough to encode the *global* structure of weighted DAGs. By focusing on local embeddings, our neural spacetime model can be trained to embed very large directed graphs, since local distance information is readily available, but global distance information is generally expensive to compute. Interestingly,

**Remark 4** (Local vs. Global NST.). *The global and local embeddability of directional information is equivalent in our case due to the transitivity properties of DAGs and our representation spaces.*

For completeness, we compare neural spacetimes to hyperbolic representation in Appendix A.6.

## 3.2 COMPUTATIONAL IMPLEMENTATION

In this section, we provide a discussion on how to bridge the theoretical embedding guarantees presented earlier into a computationally tractable model that can be optimized via gradient descent. Further details are provided in Appendix C.

**Representing graph edge directionality as a partial order in the embedding manifold.** Given a weighted directed graph $G_D = (E_D, V, W_D)$, let $x_u, x_v \in \mathbb{R}^N$ be the feature vectors associated with the nodes $u, v \in V$, which we want to embed as events in our spacetime: $\mathcal{S}$ takes as input these features and maps them to an embedding in the manifold. The edge set $E_D$ induces a binary non-symmetric adjacency matrix $\mathbf{A}$. In our case, $G_D$ is not any digraph but a DAG, hence if $A_{uv} = 1 \Rightarrow A_{vu} = 0$. Both entries can only be equal when they are 0, $A_{uv} = A_{vu} = 0$. In the latter case, the nodes may either be causally disconnected, or causally connected but not neighbors. If the input graph does not satisfy these properties our spacetime embedding parametrized by the NST will not be able to faithfully embed the input graph in terms of its edge directionality.

**Representing graph edge weights as distances in the embedding manifold.** Likewise, $W_D$ induces a distance matrix $\mathbf{D}$ with entries $D_{uv}$ being the causal distance between events $u, v$. Although theoretically this could be computed as the shortest path graph geodesic distance for two causality connected and distant $u$ and $v$ (similar to equation 1 but for weighted directed graphs, see Appendix D.1), in practice we only optimize for the one-hop neighborhood. In particular, if $u = v \Rightarrow D_{uv} = 0$, if $u \preccurlyeq v \wedge v \in \mathcal{N}(u) \wedge u \neq v \Rightarrow D_{uv} > 0$. The distance $D_{uv}$ is ignored otherwise, that is, we do not model the distance between nodes in the original graph that are causally connected but outside the one-hop neighborhood of each other nor do we model the distance between events that are not causality connected (anti-chains). This is in line with the literature on graph construction via Lorentzian pre-length spaces (Law & Lucas, 2023). In the case of NSTs, it is achieved using the connectivity $A_{uv}$ as a mask in the loss function.

**Local geometry optimization and global geometry implications.** Note that although we optimize for the one-hop neighborhood of each node only, transitivity of the causal connectivity of nodes across hops will be satisfied by definition of the partial order (Remark 4). Additionally, although the partial order between anti-chains is not directly optimized, as the number of time dimensions increases, it is increasingly probable that $\lesssim^{\mathcal{T}}$ will not be satisfied for nodes not causally connected, as desired. Conversely, there is no guarantee that when directly evaluating the distance between causally connected nodes using $\mathcal{D}$ we will obtain the graph geodesic distance between nodes. In summary, if $x_u, x_v, x_w \in \mathbb{R}^N$ are feature vectors for nodes $u, v, w \in V$, and $\hat{x}_u, \hat{x}_v, \hat{x}_w \in \mathbb{R}^{D+T}$ are their $\mathcal{E}$ encodings in the intermediate Euclidean space used by the NST, if $(\hat{x}_u \lesssim^{\mathcal{T}} \hat{x}_v) \wedge (\hat{x}_v \lesssim^{\mathcal{T}} \hat{x}_w) \Rightarrow \hat{x}_u \lesssim^{\mathcal{T}} \hat{x}_w$. On the other hand, in general $\mathcal{D}(\hat{x}_u, \hat{x}_v) + \mathcal{D}(\hat{x}_v, \hat{x}_w) = D_{uv} + D_{vw}$, and provided that this particular path corresponds to the graph geodesic $D_{uv} + D_{vw} = d_G(u, w)$, but $\mathcal{D}(\hat{x}_u, \hat{x}_v) + \mathcal{D}(\hat{x}_v, \hat{x}_w) \neq \mathcal{D}(\hat{x}_u, \hat{x}_w)$. The latter inequality is acceptable and not required to faithfully encode the graph edge weights.

**Training and Loss Function.** Let us use the matrix $\mathbf{X} \in \mathbb{R}^{M \times N}$ to denote the collection of feature vectors for $M = |\mathcal{V}|$ nodes associated with the DAG, $G_D$. The NST minimizes the following loss:

Table 1: DAG embedding results. Embedding dimension $D = T = 2, 4, 10$.

| Metric | Embedding Dim | Distortion (average ± std) | Max Distortion | Directionality | |
|---|---|---|---|---|---|
| $\|\mathbf{x} - \mathbf{y}\|^{0.5} \log(1 + \|\mathbf{x} - \mathbf{y}\|)^{0.5}$ | 2 | $1.09 \pm 0.24$ | 3.18 | 1.0 | |
| | 4 | $1.02 \pm 0.06$ | 1.51 | 1.0 | |
| | 10 | $1.00 \pm 0.03$ | 1.24 | 1.0 | |
| $\|\mathbf{x} - \mathbf{y}\|^{0.1} \log(1 + \|\mathbf{x} - \mathbf{y}\|)^{0.9}$ | 2 | $1.16 \pm 0.45$ | 6.21 | 1.0 | Neural Spacetime |
| | 4 | $1.02 \pm 0.07$ | 1.75 | 1.0 | |
| | 10 | $1.00 \pm 0.04$ | 1.47 | 1.0 | |
| $1 - \exp \frac{-(\|\mathbf{x} - \mathbf{y}\| - 1)}{\log(\|\mathbf{x} - \mathbf{y}\|)}$ | 2 | $1.51 \pm 1.18$ | 13.55 | 1.0 | |
| | 4 | $1.11 \pm 0.41$ | 8.92 | 1.0 | |
| | 10 | $1.01 \pm 0.05$ | 1.31 | 1.0 | |
| $\|\mathbf{x} - \mathbf{y}\|^{0.5} \log(1 + \|\mathbf{x} - \mathbf{y}\|)^{0.5}$ | 2 | $2.86 \pm 5.22$ | 72.66 | 0.99 | |
| | 4 | $1.70 \pm 2.77$ | 71.09 | 0.99 | |
| | 10 | $1.21 \pm 1.33$ | 35.58 | 0.99 | |
| $\|\mathbf{x} - \mathbf{y}\|^{0.1} \log(1 + \|\mathbf{x} - \mathbf{y}\|)^{0.9}$ | 2 | $6.77 \pm 133.68$ | 1669.83 | 0.99 | Minkowski |
| | 4 | $1.70 \pm 5.21$ | 77.03 | 0.99 | |
| | 10 | $1.19 \pm 1.09$ | 25.18 | 0.99 | |
| $1 - \exp \frac{-(\|\mathbf{x} - \mathbf{y}\| - 1)}{\log(\|\mathbf{x} - \mathbf{y}\|)}$ | 2 | $11.37 \pm 114.98$ | 1876.54 | 0.98 | |
| | 4 | $2.49 \pm 8.72$ | 198.04 | 0.98 | |
| | 10 | $1.18 \pm 2.49$ | 82.67 | 0.99 | |
| $\|\mathbf{x} - \mathbf{y}\|^{0.5} \log(1 + \|\mathbf{x} - \mathbf{y}\|)^{0.5}$ | 2 | $\infty \pm$ | $\infty$ | 0.99 | |
| | 4 | $-4.33 \pm 816.47$ | 10235.71 | 0.99 | |
| | 10 | $288.17 \pm 9794.97$ | 324027.5 | 0.99 | |
| $\|\mathbf{x} - \mathbf{y}\|^{0.1} \log(1 + \|\mathbf{x} - \mathbf{y}\|)^{0.9}$ | 2 | $9.40 \pm 2226.84$ | 63968.21 | 0.99 | De Sitter |
| | 4 | $174.69 \pm 3637.32$ | 115851.88 | 0.99 | |
| | 10 | $-10.66 \pm 739.71$ | 8997.62 | 0.99 | |
| $1 - \exp \frac{-(\|\mathbf{x} - \mathbf{y}\| - 1)}{\log(\|\mathbf{x} - \mathbf{y}\|)}$ | 2 | $-183.71 \pm 9600.71$ | 39648.66 | 0.94 | |
| | 4 | $83.04 \pm 4313.82$ | 97524.21 | 0.94 | |
| | 10 | $418.26 \pm 6599.80$ | 150543.73 | 0.94 | |

$$\mathcal{L}_{G_D}(\mathbf{X}, \mathbf{A}, \mathbf{D}) \stackrel{\text{def.}}{=} \sum_{u=u_1}^{u_M} \sum_{v=v_1}^{v_M} \mathcal{L}_{uv}\left(\mathcal{S}(x_u, x_v), (A_{uv}, D_{uv})\right), \qquad (6)$$

where $x_u$ extracts the row feature vector for $u$ from matrix $\mathbf{X}$. The goal is to learn appropriate embeddings so that nodes connected by directed edges are represented as causally connected events in the time dimensions of the manifold, and respect the distance $D_{uv}$ induced by the graph weights in the space dimensions. To do so, we can further divide the loss function into a (causal) distance loss, $\mathcal{L}_{uv}^D$, and a causality loss, $\mathcal{L}_{uv}^C$. We remind the reader of the definition $\hat{x}_u, \hat{x}_v \stackrel{\text{def.}}{=} \mathcal{E}(x_u), \mathcal{E}(x_v) \in \mathbb{R}^{D+T}$. Hence the loss can be partitioned into:

$$\mathcal{L}_{uv} \stackrel{\text{def.}}{=} \mathcal{L}_{uv}^D\left(\mathcal{D}(\hat{x}_u, \hat{x}_v), (A_{uv}, D_{uv})\right) + \mathcal{L}_{uv}^C\left(\mathcal{T}(\hat{x}_u), \mathcal{T}(\hat{x}_v), A_{uv}\right). \qquad (7)$$

We dissect the two loss terms. The distance loss for a pair of nodes is the mean squared error (MSE) loss between the predicted and ground truth distance, with masking given by the adjacency matrix:

$$\mathcal{L}_{uv}^D \stackrel{\text{def.}}{=} A_{uv}\text{MSE}\left(\mathcal{D}(\hat{x}_u, \hat{x}_v), D_{uv}\right). \qquad (8)$$

On the other hand, the causality loss (with respect to the one-hop neighborhood) is

$$\mathcal{L}_{uv}^C \stackrel{\text{def.}}{=} A_{uv}\mathcal{L}_C^*\left(\sum_{t=1}^T \text{SteepSigmoid}(\mathcal{T}(\hat{x}_u)_t - \mathcal{T}(\hat{x}_v)_t)\right), \qquad (9)$$

where $\mathcal{L}_C^*$ is a function that takes as input the expression above and $\text{SteepSigmoid}(x) = \frac{1}{1+e^{-10x}}$ (we omit some details here for simplicity, see Appendix C.3 for details). The loss for two causally connected events $u \preccurlyeq v$ in the first neighborhood of each other ($A_{uv} = 1$) is minimized when $\hat{x}_u \lesssim^{\mathcal{T}} \hat{x}_v$ is satisfied (equation 5). If the time coordinates of $\hat{x}_v$ associated with the event $v$ are all greater than those of $\hat{x}_u$ for $u$, then all SteepSigmoid activation functions will return $\approx 0$.

## 4 EXPERIMENTAL RESULTS

**Synthetic Weighted DAG Embedding.** We generate DAGs (see Appendix D.2) embedded in the 2D plane with associated local distances between nodes given by several metrics. We measure the embedding capabilities of NSTs compared to closed-form spacetimes such as the Minkowski and De Sitter spaces in Law & Lucas (2023). Although we do not optimize the geometries in the case of the baselines, we do use a neural network encoder to map points to events in the manifolds. We quantify both the average and maximum metric distortion (the ratio between true and predicted distances) following Kratsios et al. (2023b), as well as the accuracy of the time embedding. As can be seen

Table 2: Embedding results for real-world web page hyperlink and gene regulatory networks.

| Dataset | Embed. Dim | Neural Spacetime | | | Minkowski | | | De Sitter | | |
|---|---|---|---|---|---|---|---|---|---|---|
| | | Distortion | Max Distortion | Directionality | Distortion | Max Distortion | Directionality | Distortion | Max Distortion | Directionality |
| Cornell | 2 | $1.00 \pm 0.07$ | 1.31 | 0.93 | $1.07 \pm 0.70$ | 9.43 | 0.94 | $-55.83 \pm 890.45$ | 3950.88 | 0.92 |
| | 4 | $1.00 \pm 0.04$ | 1.08 | 0.94 | $1.00 \pm 0.00$ | 1.01 | 0.94 | $-20.60 \pm 249.49$ | 403.46 | 0.94 |
| | 10 | $1.00 \pm 0.04$ | 1.08 | 0.94 | $1.00 \pm 0.00$ | 1.00 | 0.94 | $0.80 \pm 126.26$ | 1543.07 | 0.93 |
| Texas | 2 | $1.01 \pm 0.10$ | 2.27 | 0.89 | $1.12 \pm 1.73$ | 31.27 | 0.90 | $-0.29 \pm 84.42$ | 818.10 | 0.90 |
| | 4 | $1.00 \pm 0.01$ | 1.05 | 0.90 | $1.00 \pm 0.00$ | 1.00 | 0.90 | $42.03 \pm 795.51$ | 13939.25 | 0.90 |
| | 10 | $1.00 \pm 0.00$ | 1.00 | 0.90 | $1.01 \pm 0.01$ | 1.05 | 0.90 | $2.60 \pm 70.33$ | 1107.60 | 0.90 |
| Wisconsin | 2 | $1.00 \pm 0.10$ | 1.67 | 0.89 | $5.07 \pm 65.99$ | 1410.03 | 0.90 | $2.06 \pm 63.46$ | 1291.31 | 0.89 |
| | 4 | $1.00 \pm 0.04$ | 1.16 | 0.89 | $1.00 \pm 0.04$ | 1.19 | 0.90 | $-0.78 \pm 27.91$ | 114.24 | 0.90 |
| | 10 | $1.00 \pm 0.04$ | 1.20 | 0.89 | $1.13 \pm 0.69$ | 16.28 | 0.90 | $0.04 \pm 215.94$ | 2862.19 | 0.89 |
| In silico | 2 | $1.06 \pm 0.47$ | 18.54 | 1.00 | $105.42 \pm 4671.85$ | 209248.72 | 0.94 | $-63.59 \pm 1866.69$ | 56626.97 | 0.92 |
| | 4 | $1.00 \pm 0.09$ | 1.73 | 1.00 | $0.25 \pm 54.57$ | 1315.76 | 0.95 | $-468.81 \pm 33021.14$ | 65289.22 | 0.92 |
| | 10 | $1.00 \pm 0.05$ | 1.32 | 1.00 | $1.00 \pm 0.05$ | 3.69 | 0.99 | $-129.13 \pm 9623.30$ | 261531.59 | 0.93 |
| E. coli | 2 | $1.02 \pm 0.45$ | 15.37 | 1.00 | $-4.25 \pm 149.61$ | 438.34 | 0.97 | $34.65 \pm 2637.50$ | 119047.23 | 0.91 |
| | 4 | $1.00 \pm 0.06$ | 2.62 | 1.00 | $1.00 \pm 0.01$ | 1.08 | 0.98 | $-2.00 \pm 3294.81$ | 130509.59 | 0.91 |
| | 10 | $1.00 \pm 0.05$ | 1.17 | 1.00 | $1.00 \pm 0.01$ | 1.01 | 0.99 | $-8.26 \pm 94.57$ | 652.96 | 0.91 |
| S. cerevisiae | 2 | $1.05 \pm 0.34$ | 10.18 | 1.00 | $-2.38 \pm 173.57$ | 151.43 | 0.91 | $55.36 \pm 3960.09$ | 160278.39 | 0.90 |
| | 4 | $1.00 \pm 0.07$ | 1.63 | 1.00 | $1.04 \pm 2.25$ | 63.17 | 0.98 | $-28.60 \pm 1175.67$ | 63086.54 | 0.90 |
| | 10 | $1.00 \pm 0.05$ | 1.57 | 1.00 | $1.01 \pm 0.02$ | 1.39 | 0.99 | $-121.17 \pm 7550.16$ | 84724.25 | 0.91 |

in Table 1, we are always able to embed edge directionality (0 for no edges embedded correctly, 1 for all edges embedded correctly). In terms of metric distortion, as the embedding dimension increases, both average and maximum distortion decrease. NSTs are particularly good at retraining low distortions in low-dimensional embedding spaces. See Table 9 for more results.

**Real-World Network Embedding.** We test our approach on real-world networks. In Table 2, we present results for the Cornell, Texas, and Wisconsin (WebKB) datasets (Rozemberczki et al., 2021), which are based on webpages represented as nodes and directed hyperlinks between them. All the nodes have bag-of-word features that we use as input of the neural spacetime encoder. The neural partial order encodes the hyperlink directionality between websites, and the neural (quasi-)metric learns the connectivity strength as the cosine similarity between connected webpage features. We achieve very low distortions, showcasing the embedding capabilities of our network. Note that, from a metric learning perspective, real-world datasets are generally less challenging than the metrics presented in Table 1, which we chose to be particularly unconventional on purpose. In terms of the time embedding, we manage to mostly encode directionality; however, these datasets are not pure DAGs since they contain some directed cycles, so it is not possible to embed them perfectly. We also work with real-world gene regulatory network datasets (Marbach et al., 2012) in line with spacetime representation learning literature (Law & Lucas, 2023; Sim et al., 2021). We achieve good spatial and causal embeddings for the In silico, Escherichia coli, and Saccharomyces cerevisiae datasets, see Table 2. Experimental details and hyperparameters for all setups can be found in Appendix D. Additionally, we present tree (spatial only) embedding experiments comparing neural snowflakes to neural (quasi-)metrics in NSTs, as well as hyperbolic neural networks (HNNs), in Appendix D.1.

## 5 CONCLUSION

We have introduced the concept of neural spacetimes, a computational framework utilizing neural networks to construct spacetime geometries with multiple time dimensions for DAG representation learning. We decouple our representation into a product manifold of space, equipped with a quasi-metric, and time, which captures causality via a partial order. We propose techniques to build, optimize, and stabilize artificial neural networks with fractalesque activation functions, ensuring the learnability of valid geometries. Our main theoretical contributions include a global embeddability guarantee for posets into neural spacetimes, with embeddings being causal in time and asymptotically isometric in space. We demonstrate the efficacy of our approach through experiments on synthetic metric DAG embedding datasets and real-world directed graphs, showcasing its superiority over existing spacetime representation learning methods with fixed closed-form geometries.

**Limitations.** Our guarantees are restricted to embeddings for DAGs rather than arbitrary digraphs. We also find that, from an optimization perspective, it is easier to optimize the geometry locally rather than globally. This limitation is computational; as the number of nodes in the DAG grows, it becomes increasingly challenging to compute the ground truth shortest-path geodesic distance and the global causal structure between nodes, which are used as ground truth for training. However, all our theoretical guarantees apply both globally and locally. Additionally, the local transitivity of causal connectivity will implicitly hold globally, even when performing local optimization only.

## ETHICS STATEMENT

We believe that the potential societal consequences are minimal and do not require specific highlighting at this time. We commit to ongoing awareness of the broader implications of our work and will remain vigilant in assessing any future societal impacts that may emerge as our theoretical framework is applied in practical settings.

## ACKNOWLEDGMENTS

The authors thank James Lucas and the anonymous reviewers for helpful feedback on early versions of this manuscript. M. Bronstein acknowledges this research is partially supported by the EPSRC Turing AI World-Leading Research Fellowship No. EP/X040062/1 and the EPSRC AI Hub on Mathematical Foundations of Intelligence: An "Erlangen Programme" for AI No. EP/Y028872/1. X. Dong acknowledges support from the Oxford-Man Institute of Quantitative Finance and the EPSRC (EP/T023333/1). A. Kratsios acknowledges financial support from an NSERC Discovery Grant No. RGPIN-2023-04482 and No. DGECR-2023-00230.

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

## A  ADDITIONAL BACKGROUND

In this section, we discuss relevant background on the dimension of a metric space and invertible matrices, which are used as part of our proofs in Appendix B. We also review the original neural snowflake model, pseudo-Riemannian manifolds, Lorentzian manifolds, and spacetimes in physics. Additionally, we provide an extended discussion on past literature in spacetime representation learning in the context of machine learning, and compare neural spacetimes to hyperbolic spaces.

### A.1  DIMENSION AND SIZE OF A METRIC SPACES

The doubling constant in fractal geometry quantifies the complexity of a metric space by indicating how many smaller balls are needed to cover a larger ball. It is related to the Assouad dimension and other fractal dimensions. A small doubling constant suggests a low fractal dimension, while a large constant indicates a higher fractal dimension. This constant helps analyze self-similarity in fractals by providing a scale-invariant description of the space.

Our quantitative results use the following fractal notion of dimension/size of a metric space.

**Definition 5** (Doubling Constant). *The doubling constant $K \geq 1$ of a metric space $(X, d)$ is the smallest integer $k \geq 1$ for which the following holds: for every center $x \in X$ and each radius $r > 0$, every (open) ball $B(x, r) \stackrel{\text{def.}}{=} \{u \in X : d(x, u) < r\}$ can be covered by at most $k$ (open) balls of radius $r/2$; i.e., there exist $x_1, \dots, x_k \in X$ such that $B(x, r) \subseteq \bigcup_{i=1}^{k} B(x_i, r/2)$.*

By this definition, $K$ is the smallest possible value of $k$ that satisfies the covering condition for all $x \in X$ and all $r > 0$. Note that every finite metric space, such as every weighted graph with the shortest path/geodesic distance, has a finite doubling constant. For example, if $X$ is finite and has at least two points, then $K \geq 2$.

The size of a metric space can also be quantified by its diameter and its separation. These respectively quantify the maximal and minimal distances between points therein.

**Definition 6** (Diameter). *Let $(X, d)$ be a metric space. The diameter of $(X, d)$ is defined to be $\operatorname{diam}(X, d) \stackrel{\text{def.}}{=} \sup_{x, \tilde{x} \in X} d(x, \tilde{x})$.*

**Definition 7** (Separation). *Let $(X, d)$ be a metric space. If $X$ has at-least two points, then its separation is defined to be $\operatorname{sep}(X, d) \stackrel{\text{def.}}{=} \inf_{x, \tilde{x} \in X,\, x \neq \tilde{x}} d(x, \tilde{x})$, otherwise $\operatorname{sep}(X, d) \stackrel{\text{def.}}{=} 1$.*

### A.2  INVERTIBLE POSITIVE MATRICES

We employ square matrices that are both invertible and have positive entries for constructing our neural quasi-metric. These matrices guarantee that our snowflake produces trainable quasi-metrics, allowing the implemented distance function to distinguish between points (i.e., identify when two vectors are equal).

**Definition 8** (Invertible Positive Matrices $I_D^+$). *Let $D \in \mathbb{N}_+$. A $D \times D$ matrix $\mathsf{W}$ is invertible and positive if it can be represented as*

$$\mathsf{W} = \lambda\, I_D + |\tilde{\mathsf{W}}|,$$

*where $\lambda > 0$, $\tilde{\mathsf{W}}$ is an arbitrary $D \times D$ matrix, and $|\cdot|$ denotes entrywise absolute value applied to the matrix $\tilde{\mathsf{W}}$. The set of all such $D \times D$ invertible positive matrices $W$ is denoted by $I_D^+$.*

In Appendix B, we demonstrate that all matrices in $I_D^+$ are invertible and preserve $[0, \infty)^D$ under matrix multiplication.

### A.3  NEURAL SNOWFLAKES

Neural spacetimes build on the neural snowflake model for weighted undirected graph embedding. Here, we describe the original construction in Borde & Kratsios (2023).

**Neural Snowflakes.** Neural snowflakes leverage quasi-metric spaces and implement a learnable adaptive geometry $\|\hat{x}_u - \hat{x}_v\|_f \stackrel{\text{def.}}{=} f(\|\hat{x}_u - \hat{x}_v\|)$, where $\hat{x}_u, \hat{x}_v \in \mathbb{R}^D$. In particular, it is a map

$f : [0, \infty) \to [0, \infty)$, with iterative representation

$$
\begin{aligned}
f(t) &= u_J^{1+|p|} \\
u_j &= \mathsf{B}_j \, \psi_{a_j, b_j}(\mathsf{A}_j u_{j-1}) \, \mathsf{C}_j \qquad && \text{for } j = 1, \ldots, J \\
u_0 &= u
\end{aligned}
\tag{10}
$$

where for $j = 1, \ldots, J$, $\mathsf{A}_j$ is a $\tilde{d}_j \times d_{j-1}$ matrix, $\mathsf{B}_j$ is a $d_j \times \tilde{d}_j$-matrix, and $\mathsf{C}_j$ is a $3 \times 1$ matrix all of which have non-negative weights and at least one non-zero weight. Furthermore, for $j = 1, \ldots, J$, $0 < a_j \leq 1$, $0 \leq b_j \leq 1$, $d_1, \ldots, d_J \in \mathbb{N}_+$, and $d_0 = 1 = d_J$. Also, $p \in \mathbb{R}$. Note that $p$ controls the relaxation of the triangle inequality ($C = 2^p$) and that for $p \neq 0$, the neural snowflake is a quasi-metric. The input to the neural snowflake, denoted as $u$, represents the Euclidean distance between $\hat{x}_u$ and $\hat{x}_v$. This value serves as an intermediate representation within the neural snowflake, which is subsequently warped. In principle, $u$ need not be the Euclidean distance. In the original neural snowflake implementation, $\psi_{a,b} : \mathbb{R} \to [0, \infty)$ is a tensorized trainable activation function which sends any vector $u \in \mathbb{R}^K$, for some $K \in \mathbb{N}_+$, to the $K \times 3$ matrix $\psi_{a,b}(u)$ whose $k^{th}$ row is

$$
\psi_{a,b}(u)_k \stackrel{\text{def.}}{=} \left(1 - e^{-|u_k|}, |u_k|^a, \log(1 + |u_k|)^b\right),
\tag{11}
$$

with $0 < a$ and $0 \leq b \leq 1$ being trainable. Such a construction provides universal graph embedding guarantees for weighted undirected graphs (Borde & Kratsios, 2023). In this work, we will generalize this learnable adaptive geometry formulation to directed graphs borrowing ideas from spacetimes and pseudo-Riemannian manifolds.

### A.4 PSEUDO-RIEMANNIAN MANIFOLDS AND LORENTZIAN SPACETIMES

Next, we discuss pseudo-Riemannian manifolds and spacetimes, from which we draw inspiration for our neural spacetime construction (Section 3). Our framework generalizes neural snowflakes and their weighted undirected graph embedding properties to also incorporate causal connectivity and graph directionality. In other words, they implement an isocausal trainable embedding.

**Pseudo-Riemannian manifolds** are a generalization of Riemannian manifolds where the metric tensor $g$ is not constrained to be positive definite. A $d$-dimensional pseudo-Riemannian manifold is a smooth manifold $M$ equipped with a pseudo-Riemannian metric $g$. Formally, a pseudo-Riemannian metric tensor is a smooth, symmetric, non-degenerate bilinear form (called a *scalar product*) defined on the tangent bundle $TM$ (disjoint union of all tangent spaces). The metric tensor at each point $p \in M$ is denoted by $g_p$. The metric assigns to each tangent space $T_pM$ a signature $(\pm, \pm, \ldots, \pm)$. In particular, the tangent space $T_pM$ admits an orthonormal basis $\{\mathbf{e}_1, \mathbf{e}_2, \ldots, \mathbf{e}_d\}$ that satisfies $\forall i \in \{1, \ldots, d\}$, $g_p(\mathbf{e}_i, \mathbf{e}_i) = \pm 1$. If $g_p(\mathbf{e}_i, \mathbf{e}_i) = 1$ for all $i$, then $M$ is a Riemannian manifold, its metric tensor is positive definite, and its metric signature is $(+, +, \ldots, +)$.

**Lorentzian manifolds** are a subset of pseudo-Riemannian manifolds with a specific metric signature, typically denoted as $(-, +, +, \ldots, +)$ or $(+, -, -, \ldots, -)$. In our case, we define distances between causally connected events to be positive, and hence adopt the second sign convention. With this convention, a nonzero tangent vector $\mathbf{v}$ of a Lorentzian manifold is called causal if it satisfies $g_p(\mathbf{v}, \mathbf{v}) \geq 0$, and it is called chronological if $g_p(\mathbf{v}, \mathbf{v}) > 0$.

**Spacetimes.** The definition of a *spacetime* in general relativity varies depending on the author. Spacetimes are a subset of Lorentzian manifolds often defined so that they are equipped with a *causal structure* (Law & Lucas, 2023). Some spacetimes are called globally hyperbolic (Geroch, 1970). Global hyperbolicity ensures that the spacetime possesses a well-defined causal structure, meaning there are no closed timelike curves (CTCs) and every inextendible causal curve intersects every Cauchy surface (a spacelike hypersurface that slices through the entire manifold exactly once). This property allows for the deterministic evolution of physical fields and particles forward in time. In mathematical terms, if there exists a smooth vector field $\mathcal{V}$ on $M$ such that $g_p(\mathcal{V}(p), \mathcal{V}(p)) > 0$ for all $p \in M$, then the Lorentzian manifold is called a *spacetime* and it contains a *causal structure*. 4-dimensional spacetimes with signature $(-, +, +, +)$ or $(+, -, -, -)$ are used to model the *physical* world in general relativity. Spacetimes can exhibit various geometries, from flat *Minkowski spacetime* (used in special relativity) to curved spacetimes that account for gravitational effects predicted by Einstein's field equations. In summary, a spacetime often refers to a Lorentzian manifold that additionally satisfies certain physical and causal conditions. We refer the reader to Beem et al. (1996) for details.

**Relation to Neural Spacetimes.** Neural Spacetimes are an adaptive geometry aiming to learn, in a differentiable manner, a suitable embedding in which nodes in a DAG can be represented as causally connected events. Although Neural Spacetimes (Section 3) draw inspiration from the concepts above—separating space and time into different dimensions and capturing causal connectivity between events—strictly speaking, Neural Spacetimes utilize higher time dimensions ($T > 1$) than classical spacetimes used to model the physical world. Moreover, they also incorporate a quasi-metric to model distance in the space dimensions, which relaxes the smoothness and other requirements inherent to pseudo-Riemannian metrics.

## A.5 RELATED WORK ON SPACETIME REPRESENTATION LEARNING

Related work in the machine learning literature that considers pseudo-Riemannian manifolds with multiple time dimensions includes *ultrahyperbolic representations* (Law & Stam, 2020; Law, 2021). Ultrahyperbolic geometry is a generalization of the hyperbolic and elliptic geometries to pseudo-Riemannian manifolds. However, the works in Law & Stam (2020); Law (2021) mostly focus on the optimization of such representations, they only consider undirected graphs and do not consider partial ordering.

Spacetime representation learning (Law & Lucas, 2023) considers Lorentzian spacetimes (i.e. with one time dimension) to represent directed graphs. Following the formalism in Appendix A.4 with a metric signature $(+, -, \ldots, -)$, and noting $\overrightarrow{x_u x_v}$, the logarithmic map of $x_v$ at $x_u$, they consider that there exists an edge from $u$ to $v$ iff $g_{x_u}(\overrightarrow{x_u x_v}, t) > 0$ and $0 < g_{x_u}(\overrightarrow{x_u x_v}, \overrightarrow{x_u x_v}) < \varepsilon$ where $t$ is an arbitrary tangent vector that defines the future direction and $\varepsilon > 0$ is a hyperparameter that defines the maximal length of the geodesic from $x_u$ to $x_v$ to draw a directed edge. The hyperparameter $\varepsilon$ allows them to avoid connecting $u$ to all the descendants of $v$. Their work is limited to the optimization of embeddings (i.e. not neural networks). We draw inspiration from Law & Lucas (2023) but consider multiple time dimensions in a framework that is easy to optimize for neural networks. Our local geometry optimization framework also allows us to ignore the hyperparameter $\varepsilon$, which is necessary when there is only one time dimension. In our case causality is controlled by the multi-dimensional neural partial order $\lesssim^{\mathcal{T}}$ instead.

Also, note that all the previous works discussed above require selecting an appropriate manifold to model the embedding space a priori. Our work, on the other hand, not only optimizes the location of points in space but also the manifold itself. In other words, our framework models a trainable geometry.

## A.6 HYPERBOLIC SPACES AS COMPARED TO NEURAL SPACETIME EMBEDDINGS

Hyperbolic geometry has been shown to be relevant for describing undirected graphs without cycles, which are called trees. Indeed, in Sarkar (2011), it was shown that these trees can be algorithmically embedded into hyperbolic space with low distortion. Additionally, Ganea et al. (2018) constructed a class of hyperbolic neural networks (HNNs) which can exploit trees represented in these spaces, and Kratsios et al. (2023b) demonstrated that finite trees can indeed be embedded with arbitrarily low distortion. However, in Borde et al. (2023a), it was shown that an isometric embedding is not possible, except at "infinity" in the hyperbolic boundary of such a space Dyubina & Polterovich (2001) (which lies outside the manifold itself). The issue arises because hyperbolic space is a Riemannian manifold, and thus, it is incompatible with the branches in the tree itself.

This is not the case for the fractal geometry implemented by the neural spacetime and neural snowflake of Borde & Kratsios (2023). Indeed, in (Maehara, 2013, Theorem 3.6), it was shown that any finite tree can be isometrically embedded in Euclidean space $\mathbb{R}^d$, for $d$ large enough, but with the "snowflaked" Euclidean distance between any two points $x, y \in \mathbb{R}^d$ given by $\|x - y\|^\alpha$ where $\alpha = 1/2$. Since neural spacetime and neural snowflake can implement such fractal geometries, they are better suited to the geometry of trees.

In this paper, we propose a framework to represent DAGs that do not contain directed cycles. However, the underlying undirected graph of a DAG can contain cycles. This is for example the case for the DAG containing four nodes $\{\nu_i\}_{i=1}^4$ that satisfy only the following partial orders: $\nu_1 \preccurlyeq \nu_2 \preccurlyeq \nu_4$ and $\nu_1 \preccurlyeq \nu_3 \preccurlyeq \nu_4$, without causality relation between $\nu_2$ and $\nu_3$. The underlying undirected graph is an undirected cycle $\nu_1, \nu_2, \nu_4, \nu_3, \nu_1$, which is not appropriate for hyperbolic geometry.

## B    PROOFS

### B.1    PROPERTIES OF NEURAL SPACETIMES

*Proof of Proposition 1.* Fix $T \in \mathbb{N}_+$ and set $D = 0$. Let $\mathcal{T} : \mathbb{R}^{D+T} \to \mathbb{R}^T$ be a mapping with representation equation 4. We define the relation

$$x \lesssim^{\mathcal{T}} y \quad \text{if and only if} \quad \mathcal{T}(x)_t \leq \mathcal{T}(y)_t \quad \text{for each } t = 1, \ldots, T.$$

We now show that $\lesssim^{\mathcal{T}}$ is a partial order on $\mathbb{R}^{D+T}$ by verifying the three required properties.

**Reflexivity:** For any $x \in \mathbb{R}^{D+T}$, we have $\mathcal{T}(x)_t \leq \mathcal{T}(x)_t$ for every $t = 1, \ldots, T$. Thus, $x \lesssim^{\mathcal{T}} x$.

**Antisymmetry:** Suppose $x, y \in \mathbb{R}^{D+T}$ satisfy

$$x \lesssim^{\mathcal{T}} y \quad \text{and} \quad y \lesssim^{\mathcal{T}} x.$$

Then, for every $t = 1, \ldots, T$,

$$\mathcal{T}(x)_t \leq \mathcal{T}(y)_t \quad \text{and} \quad \mathcal{T}(y)_t \leq \mathcal{T}(x)_t,$$

which implies that $\mathcal{T}(x)_t = \mathcal{T}(y)_t$ for all $t$. Since each matrix $V_j \in I_T^+$ is injective (as noted in equation 12), and both the leaky ReLU function and the affine shifts are injective, the composition $\mathcal{T}$ is injective. Therefore, $\mathcal{T}(x) = \mathcal{T}(y)$ implies that $x = y$.

**Transitivity:** Let $x, y, z \in \mathbb{R}^{D+T}$ be such that

$$x \lesssim^{\mathcal{T}} y \quad \text{and} \quad y \lesssim^{\mathcal{T}} z.$$

Then, for every $t = 1, \ldots, T$, we have

$$\mathcal{T}(x)_t \leq \mathcal{T}(y)_t \quad \text{and} \quad \mathcal{T}(y)_t \leq T(z)_t.$$

By the transitivity of the standard order on $\mathbb{R}$, it follows that $\mathcal{T}(x)_t \leq T(z)_t$ for all $t$. Hence, $x \lesssim^{\mathcal{T}} z$.

Since $\lesssim^{\mathcal{T}}$ is reflexive, antisymmetric, and transitive, it is a partial order on $\mathbb{R}^{D+T}$. $\qquad\square$

*Proof of Proposition 2.*
**Comment:** *The main challenge in this proof is showing that $\mathcal{D}$ separates points in $\mathbb{R}^D$; that is, for any $x, y \in \mathbb{R}^D$ we have $\mathcal{D}(x, y) = 0$ if and only if $x = y$. Note that in the main text $\mathcal{D}$ takes as input the spatial coordinates of the Euclidean embeddings $\hat{x}_u, \hat{x}_v \in \mathbb{R}^{D+T}$ instead, but we avoid this notation here for simplicity.*

**Symmetry of $\mathcal{D}$:**
The map $\mathcal{D}$ is symmetric since

$$\left( |\sigma_{s_0,l_0}(x) - \sigma_{s_0,l_0}(y)| \right)_{i=1}^d = \left( |\sigma_{s_0,l_0}(y) - \sigma_{s_0,l_0}(x)| \right)_{i=1}^d;$$

meaning that the map $u \mapsto u_0$ is symmetric. Consequentially, $\mathcal{D}$ is symmetric.

**Non-Negativity of $\mathcal{D}$:**
Since each of the matrices $W_1, \ldots, W_J$ have non-negative entries then they map $[0, \infty)^D$ to itself. To see this note that for any $d, D \in \mathbb{N}_+$ if $u \in [0, \infty)^D$, $W \in \mathbb{R}^{d \times D}$, and $(W)_{i,k} \geq 0$ for each $i = 1 \ldots, D$ and $k = 1, \ldots, d$ then: for $i = 1, \ldots, D$ we have that

$$(Wu)_i = \sum_{j=1}^D W_{i,j} u_j \geq 0.$$

Thus, $\mathcal{D}(x, y) \geq 0$ for each $x, y \in \mathbb{R}^D$.

**Case I : $\mathcal{D}$ Separates Points if $T = 0$:**
Suppose that $T = 0$; otherwise we will show that $\mathcal{D}$ does not separate points.

It is sufficient to show the result for $J = 1$, with the general case following directly by recursion. Consider the map $\mathcal{D} : \mathbb{R}^{D+T} \times \mathbb{R}^{D+T} \to [0, \infty)$ with iterative representation given in equation 3; i.e. for each $x, y \in \mathbb{R}^{D+T} \cong \mathbb{R}^D$

$$\mathcal{D}(x, y) \overset{\text{def.}}{=} W_J \sigma_{s_{J+1}, l_{J+1}}(u_J),$$

$$u_j \stackrel{\text{def.}}{=} \mathsf{W}_j \sigma_{s_j, l_j} \bullet (u_{j-1}) \qquad \text{for } j = 1, \ldots, J-1,$$

$$u_0 \stackrel{\text{def.}}{=} |\sigma_{s_0, l_0}(x) - \sigma_{s_0, l_0}(y)|,$$

where, we recall equation 2, which states that for each $s, l > 0$ and $x \in \mathbb{R}$ (the activation function is applied pointwise)

$$\sigma_{s,l}(x) \stackrel{\text{def.}}{=} \begin{cases} \operatorname{sgn}(x) |x|^s & \text{if } |x| < 1 \\ \operatorname{sgn}(x) |x|^l & \text{if } |x| \geq 1 \end{cases}.$$

Since $\mathsf{W}_1 = \lambda I_D + |\tilde{\mathsf{W}}|$ and some matrix $\tilde{\mathsf{W}} \in \mathbb{R}^{D \times D}$; then, the entries of $\mathsf{W}$ are non-negative and for $i, j = 1, \ldots, D$ we have that

$$(\mathsf{W}_1)_{i,i} = \lambda + \sum_{j=1}^{D} |\tilde{\mathsf{W}}_{i,j}| > \sum_{j=1}^{D} |\tilde{\mathsf{W}}_{i,j}| \geq \sum_{j=1; \, j \neq i}^{D} |\tilde{\mathsf{W}}_{i,j}| \stackrel{\text{def.}}{=} R_i, \tag{12}$$

where, we emphasize that the strictness of the first inequality is due to the positivity of $\lambda$. Thus, the Gershgorin circle theorem, see Gershgorin (1931), implies that the eigenvalues of $\mathsf{W}_1$ belong to the set $\Lambda_1 \subseteq \mathbb{C}$ defined by

$$\Lambda_1 \stackrel{\text{def.}}{=} \bigcup_{i=1}^{D} \overline{\mathrm{B}}_2 \big( (\mathsf{W}_1)_{i,i}, R_i \big)$$

where, for $u \in \mathbb{C}$ and $r \geq 0$ we define $\overline{\mathrm{B}}_2(u, r) \stackrel{\text{def.}}{=} \{z \in \mathbb{C} : \|u - z\| \leq r\}$. Since the computation in equation 12 showed that $R_i < (\mathsf{W}_1)_{i,i}$ for each $i = 1, \ldots, D$ then $0 \notin \Lambda_1$. Thus, $\mathsf{W}_1$ is invertible.

Consequentially, $\mathbb{R}^D \ni u \mapsto \mathsf{W}_1 u \in \mathbb{R}^D$ is injective.

For any specification of $s, l > 0$ the (componentwise) map $\sigma_{s,l} \bullet : \mathbb{R}^D \ni u \to (\sigma_{s,l}(u_i))_{i=1}^{D} \in \mathbb{R}^D$ is injective as it is componentwise monotone increasing. In the notation of equation 3: Since $u_0 = 0$ if and only if $x = y$ then the map

$$\mathbb{R}^D \times \mathbb{R}^D \ni (x, y) \mapsto \mathsf{W}_1 \mapsto u_1 \in \mathbb{R}^D$$

is equal to the zero vector if and only if $x = y$. Since $\mathsf{W}_2 \in [0, \infty)^d$ then $\mathcal{D}(x, y) \geq 0$; thus, $\mathcal{D}(x, y) = 0$ if and only if $x = y$. We have just shown that $\mathcal{D}$ is point-separating (i.e. $\mathcal{D}(x, y) = 0$ if and only if $x = y$) and in the process have seen that $\mathcal{D}$ is positive (i.e. $\mathcal{D}(x, y) \geq 0$). That is, $\mathcal{D}$ separates points in $\mathbb{R}^D$.

**Case II: Pseudo-Metric for Positive Time Dimensions if $T > 0$:**
If instead $T \in \mathbb{N}_+$, then let $\mathbf{0}_T$ denote the zero vector in $\mathbb{R}^T$, and $x, y \in \mathbb{R}^D$ with $x \neq y$. Then, $\mathcal{D}\big((\mathbf{0}_T, x), (\mathbf{0}_T, y)\big) = 0$. Thus, $\mathcal{D}$ does not separate points whenever $T > 0$. Note that from a computational perspective, we account for this using masking during training, see Section 3.2 and Appendix C.

**Relaxed Triangle Inequality:**
It remains to show that a relaxed triangle inequality holds. Again, we consider the case when $T = 0$, with the general case following identically up to a more cumbersome notation. Let $x, y \in \mathbb{R}^D$, using the notation of equation 3, for $j = 0, \ldots, J$ define the constant

$$\beta_j \stackrel{\text{def.}}{=} \max\{\max\{s_j, l_j\} - 1, 0\}.$$

Note that, $\beta_j = 0$ whenever both $s_j$ and $l_j \leq 1$ and it equals to $(\max\{s_j, l_j\} - 1)$ otherwise. Further, note that if $s_j = l_j > 1$, then $\beta_j = s_j - 1$.

By definition of the operator norm of each matrix $\mathsf{W}_j$ we have that: for $j = 1, \ldots, J$

$$\|u_j\| \leq \|\mathsf{W}_j\|_{op} \|\sigma_{s_j, l_j}(u_{j-1})\|. \tag{13}$$

By (Xia, 2009, Example 2.2): for each $j = 1, \ldots, J$ we have that

$$\|\sigma_{s_j, l_j}(u_{j-1})\| \leq 2^{\beta_j} \|u_{j-1}\|. \tag{14}$$

Upon combining the bounds in equation 13 and equation 14 for each $j = 1, \ldots, J$ we arrive at

$$\mathcal{D}(x, y) \leq \prod_{j=1}^{J} \left( \|\mathsf{W}_j\|_{op} 2^{\beta_j} \right) \|u_0\| = 2^{\sum_{j=1}^{J} \beta_j} \left( \prod_{j=1}^{J} \|\mathsf{W}_j\|_{op} \right) \|u_0\|. \tag{15}$$

Again using (Xia, 2009, Example 2.2) we have that

$$
\begin{aligned}
\|u_0\| &= \left( \sum_{i=1}^{d} |\sigma_{s_0, l_0}(x_i) - \sigma_{s_0, l_0}(y_i)|^2 \right)^{1/2} \\
&\leq \left( \sum_{i=1}^{d} \left( 2^{\beta_0} |x_i - y_i| \right)^2 \right)^{1/2} \\
&= 2^{\beta_0} \left( \sum_{i=1}^{d} |x_i - y_i|^2 \right)^{1/2} \\
&= 2^{\beta_0} \|x - y\|.
\end{aligned}
\tag{16}
$$

Upon combining the estimates in equation 15 with those in equation 16 we find that

$$
\begin{aligned}
\mathcal{D}(x, y) &\leq 2^{\sum_{j=1}^{J} \beta_j} \left( \prod_{j=1}^{J} \|W_j\|_{op} \right) \|u_0\| \\
&\leq 2^{\sum_{j=1}^{J} \beta_j} \left( \prod_{j=1}^{J} \|W_j\|_{op} \right) 2^{\beta_0} \|x - y\| \\
&= 2^{\sum_{j=0}^{J} \beta_j} \left( \prod_{j=1}^{J} \|W_j\|_{op} \right) \|x - y\|.
\end{aligned}
\tag{17}
\tag{18}
$$

Finally, notice that if for each $j = 1, \ldots, J$ the matrix $W_j$ is orthogonal and $0 < s_j, l_j \leq 1$ then equation 18 becomes $1 \|x - y\|$; in which case $\mathcal{D}$ is a metric. This concludes the proof. $\qquad \square$

### B.2 EMBEDDING RESULTS

We will routinely use the following ordering.

**Definition 9** (Product Order). *Let $T \in \mathbb{N}_+$. The product, or coordinate, order $\lesssim^{\times}$ on $\mathbb{R}^T$ is defined for each $x, y \in \mathbb{R}^T$ by*

$$
x \lesssim^{\times} y \Leftrightarrow x_t \leq y_t \text{ for all } t = 1, \ldots, T.
$$

*Equivalently, $x \lesssim^{\times} y$ if $1 = \prod_{t=1}^{T} I_{x_t \leq y_t}$. (We use $\lesssim^{\mathcal{T}}$ to refer to the ordering given by the neural partial order specifically).*

A key step in our main results is the ability of neural spacetimes to encode the product order in time and snowflake metrics in space. This is quantified by the following helper lemma.

**Lemma 1** (An Implementation Lemma for Neural Spacetimes). *Let $\alpha > 0$, $1 \leq p < \infty$, and $T, D \in \mathbb{N}_+$. Then, there exist maps $\mathcal{T} : \mathbb{R}^{D+T} \to \mathbb{R}^T$ and $\mathcal{D} : \mathbb{R}^{D+T} \times \mathbb{R}^{D+T} \to [0, \infty)$, with respective representations equation 3 and equation 4, such that for all $x, y \in \mathbb{R}^{D+T}$*

    *(i) **Implementation of Product Ordering:***

$$
(x)_{D+1:D+T} \lesssim^{\mathcal{T}} (y)_{D+1:D+T} \Leftrightarrow \mathcal{T}(x)_t \leq \mathcal{T}(y)_t \text{ for } t = D+1, \ldots, D+T
$$

    *(ii) **Implementation of Snowflake of the $\ell^p$ metric:***

$$
\mathcal{D}(x, y) = \|(x)_{1:D} - (y)_{1:D}\|_p^{\alpha}
$$

*Furthermore, the parametric complexity of $\mathcal{T}$ and $\mathcal{D}$ are given by:*

    *(a) **Depth:** $\mathrm{Depth}(\mathcal{T}) = 1$ and $\mathrm{Depth}(\mathcal{D}) = 2$*

    *(b) **Width:** $\mathrm{Width}(\mathcal{T}) = T$ and $\mathrm{Width}(\mathcal{D}) = D$*

    *(c) **No. Non-zero Parameters:** $\mathrm{No.\,Param}(\mathcal{T}) = T + 2$ and $\mathrm{No.\,Par}(\mathcal{D}) = 2(3 + D)$.*

*Proof of Lemma 1.* In what follows we use, for any $i \in \mathbb{N}_+$, let $I_i$ denotes the $i \times i$ identity matrix, let $\mathbf{0}_i$ denote the zero vector in $\mathbb{R}^i$, and we use $1_{\mathbb{R}^i}$ to denote the identity map on $\mathbb{R}^i$.

**Step 1 - Implementation of the Product Order**

Observe that the product ordering (Definition 9) and the ordering equation 5 coincide if $\mathcal{T}(x) = (x)_{D+1:D+T}$ for all $x \in \mathbb{R}^{D+T}$. Consider the map $\mathcal{T} : \mathbb{R}^{D+T} \to \mathbb{R}^T$ defined for any $x \in \mathbb{R}^{D+T}$ by

$$\mathcal{T}(x) \overset{\text{def.}}{=} I_T \sigma_{1,1} \bullet (x)_{D+1:D+T} + \mathbf{0}_T \tag{19}$$

is of the form of equation 4 and by construction $\mathcal{T}(x) = x_{D+1:D+T}$ for all $x \in \mathbb{R}^{D+T}$. By construction, $\mathcal{T}$ has: depth 1, width $T$, and $T + 2$ non-zero parameters (the identity matrix diagonal entries plus $s$ and $l$).

**Step 2 - Implementation of the $\alpha$-Snowflake of the $\ell^p$-Quasi-Metric**

Fix $J = 2$, set $s_0 = l_0 \overset{\text{def.}}{=} 1$, $s_1 = l_1 \overset{\text{def.}}{=} p$, $\mathsf{W}_1 = I_D$, and $s_2 = l_2 \overset{\text{def.}}{=} \alpha/p$, $\mathbf{1}_D \overset{\text{def.}}{=} \mathsf{W}_2 \in \mathbb{R}^{1 \times D}$ with $(\mathsf{W}_2)_i = 1$ for each $i = 1, \dots, D$. Therefore,

$$u_0 \overset{\text{def.}}{=} \left( |\sigma_{1,1}(x_i) - \sigma_{1,1}(y_i)| \right)_{i=1}^d = \left( |x_i - y_i| \right)_{i=1}^d$$
$$\therefore \ u_1 \overset{\text{def.}}{=} I_D \sigma_{p,p} \bullet (u_0) = \left( |x_i - y_i|^p \right)_{i=1}^d \tag{20}$$
$$\therefore \ u_2 \overset{\text{def.}}{=} \sigma_{\alpha/p, \alpha/p}(\mathbf{1}_D u_1) = \left( \sum_{i=1}^{D} |x_i - y_i|^p \right)^{\frac{\alpha}{p}} = \|(x)_{1:D} - (y)_{1:D}\|_p^\alpha.$$

Consequentially, the map $\mathcal{D}(x, y) \overset{\text{def.}}{=} u_2$, with $u_2$ defined by equation 20, is of the form of equation 3 and satisfies: for each $x, y \in \mathbb{R}^{D+T}$

$$\mathcal{D}(x, y) = \|(x)_{1:D} - (y)_{1:D}\|_p^\alpha.$$

Observe that $\mathcal{D}$ has depth 2, width $d$, and $2 \cdot 3 + D + D = 2(3 + D)$ non-zero parameters. $\qquad\square$

The following is a more technical version of Theorem 1, which we prove here as it implies the version found in the main text of our manuscript. Note that so far we have used $x, y \in \mathbb{R}^{D+T}$ for our proofs and derivations. Next, we use $x_u, x_v \in \mathbb{R}^N$ to denote the original node features of two given nodes $u$ and $v$. Recall the identification $V \in v \leftrightarrow x_v \in \mathbb{R}^N$ from Section 2.1. $x, y$ used thus far would correspond to the encoded node features $\mathcal{E}(x_u), \mathcal{E}(x_v)$.

**Theorem 3** (Universal Spacetime Embeddings). *Fix $W, N, k \in \mathbb{N}_+$, $K > 0$, and let $(P, \lesssim, d)$ be a $k$-point causal metric space with doubling constant $K$, width $W$, and feature encoding $P \overset{\text{def.}}{=} \{x_v\}_{v \in V} \subseteq \mathbb{R}^N$. There exists a $D \in \mathcal{O}(\log(K))$, $T \overset{\text{def.}}{=} W$, an MLP $\mathcal{E} : \mathbb{R}^N \to \mathbb{R}^{D+T}$, $\mathcal{T} : \mathbb{R}^{D+T} \to \mathbb{R}^T$ with representation equation 4, and $\mathcal{D} : \mathbb{R}^{D+T} \times \mathbb{R}^{D+T} \to [0, \infty)$ with representation equation 3 such that: for each $x_u, x_v \in P$*

  (i) *Order Embedding: $x_u \preccurlyeq x_v$ if and only if $\mathcal{E}(x_u) \lesssim^{\mathcal{T}} \mathcal{E}(x_v)$,*

  (ii) *Metric Embedding: $d(x_u, x_v) \leq \mathcal{D}\big(\mathcal{E}(x_u), \mathcal{E}(x_v)\big) \leq \mathcal{O}\big(\log(K)^5\big) d(x_u, x_v)$,*

*Furthermore, there is some $D \leq k$ such that (ii) can be improved to*

$$d(x_u, x_v) = \mathcal{D}\big(\mathcal{E}(x_u), \mathcal{E}(x_v)\big)$$

*In either case, we have the following parametric complexity estimates:*

  (i) *Geometric Complexity: Together, $\mathcal{D}$ and $\mathcal{T}$ are defined by a total of*

$$D + T + 8$$

  *non-zero parameters,*

  (ii) *Encoding Complexity: $\mathcal{E}$ depends on*

$$\mathcal{O}\left( k^{5/2} D^4 N \log(N) \ \log\left( \frac{k^2 \ \mathrm{diam}(P, d)}{\mathrm{sep}(P, d)} \right) \right)$$

  *non-zero parameters.*

*Consequentially, together, $\mathcal{D}$, $\mathcal{T}$, and $\mathcal{E}$ depend at-most on a total of*

$$\mathcal{O}\left( D + T + k^{5/2}D^4 N \, \log(N) \, \log\left( \frac{k^2 \, \mathrm{diam}(P, d)}{\mathrm{sep}(P, d)} \right) \right)$$

*non-zero parameters.*

*Proof of Theorem 1.* Let $\alpha \in (1/2, 1)$ and $\delta \in (0, 1]$, both of which will be fixed retroactively.

**Step 1 - Causal Embedding:**
By (Dilworth, 1950, Theorem 1.1), $(P, \lesssim)$ has width $W$ only if there exists an order embedding $\tilde{\mathcal{E}}^\star : P \to (\{0, 1\}^W, \lesssim^\times)$. Since the inclusion $\iota$ of $(\{0, 1\}^W, \lesssim^\times)$ into $(\mathbb{R}^W, \lesssim^\times)$ trivially defines an order embedding, then $\mathcal{E}^{(1)} \overset{\mathrm{def.}}{=} \iota \circ \tilde{\mathcal{E}} : (P, \lesssim) \to (\mathbb{R}^W, \lesssim^\times)$ is an order embedding.

**Step 2 (Case I) - Metric Embedding - Low-Distortion Case:**
By Naor and Neiman's Assouad embedding theorem, as formulated in (Naor & Neiman, 2012, Theorem 1.2), there exists an absolute constant $c > 0$ such that for each $1/2 < \alpha < 1$ and $0 < \delta \le 1$, there exists a bi-Lipschitz embedding $\mathcal{E}^{(2)}(P, d^{1-\alpha}) \to (\mathbb{R}^D, \|\cdot\|_2)$ satisfying: for each $x_u, x_v \in P$

$$d^{1-\alpha}(x_u, x_v) \le \|\mathcal{E}^{(2)}(x_u) - \mathcal{E}^{(2)}(x_v)\|_2 \le c\Big(\frac{\log(K)}{1-\alpha}\Big)^{1+\delta} d^{1-\alpha}(x_u, x_v) \tag{21}$$

where[4] $D \in \mathcal{O}(\log(K)/\delta)$. Equivalently, for each $x_u, x_v \in P$

$$d(x_u, x_v) \le \|\mathcal{E}^{(2)}(x_u) - \mathcal{E}^{(2)}(x_v)\|_2^{1/(1-\alpha)} \le c^{1/(1-\alpha)} \Big(\frac{\log(K)}{1-\alpha}\Big)^{\frac{1+\delta}{1-\alpha}} d(x_u, x_v). \tag{22}$$

**Step 2 (Case II) - Metric Embedding - Isometric Case:**

Since $(P, d)$ is such that $P$ is finite, i.e. $k = \#P < \infty$ then, [5] implies that setting $\alpha \overset{\mathrm{def.}}{=} \gamma(k-1)/2 \overset{\mathrm{def.}}{=} \log_2(1 + 1/(k-1))/2 \in (0, 1]$ (with the case where $\alpha = 1$ only being achieved when $P$ is a singleton) there exists some $\tilde{D} \in \mathbb{N}_+$ and $\tilde{\mathcal{E}}^{(2)} : P \to \mathbb{R}^{\tilde{D}}$ such that: for each $x_u, x_v \in P$

$$d(x_u, x_v)^\alpha = \|\tilde{\mathcal{E}}^{(2)}(x_u) - \tilde{\mathcal{E}}^{(2)}(x_v)\|_2. \tag{23}$$

Since $D \overset{\mathrm{def.}}{=} \dim\big(\mathrm{span}\{\tilde{\mathcal{E}}(x_u)\}_{x_u \in P}\big) \le \#P = k$ and since all $D$ dimensional linear of $\mathbb{R}^{\tilde{D}}$ subspaces are isometrically isomorphic to the $D$-dimensional Euclidean space; then there exists some linear surjection $T : \mathbb{R}^{\tilde{D}} \to \mathbb{R}^D$ which restricts to a bijective isomorphism from $\mathrm{span}\{\tilde{\mathcal{E}}(x_u)\}_{x_u \in P}$ to $\mathbb{R}^D$ (both equipped with Euclidean metrics). Then, the composite map $\mathcal{E}^{(2)} \overset{\mathrm{def.}}{=} T \circ \tilde{\mathcal{E}}^{(2)} : (P, d^\alpha) \to (\mathbb{R}^D, \|\cdot\|)$ is an isometric embedding. Consequentially: for each $x_u, x_v \in P$ we have that

$$d(x_u, x_v) = \|\mathcal{E}^{(2)}(x_u) - \mathcal{E}^{(2)}(x_v)\|_2^{1/\alpha} \tag{24}$$

and we emphasize that $D \le k$.

**Step 3 - Interpolation:**
Define the "spacetime embedding" $\mathcal{E} : P \to \mathbb{R}^{D+W}$ by: for each $x_u \in P$

$$\mathcal{E}(x_u) \overset{\mathrm{def.}}{=} \big(\mathcal{E}^{(1)}(x_u), \mathcal{E}^{(2)}(x_u)\big). \tag{25}$$

We memorize/interpolate $\mathcal{E}$ using (Kratsios et al., 2023a, Lemma 20); thus, there exists a ReLU MLP $\hat{\mathcal{E}} : \mathbb{R}^N \to \mathbb{R}^{D+W}$ satisfying: for each $x_u \in P$ we have that

$$\hat{\mathcal{E}}(x_u) = \mathcal{E}(x_u). \tag{26}$$

Furthermore, (Kratsios et al., 2023a, Lemma 20) guarantees that number of trainable parameters defining $\mathcal{E}$ is

$$\mathcal{O}\Big(k^{5/2}D^4 N \, \log(N) \, \log\big(k^2 \, \mathrm{aspect}(P, d)\big)\Big),$$

---

[4]Note that $\varepsilon \overset{\mathrm{def.}}{=} 1 - \alpha$ in the notation of Naor & Neiman (2012).

[5]The isometric embeddability into a Euclidean snowflake characterizes finite metric spaces; see (Le Donne et al., 2018, Corollary 2.2).

where similarly to Krauthgamer et al. (2005) the *aspect ratio* of the finite metric space $(P, d)$ is defined by

$$\text{aspect}(P, d) \stackrel{\text{def.}}{=} \frac{\max_{x_u, x_v \in P} d(x_u, x_v)}{\min_{x_u, x_v \in P \,;\, x_u \neq x_v} d(x_u, x_v)} \stackrel{\text{def.}}{=} \frac{\text{diam}(P, d)}{\text{sep}(P, d)}.$$

Thus, the number of non-zero parameters determining $\mathcal{E}$ is at-most

$$\mathcal{O}\left(k^{5/2} D^4 N \log(N) \log\left(\frac{k^2 \, \text{diam}(P, d)}{\text{sep}(P, d)}\right)\right).$$

Furthermore, by equation 22 (in case I) and by definition of $\mathcal{E}$, we have that: for each $x_u, x_v \in P$

$$d(x_u, x_v) \leq \|\pi \circ \mathcal{E}(x_u) - \pi \circ \mathcal{E}(x_v)\|_2^{1/(1-\alpha)} \leq c^{1/(1-\alpha)} \left(\frac{\log(K)}{1-\alpha}\right)^{\frac{1+\delta}{1-\alpha}} d(x_u, x_v) \qquad (27)$$

$$x_u \lesssim x_v \Leftrightarrow \mathcal{E}(x_u) \lesssim^{\mathcal{T}} \mathcal{E}(x_v). \qquad (28)$$

Retroactively set $\delta \stackrel{\text{def.}}{=} 1/4 \in (0, 1]$ and $\alpha \stackrel{\text{def.}}{=} 3/4 \in (1/2, 1)$. Then, equation 27 and equation 28 become

$$d(x_u, x_v) \leq \|\pi \circ \mathcal{E}(x_u) - \pi \circ \mathcal{E}(x_v)\|_2^{1/(1-\alpha)} \leq \left(1024 \, c^4 \, \log(K)^5\right) d(x_u, x_v) \qquad (29)$$

$$x_u \lesssim x_v \Leftrightarrow \mathcal{E}(x_u) \lesssim^{\mathcal{T}} \mathcal{E}(x_v). \qquad (30)$$

In case II, instead, equation 24 implies that: for each $u, v \in V$

$$d(x_u, x_v) = \|\pi \circ \mathcal{E}(x_u) - \pi \circ \mathcal{E}(x_v)\|_2^{1/\alpha} \qquad (31)$$

$$x_u \lesssim x_v \Leftrightarrow \mathcal{E}(x_u) \lesssim^{\mathcal{T}} \mathcal{E}(x_v). \qquad (32)$$

**Step $4$ - Encoding Product Order in time and snowflake in space as $(\mathcal{T}, \mathcal{D})$**

In either case, since $\mathbb{R}^{D+W}$ is equipped with the product order on its last $W$ (temporal) dimensions and the $1/(1-\alpha)$ (resp. $1/\alpha$)-snowflake of the Euclidean ($\ell^2$) metric on its first $D$ (spatial) dimension, then the conditions for Lemma 1 are met. Applying Lemma 1 concludes the proof. □

*Proof of Proposition 3.* Set $W \in \{1, 2\}$ and $D \in \mathbb{N}_+$. By definition, of the order on $\mathbb{R}^{D+W}$, given in Lemma 1 (i), a spacetime embedding $\mathcal{E} : P \to \mathbb{R}^{D+W}$ exists if and only if $p \circ \mathcal{E} : P \to \mathbb{R}^W$ is an order embedding into $(\mathbb{R}^W, \lesssim^\times)$ where $\lesssim^\times$ is the product order and $p : \mathbb{R}^{D+W} \ni (x_i)_{i=1}^{D+W} \to (x_i)_{i=1}^W \in \mathbb{R}^W$ is the canonical projection. By (Baker et al., 1970, Theorem 6.1) an order embedding of $(P, \lesssim)$ into $(\mathbb{R}^2, \lesssim^\times)$ exists if and only if $(P, \lesssim)$ has a planar Hasse diagram. □

*Proof of Theorem 2.* **Step 1 - Time Embedding**
Since the Hasse diagram of $(P, \lesssim)$ has been assumed to be planar, then (Baker et al., 1970, Theorem 6.1) implies that there exists an other embedding

$$\mathcal{E}^{(1)} : (P, \lesssim) \to (\mathbb{R}^2, \lesssim^\times).$$

**Step 2 - Spatial Embedding**
Since we have assumed that the Hasse diagram of $(P, \lesssim)$ is planar, then we may instead use (Rao, 1999, Theorem 9) to deduce that there exists a $D \in \mathbb{N}_+$ and an injective map $\mathcal{E}^{(2)} : P \to \mathbb{R}^D$ such that: for each $x_u, x_v \in P$

$$\frac{1}{c \sqrt{\log(k)}} d_{\text{H}}(x_u, x_v) \leq \|\mathcal{E}^{(2)}(x_v) - \mathcal{E}^{(2)}(x_u)\|_2 \leq d_{\text{H}}(x_u, x_v), \qquad (33)$$

where $c > 0$ is an absolute constant. Resealing $\mathcal{E}^{(2)}$ by a factor of $c \sqrt{\log(k)}$, and multiplying across equation 33 by $c\sqrt{\log(k)}$, we find that: for each $x_u, x_v \in P$

$$d_{\text{H}}(x_u, x_v) \leq \|\mathcal{E}^{(2)}(x_v) - \mathcal{E}^{(2)}(x_u)\|_2 \leq c \sqrt{\log(k)} \, d_{\text{H}}(x_u, x_v). \qquad (34)$$

Furthermore, by remark at the beginning of (Rao, 1999, Section 4) by the Johnson-Lindenstrauss lemma, as formulated in (Dubhashi & Panconesi, 2009, Theorem 2.1), one may take $k \stackrel{\text{def.}}{=} \#P$ an incur an additional factor of $\tilde{c}\sqrt{\log(k)}$, for an absolute constant $\tilde{c} > 0$, in the distortion in equation 34; that is find that: for each $u, v \in P$

$$d_{\text{H}}(x_u, x_v) \leq \|\mathcal{E}^{(2)}(x_v) - \mathcal{E}^{(2)}(x_u)\|_2 \leq C \log(k) \, d_{\text{H}}(x_u, x_v), \tag{35}$$

where $C \stackrel{\text{def.}}{=} c\tilde{c} > 0$. Set $p \stackrel{\text{def.}}{=} 1$ and recall that $\| \cdot \|_2 \leq \| \cdot \|_1 \leq \tilde{C}\sqrt{\log(k)}\| \cdot \|_2$ on $\mathbb{R}^{\tilde{C}\log(k)}$, for any $\tilde{C} > 0$. Thus, equation 35 implies that: for each $x_u, x_v \in P$ we have

$$d_{\text{H}}(x_u, x_v) \leq \|\mathcal{E}^{(2)}(x_v) - \mathcal{E}^{(2)}(x_u)\|_1^p \leq C' \log(k)^2 \, d_{\text{H}}(x_u, x_v), \tag{36}$$

where $C' \stackrel{\text{def.}}{=} C\tilde{C} > 0$ and $p \stackrel{\text{def.}}{=} 1$.

**Step 3 - Interpolation Embedding**
Pick some $x^\star \in \mathbb{R}^{D+2} \setminus [\cup_{i=1}^2 \mathcal{E}^{(i)}(P)]$. Since $P \subset \mathbb{R}^N$ then consider the map $\mathcal{E} : \mathbb{R}^N \to \mathbb{R}^{D+2}$ defined for each $x \in \mathbb{R}^N$ by

$$\mathcal{E}(x) \stackrel{\text{def.}}{=} \begin{cases} \big(\mathcal{E}^{(1)}(x), \mathcal{E}^{(2)}(x)\big) & \text{if } x \in P \\ x^\star & \text{if } x \notin P. \end{cases}$$

We memorize/interpolate $\mathcal{E}$ using (Kratsios et al., 2023a, Lemma 20) over the finite set $P$. Whence, there exists a ReLU MLP $\hat{\mathcal{E}} : \mathbb{R}^N \to \mathbb{R}^{D+W}$ satisfying: for each $x_u \in P$

$$\hat{\mathcal{E}}(x_u) = \mathcal{E}(x_u). \tag{37}$$

Again, as in the proof of Theorem 1, (Kratsios et al., 2023a, Lemma 20) guarantees that number of trainable parameters defining $\mathcal{E}$ is

$$\mathcal{O}\Big(k^{5/2} D^4 N \log(N) \, \log\big(k^2 \, \text{aspect}(P, d_{\text{H}})\big)\Big),$$

where, in this case, *aspect ratio* of the finite metric space $(P, d_{\text{H}})$ is defined by

$$\text{aspect}(P, d_{\text{H}}) \stackrel{\text{def.}}{=} \frac{\max_{x_u, x_v \in P} d_{\text{H}}(x_u, x_v)}{\min_{x_u, x_v \in P \, ; \, x_u \neq x_v} d_{\text{H}}(x_u, x_v)} \stackrel{\text{def.}}{=} \frac{\text{diam}(P, d_{\text{H}})}{\text{sep}(P, 1)} = \text{diam}(P, d_{\text{H}}),$$

where we used the fact that the minimal edge weights between adjacent distance nodes are equal to 1 in an unweighted graph. Consequentially, the number of non-zero parameters determining $\mathcal{E}$ is

$$\mathcal{O}\Big(k^{5/2} D^4 N \log(N) \, \log\big(k^2 \, \text{diam}(P, d_{\text{H}})\big)\Big).$$

**Step 4 - Encoding Product Order in time and snowflake in space as** $(\mathcal{T}, \mathcal{D})$

Since $\mathbb{R}^{D+2}$ is equipped with the product order on its last two (time) dimensions and the $p \stackrel{\text{def.}}{=} 1/\alpha$-snowflake of the Euclidean $(\ell^2)$ metric on its first $D$ (space) dimension, then the conditions for Lemma 1 are met. Applying Lemma 1 concludes the proof. □

## C  COMPUTATIONAL IMPLEMENTATION OF NEURAL SPACETIMES

In this appendix, we provide an extended discussion and additional details regarding the computational implementation of neural spacetimes. First, we explain what makes an activation function fractalesque, followed by an analysis of the behavior of our proposed activation function for neural (quasi-)metrics and neural partial orders. Moreover, we provide an algorithmic description of the entire pipeline, extend the discussion on causality loss enforcement and time embeddings, and propose optimization strategies at both the local and global geometry levels. Additionally, we explore potential classical algorithms for computing the optimal embedding dimensions, weight initialization strategies for the networks, and the algorithmic differences between neural snowflakes and neural (quasi-)metrics.

## C.1 WHAT MAKES AN ACTIVATION FUNCTION FRACTALESQUE?

The terminology *snowflake* arises as follows. Consider the activation function $\sigma : \mathbb{R} \to \mathbb{R}$ given by $\sigma(x) = |x|^\alpha$ and set $\alpha = \log(4)/\log(3)$. Then, $d_\alpha(x, y) \overset{\text{def.}}{=} \sigma(|x - y|)$ defines a metric on $\mathbb{R}$ and the metric space $(\mathbb{R}, d_\alpha)$ is *isometric* to the (von) Koch snowflake fractal (Figure 4) $\mathcal{X} \subset \mathbb{R}^2$ endowed with the distance obtained by restriction of the Euclidean distance on $\mathbb{R}^2$ to $\mathcal{X}$. Here, $\alpha$ is chosen such that $(\mathcal{X}, d_\alpha)$ has positive and finite $\alpha$-Hausdorff measure.

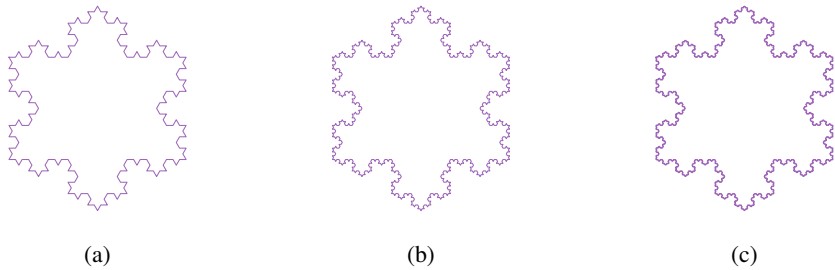

|       |       |       |
| :---: | :---: | :---: |
| (a)   | (b)   | (c)   |

Figure 4: Koch snowflake with increasing number of refinement iterations from left to right.

Indeed, for any general $\alpha \in (0, 1)$ one can show that $d_\alpha$ defines a metric on $\mathbb{R}$ with the property that any line segment has *infinite length*, see Semmes (1996). Thus, intuitively, the snowflake space $(\mathbb{R}, d_\alpha)$ contains infinitely more space to place points in while also maintaining a similar geometry to the Euclidean line; see e.g. (Acciaio et al., 2024, Lemma 7.1). See Tyson & Wu (2005) for an intrinsic characterization of metric spaces which arise as snowflakes of a subset of a (possibly multidimensional) Euclidean space.

More generally, consider a continuous activation function $\sigma : \mathbb{R} \to \mathbb{R}$. Since $\sigma$ is continuous and $[0, 1]$ is compact then $\sigma$ admits a modulus of continuity $\omega : [0, \infty) \to \mathbb{R}$ on $[0, 1]$; i.e. for each $x, y \in \mathbb{R}$

$$|\sigma(x) - \sigma(y)| \le \omega(|x - y|). \tag{38}$$

We think of $\sigma$ as being *fractalesque* if it is sub-Hölder and non-Lipschitz near 0. That is, there is an $\alpha \in (0, 1)$ (note that $\alpha < 1$) and some $L > 0$, such that the right-hand side of equation 38 can be bounded above as

$$|\sigma(x) - \sigma(y)| \le \omega(|x - y|) \le L |x - y|^\alpha. \tag{39}$$

Thus, if the upper-bound in equation 39 holds for an activation function $\sigma$, e.g. our snowflake activation function defined in equation 11, then the Euclidean distance between points in the image of the componentwise application of $\sigma$ are comparable to those of the snowflaked space $(\mathbb{R}^d, \|\cdot\|^\alpha)$; making it fractalesque.

### C.1.1 THE ACTIVATION FUNCTION IN EQUATION 2

In this work, we leverage fractalesque activation functions, which exhibit different training dynamics from typical activation functions used in artificial neural networks. We aim to visualize this type of activations and to gain an intuitive understanding of their nature.

Neural spacetimes leverage the following equation:

$$\sigma_{s,l}(x) \overset{\text{def.}}{=} \begin{cases} \operatorname{sgn}(x) |x|^s & \text{if } |x| < 1 \\ \operatorname{sgn}(x) |x|^l & \text{if } |x| \ge 1, \end{cases}$$

where the sign function $\operatorname{sgn}$ returns 1 for $x \ge 0$ and $-1$ for $x < 0$. Both neural (quasi-)metrics and neural partial orders in neural spacetimes implement variations of this expression.

We visualize the activation using different values for $s$ and $l$ in Figure 5a. As we can see from the plot, the function is antisymmetric about the y-axis, monotonically increasing, and behaves differently depending on the magnitude of the input. The network can learn to optimize $s$ and $l$ alongside linear projection weights, which can route inputs to different regions of the function, to model different scales independently.

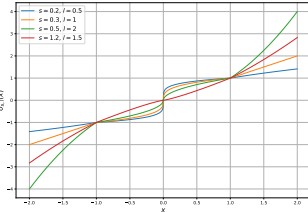
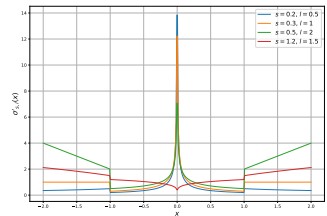

(a) Neural (quasi-)metric activation function used for spatial embeddings (equation 2).

(b) Neural (quasi-)metric activation function derivative.

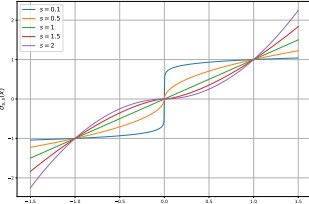
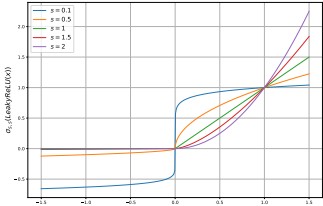

(c) Activation function used by neural partial order without LeakyReLU.

(d) Activation function used by neural partial order (equation 4).

Figure 5: NST activation visualizations.

In Figure 5b we plot a finite difference approximation of its derivative. The derivative rapidly increases for absolute values of the input near 0 for exponents $s < 1$, meaning it has a very high second derivative around these values. This can indirectly destabilize training. Moreover, the absolute value of the derivative itself is also large. This can easily make training unstable as well, especially if we compose this function with itself over multiple layers; when computing backpropagation using the chain rule, the gradients can easily explode.

In the case of the neural partial order, we implement a variation of the activation which is not able to distinguish between small and large scales since $s = l$. Although this would be theoretically enough, and corresponds to the activation in Figure 5c, we empirically found it to be slow at training. Instead, we compose the fractalesque activation with a LeakyReLU (refer back to equation 4), which we find aids optimization and accelerates learning. As we can see in Figure 5d, this results in a clear difference in the behavior of the function on the negative and positive axes of the input. By changing the exponential coefficient, which is learned by gradient descent, the network can substantially alter the behavior of the activation. As before, for exponential coefficients smaller than 1, this activation can lead to instabilities for small input values. For $s = 1$ we recover LeakyReLU.

Based on these observation we restrict all activations to learn $s$ and $l$ coefficients greater than 1 only. These allows us to use fractalesque activations while restristing the functions to behave more similarly to activations used in the literature such as ReLU, LeakyReLU or SiLU functions.

## C.2 ALGORITHMIC DESCRIPTION

We complement the mathematical descriptions of neural spacetimes in Section 3 with the following algorithm summaries. As discussed in the main text, let $N \in \mathbb{N}_+$ be the dimensionality of the feature vectors $x_u, x_v \in \mathbb{R}^N$ associated with nodes $u, v \in V$, where $V$ represents the set of nodes of the digraph $G_D = (E_D, V, W_D)$. Fix a space dimension $D \in \mathbb{N}_+$ and a time dimension $T \in \mathbb{N}_+$. A *neural spacetime* is a learnable triplet $\mathcal{S} = (\mathcal{E}, \mathcal{D}, \mathcal{T})$, where:

- $\mathcal{E} : \mathbb{R}^N \to \mathbb{R}^{D+T}$ is an *encoder network* (MLP),
- $\mathcal{D} : \mathbb{R}^{D+T} \times \mathbb{R}^{D+T} \to [0, \infty)$ is a learnable *quasi-metric* on $\mathbb{R}^D$, and
- $\mathcal{T} : \mathbb{R}^{D+T} \to \mathbb{R}^T$ is a learnable *partial order* on $\mathbb{R}^T$.

In particular, we implement $\mathcal{D}$ and $\mathcal{T}$ as two distinct artificial neural networks inspired by neural snowflakes, which process the space and time dimensions of encoded feature vectors $\hat{x}_u, \hat{x}_v \overset{\text{def.}}{=} \mathcal{E}(x_u), \mathcal{E}(x_v) \in \mathbb{R}^{D+T}$ in parallel.

---

**Algorithm 1:** Neural (quasi-)metric, $\mathcal{D}$

---

**Require:** $\hat{x}_u, \hat{x}_v$            ▷ Two Encoded Node Features Vectors
  **return** $s_{uv}$            ▷ Distance
  $u_0 \leftarrow |\sigma_{s_0, l_0} \bullet (\hat{x}_u)_{1:D} - \sigma_{s_0, l_0} \bullet (\hat{x}_v)_{1:D}|$
    **For** $j = 1$ **to** $J$
   |   $u_j \leftarrow \mathsf{W}_j \sigma_{s_j, l_j} \bullet (u_{j-1})$
       **end**
  $s_{uv} \leftarrow u_J$

---

---

**Algorithm 2:** Neural Partial Order, $\mathcal{T}$

---

**Require:** $\hat{x}_u$            ▷ Encoded Node Feature Vector
  **return** $t_u$            ▷ Temporal Encoding
  $z_0 \leftarrow (\hat{x}_u)_{D+1:D+T}$
    **For** $j = 1$ **to** $J$
   |   $z_j \leftarrow \mathsf{V}_j \sigma_{\tilde{s}_j} \circ \text{LeakyReLU} \bullet (z_{j-1}) + b_j$
       **end**
  $t_u \leftarrow z_J$

---

---

**Algorithm 3:** Neural Spacetime, $\mathcal{S} = (\mathcal{E}, \mathcal{D}, \mathcal{T})$ (Forward Pass)

---

**Require:** $x_u, x_v$            ▷ Two Node Features Vectors
  **return** $s_{uv}, t_u, t_v$            ▷ Distance and Temporal Encodings
  $\hat{x}_u, \hat{x}_v \leftarrow \mathcal{E}(x_u), \mathcal{E}(x_v)$            ▷ Enconde Feature Vectors
  $s_{uv} \leftarrow \mathcal{D}(\hat{x}_u, \hat{x}_v)$            ▷ Compute Distance
  $t_u, t_v \leftarrow \mathcal{T}(\hat{x}_u), \mathcal{T}(\hat{x}_v)$            ▷ Apply Temporal Encoding

---

### C.3   CAUSALITY LOSS AND TIME EMBEDDING

In Section 3.2 we introduced the procedure used to optimize our neural spacetime model. The metric embedding in space is relatively simple an analogous to previous works (Borde & Kratsios, 2023; Kratsios et al., 2023b). In this appendix we expand on the computational approach used for embedding causality and optimizing the time embedding.

The causality loss is given in the main text by:

$$\mathcal{L}_{uv}^C \overset{\text{def.}}{=} A_{uv} \mathcal{L}_C^* \Big( \sum_{t=1}^{T} \text{SteepSigmoid}(\mathcal{T}(\hat{x}_u)_t - \mathcal{T}(\hat{x}_v)_t) \Big),$$

with $\text{SteepSigmoid}(x) = \frac{1}{1+e^{-10x}}$ (the value 10 was found experimentally, making the function too steep can lead to training instabilities), and where $\mathcal{L}_C^*$ slightly modifies the expression (equation 40):

$$\sum_{t=1}^{T} \text{SteepSigmoid}(\mathcal{T}(\hat{x}_u)_t - \mathcal{T}(\hat{x}_v)_t).$$

For the sake of understanding, let us focus on the equation above first.

$$\text{SteepSigmoid}(x) \to 0 \quad \text{as} \quad x \to -\infty.$$

In particular, SteepSigmoid$(x)$ tends to 0 faster than Sigmoid$(x)$ as $x \to -\infty$ (see Figure 6). In asymptotic notation:

$$\lim_{x \to -\infty} \frac{\text{SteepSigmoid}(x)}{\text{Sigmoid}(x)} = 0.$$

Importantly, SteepSigmoid $(\mathcal{T}(\hat{x}_u)_t - \mathcal{T}(\hat{x}_v)_t) \approx 0 \forall (\mathcal{T}(\hat{x}_u)_t - \mathcal{T}(\hat{x}_v)_t) < 0$ even if $|\mathcal{T}(\hat{x}_u)_t - \mathcal{T}(\hat{x}_v)_t|$ is small. As discussed in Section 3.2, the loss for two causally connected events $u \preccurlyeq v$ in the first neighborhood of each other ($A_{uv} = 1$) is minimized when $\hat{x}_u \lesssim^{\mathcal{T}} \hat{x}_v$ is satisfied (equation 5).

In our mathematical formulation, the exact magnitude of the negative difference is not important. At first glance, a more straightforward way of imposing this condition is to use ReLU activation functions instead. However, these are discontinuous and lead to unstable training, which we verified experimentally. Hence, we want a continuous function that is easy to optimize and reaches zero quickly as the partial order is satisfied. Remember that we are optimizing the distance loss for the spatial component of the spacetime embedding and the time embedding using the causality loss simultaneously. If the causality loss does not reach zero quickly enough, we will be wasting computation trying to make the difference between partial embeddings more negative for no reason and, as a consequence, failing to further optimize the metric distortion of the embedding when we have already satisfied causality.

Finally, to ensure that the causality loss provides good gradients while being zero as soon as the partial order is satisfied we use the following expression, which we converged to empirically and that satisfies all our requirements:

$$\mathcal{L}^C_{uv} \overset{\text{def.}}{=} A_{uv} \Big( \sum_{t=1}^T \text{SteepSigmoid}(\mathcal{T}(\hat{x}_u)_t - \mathcal{T}(\hat{x}_v)_t) \Big) \times (1 - \text{TotalCorrect}), \qquad (40)$$

where the second term is not differentiable but makes the loss zero when all the directed edges have been correctly embedded. To compute this we can indeed use the ReLU function:

$$\text{TotalCorrect} \overset{\text{def.}}{=} \frac{\sum_{u=u_1}^{u_M} \sum_{v=v_1}^{v_M} A_{uv} \cdot \mathbb{I} \Big( \sum_{t=1}^T \text{ReLU}(\mathcal{T}(\hat{x}_u)_t - \mathcal{T}(\hat{x}_v)_t) = 0 \Big)}{|E_D|},$$

where the expression in the enumerator counts how many times the function evaluates to zero for connected nodes.

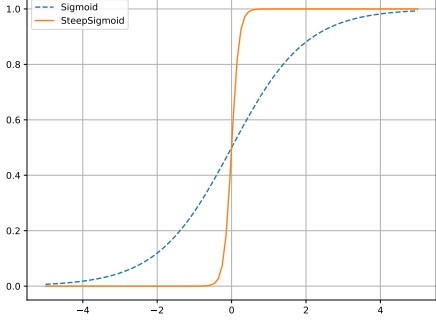

Figure 6: Comparison between (standard) Sigmoid and SteepSigmoid function.

### C.4 How to Theoretically Enforce the Global Causal Geometry

In Section 3.2, we discuss that for practical purposes, we only focus on encoding local geometry, which in turn optimizes the causal geometry of the neural spacetime implicitly for causally connected events due to transitivity. For anti-chains, the *no causality* condition will be satisfied with high probability, especially as the number of time dimensions increases. For completeness, here we provide a loss function to enforce the global causal geometry of DAGs if the causal connectivity in the input geometry was easily computable. Note that for large graphs, this becomes very computationally expensive since we need to verify that for anti-chains, there is no path between those two given nodes.

$$
\mathcal{L}_{uv}^{C} \overset{\text{def.}}{=} \underbrace{A'_{uv}\Big( \sum_{t=1}^{T} \text{ReLU}(\mathcal{T}(\hat{x}_u)_t - \mathcal{T}(\hat{x}_v)_t) \Big)}_{\text{Check: Causality}} + \underbrace{B_{uv}\big( \min_t \big( \text{ReLU}(\varepsilon + \mathcal{T}(\hat{x}_v)_t - \mathcal{T}(\hat{x}_u)_t)\big)\big)}_{\text{Check: No Causality}},
$$
(41)

where $\mathbf{B} \overset{\text{def.}}{=} (I_{(u \npreceq v \wedge v \npreceq u)})_{uv}$ with entries $B_{uv}$ for two events $u$ and $v$, which can be deduced from $\mathbf{A}'$, and where $\varepsilon > 0$ is a margin. $A'_{uv}$ in this case is not the adjacency matrix, but a mask which is $1$ for causally connected nodes or events. Note that here unlike in the main text we are not restricting causality to the first hop. Also, $\mathbf{B}$ is symmetric: $B_{uv} = B_{vu}$.

The loss above enforces causal connectivity and lack of causal connectivity. Both $\hat{x}_u \npreceq^{\mathcal{T}} \hat{x}_v$ and $\hat{x}_v \npreceq^{\mathcal{T}} \hat{x}_u$ must be satisfied by our representation for anti-chains. $T > 1$ is required, otherwise it is not possible to avoid causality in at least one direction. In the simplest case, when $T = 2$ and $t \in \{t_1, t_2\}$ the following must be satisfied if we associate $\hat{x}_u$ with $u$ and $\hat{x}_v$ with $v$: $\mathcal{T}(\hat{x}_v)_{t_1} > \mathcal{T}(\hat{x}_u)_{t_1}$ and $\mathcal{T}(\hat{x}_u)_{t_2} > \mathcal{T}(\hat{x}_v)_{t_2}$, or $\mathcal{T}(\hat{x}_v)_{t_2} > \mathcal{T}(\hat{x}_u)_{t_2}$ and $\mathcal{T}(\hat{x}_u)_{t_1} > \mathcal{T}(\hat{x}_v)_{t_1}$. To satisfy the no causality condition when $T > 2$, as long as one of the time dimensions breaks $\preceq^{\mathcal{T}}$ in equation 5 it suffices. $\varepsilon$ would be included in this hypothetical optimization objective to avoid the collapse of the encoder network to a single embedding. Lastly, we would like to highlight that this causality loss should be optimized alongside a distance loss based on the graph geodesic distance instead of the local distance. Although this is a theoretical exercise, in practice it may be best to substitute $\text{ReLU}$ with $\text{SteepSigmoid}$ for optimization purposes as in Section C.3.

### C.5 Potential Classical Algorithms to Compute Space and Time Dimensions

Theorem 1 shows that the number of time dimensions should be at least equal to the width of the poset being embedded by the NST. In theory, the width of a poset could be computed in polynomial time (Felsner et al., 2003). Furthermore, for posets with a width of less than $4$, there are algorithms with a runtime of $\mathcal{O}(n \log(n))$ (sub-quadratic time) that can detect if the poset has a width of at most $4$. Nevertheless this theoretical results become impractical for large graphs. Similarly, Theorem 1 shows that the required number of spatial dimensions needed to obtain a faithful low-distortion metric embedding of the graph underlying a poset is determined by its doubling constant. It is known that an exact computation of this doubling dimension is generally NP-hard Gottlieb & Krauthgamer (2013). Alternatives algorithms to obtain accurate upper bounds (up to an absolute constant factor) are also discussed in the literature (Har-Peled & Mendel, 2005), but these are in general impractical for large DAGs.

### C.6 Weight Initialization and Handling the Absence of Normalization

Weight initialization is important to avoid initial network predictions from shrinking or exploding. Networks with fractalesque activations are particularly susceptible to this problem, as they use activations with exponential coefficients. This is not such a prevalent issue if we use the well-established ReLU activation or its variants (ELU, LeakyReLU, etc.), since this activations only zero out part of the axis and leave the rest of the transformation linear.

In the original neural snowflake model (equation 10), the matrices $A$, $B$, and $C$ are initialized with weights sampled from a uniform distribution spanning from $0$ to $1$, which are then normalized according to the dimensions of each matrix. Specifically, for matrix $A$, the weights are drawn from a distribution ranging between $0$ and $1/(d_{A1}d_{A2})$, where $d_{A1}$ represents the number of rows and

$d_{A2}$ represents the number of columns of matrix $A$. We follow the same approach to initialize $W$ in equation 3. While better initialization alternatives may be possible, it was noted previously in (Borde & Kratsios, 2023) that drawing the weights from a normal Gaussian or using Xavier initialization can lead to instabilities and exploding numbers in the forward pass. We also observe that this initialization technique is important to avoid the metric prediction becoming either too small or too large when making the network deeper since we cannot naively apply normalization layers inside our parametric representation of the metric. Additionally, we find that neural (quasi-)metrics are better than the original neural snowflakes in maintaining the same order of magnitude in its distance prediction as the number of layers is increased.

## C.7 NEURAL SNOWFLAKES VS NEURAL (QUASI-)METRICS

Next, we discuss similarities and differences between the original neural snowflake and neural (quasi-)metrics. We first provide an algorithmic description of neural snowflakes, which can be used to compare against that presented in Appendix C.2 for the neural (quasi-)metric model.

---

**Algorithm 4:** Neural Snowflake

---

**Require:** $\hat{x}_u, \hat{x}_v$          ▷ Two Node Features Vectors
    **return** $s_{ij}$          ▷ Distance similarity measure
    $u_0 \leftarrow ||\hat{x}_u - \hat{x}_v||_2$          ▷ Euclidean Distance
    **For** $j = 1$ **to** $J$

        $\hat{u}_{j-1} \leftarrow \mathsf{A}_j u_{j-1}$          ▷ Linear Projection
        $\Sigma_j \leftarrow \psi \hat{u}_{j-1}$          ▷ Snowflake Activation
        $u_j \leftarrow \mathsf{B}_j \Sigma_j \mathsf{C}_j$          ▷ Linear Projections
       **end**

    $s_{ij} \leftarrow u_J^{1+|p|}$          ▷ Quasi-metric

---

**Embedding Experiments.** In terms of experiments, generally, we find that neural (quasi-)metrics have a smoother optimization landscape. They tend to converge faster for a low number of epochs, and in the case of snowflakes, we experimentally observed sudden drops in MSE loss, similar to those reported in Appendix I of (Borde & Kratsios, 2023). As shown in Section D.1, the neural (quasi-)metric construction performs better at tree embedding. What we observed is that if trained for a sufficient duration, snowflakes will eventually find a good local minimum. On the other hand, our new model seems more reliable in terms of optimization and achieves better performance for lower epochs.

**Optimizing exponents.** In Equation 11, $a$ and $b$ are in principle presented as learnable components via gradient descent. However, in practice, exponents can be unstable to optimize via back-prop (Borde & Kratsios, 2023). The original neural snowflake fixes $a = b = 1$, to avoid optimization issues. Nevertheless note that $p$, which controls the quasi-metric in Equation 10 is indeed optimized. Optimizing exponents closer to the final computations of the network is in general more stable. We hypothesize that this could be due to the fact that there are less chain-rule multiplications, reducing the likelihood of gradient explosion. We observe similar behaviour for neural (quasi-)metrics, note that our activation function relies on learning the exponents (equation 2). In general we find that we can learn the exponents for all layers in this configuration. But we also experimented with trying to approximate fractal metrics and observed that under some extreme cases it may be better to only optimize the exponents of the last layer for stability.

## D EXPERIMENTAL DETAILS

In this appendix we expand on the experimental procedure used to obtain the results presented in the main text in Section 4. Additionally, we present more validation experiments, including undirected tree embeddings, synthetic DAG embeddings with different metrics and varying levels of connectivity, and large DAG citation network embeddings.

D.1 UNDIRECTED TREE EMBEDDING

**Overview of Results for Preliminary Undirected Embeddings.** Before proceeding to DAGs, following Kratsios et al. (2023b), we first test the spatial component of the NST model, on embedding trees and compare it to Euclidean embeddings implemented by an MLP and to hyperbolic embeddings via HNNs Ganea et al. (2018). We use the same training procedure as in Kratsios et al. (2023b) and perform experiments for both binary and ternary trees. In the case of the snowflake geometry, inputs are mapped using an MLP, and the metric induced by the tree structure is learned by a neural (quasi-)metric with 4 layers operating on the embedding space dimensionality. We measure distortion as the ratio between true and predicted distances. Consistent with the theoretical results in (Gupta, 2000), we find that in general, the difference in average distortion between metric spaces is not as pronounced as the maximum distortion since most metric spaces are "treelike" on average. Indeed, we find that our proposed model better restricts the maximum distortion and that it is able to achieve an average distortion of 1.00 even with an embedding space of dimension 2 only. For completeness we also test the neural snowflake model in the same task, which although it is able to minimize the MSE loss, the (max) distortion is very high. Additional discussion comparing neural snowflakes and neural (quasi-)metrics can be found in Appendix C.7.

Table 3: Tree Embedding distortion leveraging Euclidean, Hyperbolic, Neural Snowflake and Neural (Quasi-)metric spaces.

| Tree Type | Embedding dim | Geometry | Avg Distortion | St dev Distortion | Max Distortion | MSE |
|-----------|---------------|----------|----------------|-------------------|----------------|-----|
| Binary | 2 | Euclidean | 1.66 | 3.53 | 1224.17 | 26.27 |
| | | Hyperbolic | 1.04 | 1.61 | 402.52 | 12.76 |
| | | Neural (Quasi-)Metric | 1.00 | 0.23 | 3.09 | 10.09 |
| | | Snowflake | 1.01 | 3.01 | 2261.35 | 9.47 |
| | 4 | Euclidean | 1.15 | 0.68 | 159.74 | 10.19 |
| | | Hyperbolic | 1.00 | 0.17 | 11.03 | 4.14 |
| | | Neural (Quasi-)Metric | 1.00 | 0.19 | 3.87 | 4.88 |
| | | Snowflake | 1.00 | 0.65 | 539.71 | 5.92 |
| Ternary | 2 | Euclidean | 1.69 | 3.17 | 602.96 | 11.55 |
| | | Hyperbolic | 1.09 | 1.23 | 135.81 | 5.55 |
| | | Neural (Quasi-)Metric | 1.00 | 0.20 | 6.33 | 3.34 |
| | | Snowflake | 1.01 | 4.78 | 4017.99 | 3.64 |
| | 4 | Euclidean | 1.17 | 0.82 | 185.73 | 4.82 |
| | | Hyperbolic | 1.00 | 0.15 | 16.56 | 1.37 |
| | | Neural (Quasi-)Metric | 1.00 | 0.16 | 4.19 | 1.72 |
| | | Snowflake | 1.00 | 0.47 | 237.88 | 2.15 |

**Experimental Procedure for Undirected Tree Embeddings.** Neural spacetimes provide guarantees in terms of global and local embeddings. In general, from a computational perspective, local embedding is more tractable. However, in these particular preliminary experiments for metric embedding, similar to (Kratsios et al., 2023b), we embed the full undirected tree geometry. The distance between any two nodes $u, v \in V$ simplifies to the usual shortest path distance on an unweighted graph

$$d_{\mathcal{T}}(u, v) = \inf \left\{ i \ : \ \exists \{v, v_1\}, \dots \{v_{i-1}, u\} \in \mathcal{E} \right\}. \tag{42}$$

Using the expression above, we compute the tree induced graph geodesic between nodes. We work with both binary and ternary trees with all edge weights being equal to 1. To find all-pairs shortest path lengths we use Floyd's algorithm (Floyd, 1962). Note that Floyd's algorithm has a running time complexity of $\mathcal{O}(|V|^3)$ and running space of $\mathcal{O}(|V|^2)$. This makes it not scalable for large graphs.

The networks receive the $x_u$ and $y_u$ coordinates of the graph nodes in $\mathbb{R}^2$ and are tasked with mapping them to a new embedding space which must preserve the distance. An algorithm is employed to generate input coordinates, mimicking a force-directed layout of the tree. In this simulation, edges act as springs, drawing nodes closer, while nodes behave as objects with repelling forces, similar to anti-gravity effects. This process continues iteratively until the positions reach equilibrium. The algorithm can be replicated using the `NetworkX` library and the `spring layout` for the graph.

The networks need to find an appropriate way to gauge the distance. We adjust the network parameters by computing the Mean Squared Error (MSE) loss, which compares the actual distance between nodes given by the tree topology, $d_{\text{true}}$, to the distance predicted by the network mappings, $d_{\text{pred}}$:

$$\text{Loss} = \text{MSE}(d_{\text{true}}, d_{\text{pred}}).$$

For the Multi-Layer Perceptron (MLP) baseline, the predicted distance is calculated as:

$$d_{\text{pred}} = \|\text{MLP}(x_u, y_u) - \text{MLP}(x_v, y_v)\|_2,$$

where $(x_u, y_u)$ and $(x_v, y_v)$ represent the coordinates in $\mathbb{R}^2$ of a synthetically generated tree. On the other hand, for the HNN, we utilize the hyperbolic distance between embeddings with fixed curvature of -1:

$$d_{\text{pred}} = d_{-1}(\text{HNN}(x_u, y_u), \text{HNN}(x_v, y_v)).$$

In the case of HNNs, we employ the hyperboloid model, incorporating an exponential map at the pole to map the representations to hyperbolic space, as detailed in Borde et al. (2023b). Notably, we do not even need any hyperbolic biases to discern the performance disparity between MLP and HNN models (Kratsios et al., 2023b).

Note that in the previous two baselines, the geometry of the space is effectively fixed, and we only optimize the location of the node embeddings in the manifold. In the case of neural snowflakes, the metric is also parametric and optimized via gradient descent. This means that we change both the location of events in the manifold and the geometry of the manifold itself during optimization. Following (Borde & Kratsios, 2023), the neural (quasi-)metric model also leverages the encoder $\mathcal{E}$ which is implemented as an MLP (same as in the Euclidean baseline). The predicted distance is

$$d_{\text{pred}} = d_{NQM}(\text{MLP}(x_u, y_u), \text{MLP}(x_v, y_v)),$$

where $d_{NQM}$ is parametrized by a neural (quasi-)metric network. Importantly, feature vectors are passed independently to the metric learning network (unlike in (Borde & Kratsios, 2023) where an intermediate Euclidean distance representation is fed instead).

In terms of hyperparameters, we generate trees with 1000 nodes and embed them into features of dimensionality 2 and 4 in different experiments. We employ a batch size of 10,000 to learn the distances, train for 10 epochs with a learning rate of $3 \times 10^{-3}$ and AdamW optimizer, and apply a max gradient norm of 1. All encoders have a total of 10 hidden layers with 100 neurons and a final projection layer to the embedding dimension. The neural (quasi-)metric has a total of 4 layers, with a hidden layer dimension equal to the event embedding dimensions, that is, either 2 or 4, and the last layer projects the representation to a scalar, i.e., the predicted distance. We use the same configuration for the snowflakes.

## D.2 METRIC DAG EMBEDDING

In this appendix, we provide additional details on the synthetic weighted DAG embedding experiments described in Section 4, focusing on local embedding. Similar to the previous setup, the networks receive the $x_u$ and $y_u$ coordinates of the graph nodes in $\mathbb{R}^2$ and are tasked with mapping these coordinates to a new embedding space that preserves both the distance and directionality of the generated DAG. We train the embedding following Section 3.2. As discussed in the main text, we optimize the spacetime embedding by running gradient descent based on both a distance and a causality loss. The procedure used to train the neural (quasi-)metric in terms of the spatial embedding is identical to that described in Appendix D.1.

We use the `Graphviz dot layout` algorithm to generate the input coordinates, positioning the nodes from top to bottom according to the directionality of the DAG. This process can be replicated using the `NetworkX` library. Additionally, the coordinates are normalized to lie between 0 and 1. The DAG is constructed from a random graph where each node has a 0.9 probability of being connected to other nodes, defining the directionality and connectivity of the DAG (see Figure 7). The metric distance is then calculated based on the spatial coordinates of the nodes after they have been embedded in the plane. The specific distance functions used in the experiments are listed in Table 1 and are based on (Borde & Kratsios, 2023).

In terms of hyperparameters, we generate DAGs with 50 nodes and embed them into spacetimes of $D = T = 2, 4, 10$. The 50 nodes induce $50^2$ possible distances. We do not apply mini-batching; we optimize all of them at the same time. We train for 5,000 epochs with a learning rate of $10^{-4}$ using the AdamW optimizer, and apply a max gradient norm of 1. The initial encoders have a total of 10 hidden layers with 100 neurons each and a final projection layer to the embedding dimension $D + T$. The neural (quasi-)metric and the neural partial order have a total of 4 layers, each with a hidden layer dimension of 10.

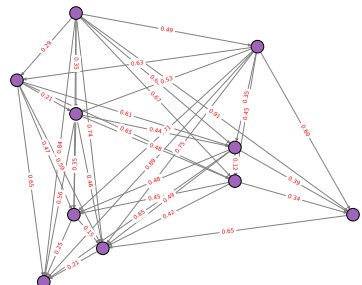

Figure 7: Example DAG with 10 nodes and using the first metric in Table 1.

### D.2.1 VARYING GRAPH CONNECTIVITY

We conduct additional experiments to analyze the effect of graph topology on the embedding results. Specifically, we fix the synthetic metric to $\|\mathbf{x}-\mathbf{y}\|^{0.5} \log(1+\|\mathbf{x}-\mathbf{y}\|)^{0.5}$ and experiment with graphs having varying connection probabilities, which control the sparsity of the random DAGs, i.e., their connectivity. Across all geometries, we observe that as the DAG becomes more sparsified, its directionality is captured less accurately. However, it also becomes easier to achieve lower distortion, as expected, due to the reduction in the number of edges to encode, see Table 4.

Table 4: Embedding results for arxiv citation network.

| Connectivity | Embed. Dim | Neural Spacetime Distortion | Neural Spacetime Max Distortion | Neural Spacetime Directionality | Minkowski Distortion | Minkowski Max Distortion | Minkowski Directionality | De Sitter Distortion | De Sitter Max Distortion | De Sitter Directionality |
|---|---|---|---|---|---|---|---|---|---|---|
| 90% | 2 | $1.09 \pm 0.24$ | 3.18 | 1.00 | $2.86 \pm 5.22$ | 72.66 | 0.99 | $\infty \pm$ | $\infty$ | 0.99 |
| | 4 | $1.02 \pm 0.06$ | 1.51 | 1.00 | $1.70 \pm 2.77$ | 71.09 | 0.99 | $-4.33 \pm 816.47$ | 10,235.71 | 0.99 |
| | 10 | $1.00 \pm 0.03$ | 1.24 | 1.00 | $1.21 \pm 1.33$ | 35.58 | 0.99 | $288.17 \pm 9,794.97$ | 324,027.5 | 0.99 |
| 50% | 2 | $1.14 \pm 0.42$ | 3.94 | 1.00 | $3.35 \pm 10.44$ | 215.65 | 1.00 | $-238.02 \pm 5,432.33$ | 13,879.30 | 0.99 |
| | 4 | $1.01 \pm 0.07$ | 1.38 | 1.00 | $1.27 \pm 0.82$ | 12.45 | 1.00 | $-12.61 \pm 2,253.31$ | 43,124.32 | 0.99 |
| | 10 | $1.00 \pm 0.18$ | 1.06 | 1.00 | $1.02 \pm 0.19$ | 4.03 | 1.00 | $-53.30 \pm 1,600.69$ | 18,756.20 | 0.99 |
| 10% | 2 | $1.00 \pm 0.30$ | 2.65 | 0.98 | $1.01 \pm 2.41$ | 15.96 | 0.93 | $-1.18 \pm 19.43$ | 69.29 | 0.92 |
| | 4 | $1.00 \pm 0.01$ | 1.01 | 1.00 | $1.00 \pm 0.02$ | 1.03 | 0.96 | $1.09 \pm 14.32$ | 130.62 | 0.91 |
| | 10 | $1.00 \pm 0.00$ | 1.00 | 0.98 | $1.00 \pm 0.02$ | 1.08 | 0.96 | $2.39 \pm 20.30$ | 176.60 | 0.93 |

### D.3 REAL-WORLD NETWORK EMBEDDING

**WebKB datasets.** In these datasets (Rozemberczki et al., 2021), nodes represent web pages, while edges denote hyperlinks between them. Each node's features are given by the bag-of-words representation of its web page. We summarize the datasets in Table 5.

As discussed in the main text in Section 4, these datasets are not pure DAGs. As a preprocessing step, we remove self-loops. However, we do not remove cycles, nor do we further process edge directionality. This explains why in Table 1, it is not possible to get a perfect embedding in time. The local metric between nodes is computed as the cosine similarity between the node feature embeddings (since the datasets do not have edge weights). Note that here we do not embed graphs in the plane; we already have bag-of-words features that can be passed to the encoder, unlike in the

Table 5: Statistics of WebKB datasets

| Name | #nodes | #edges |
|------|--------|--------|
| Cornell | 183 | 298 |
| Texas | 183 | 325 |
| Wisconsin | 251 | 515 |

synthetic DAG experiments. In terms of training, the procedure is the same as in Appendix D.2 and using a learning rate of $10^{-4}$.

Table 6: Statistics of Dream5 datasets

| Name | #nodes | #edges |
|------|--------|--------|
| In silico | 1,565 | 4,012 |
| Escherichia coli | 1,081 | 2,066 |
| Saccharomyces cerevisiae | 1,994 | 3,940 |

**Dream5 datasets.** The Dream5 datasets (Marbach et al., 2012) were originally introduced as part of the Network Inference Challenge, where participants were provided with four microarray compendia and tasked with deducing the structure of the underlying transcriptional (gene) regulatory networks. Each compendium encompasses numerous microarray experiments, spanning various genetic, drug, and environmental alterations (or simulations thereof, in the case of the in-silico network). In this work, instead of focusing on network structure prediction, we test our neural spacetime algorithm in the context of graph embedding, following the spacetime representation literature (Law & Lucas, 2023; Sim et al., 2021). We embed the graph topology in $\mathbb{R}^2$ using `Graphviz dot layout` and compute the strength between the connections based on the cosine similarity, since the original networks do not have weights. In Table 6 we record the number of nodes and edges for each dataset. Note that these are not pure DAGs either (but it is possible to embed over $99\%$ of the directed edges correctly, see Table 10). For training, we follow Appendix D.2 with learning rate $10^{-4}$.

**Ogbn-arxiv Large Citation Graph Embedding.** Given that the previous graphs are relatively small, we conduct an additional experiment on the ogbn-arxiv dataset (Hu et al., 2021), which is orders of magnitude larger (see Table 7). We use a similar procedure as before: we remove self-loops to avoid returning similarity scores of 1.0 and also remove edges between nodes whose feature vectors have a cosine similarity score greater than 0.99. In this particular dataset, we found that some nodes had almost identical features, which, due to numerical truncation when computing the distance between nodes, resulted in a distance of zero and led to numerical instabilities. This resulted in 269 rejected edges out of 1,166,243, or $0.02\%$.

Table 7: Statistics of Ogbn-arxiv dataset

| Name | #nodes | #edges |
|------|--------|--------|
| Arxiv | 169,343 | 1,166,243 |

We train all models for 40 epochs, using the AdamW optimizer with a learning rate of $10^{-4}$ and a batch size of 10,000. We report the results in Table 8 below. Once again, we observe that, under the same training conditions, NSTs are superior at embedding compared to their fixed geometry counterparts. However, we observe a degradation in performance compared to previous experiments with smaller graphs. Additionally, note that we do not modify the parameter count of the neural networks.

Table 8: Embedding results for arxiv citation network.

| Dataset | Embed. Dim | Neural Spacetime | | | Minkowski | | | De Sitter | | |
|---------|-----------|------------|----------------|----------------|------------|----------------|----------------|------------|----------------|----------------|
| | | Distortion | Max Distortion | Directionality | Distortion | Max Distortion | Directionality | Distortion | Max Distortion | Directionality |
| Arxiv | 2 | $1.88 \pm 4.37$ | 1,031.11 | 0.80 | $4.64 \pm 7,322.96$ | 1,811,858.75 | 0.83 | $-133.12 \pm 16,142.16$ | 1,845,115.5 | 0.77 |

## D.4 ADDITIONAL RESULTS

In this appendix, we provide extended tables of the results presented in the main text.

Table 9: DAG embedding results. Embedding dimension $D = T = 2, 4, 10$.

| | *Neural Spacetime* | | | |
|---|---|---|---|---|
| **Metric** | **Embedding Dim** | **Distortion (average $\pm$ stand dev)** | **Max Distortion** | **Directionality** |
| $\|\mathbf{x} - \mathbf{y}\|^{0.5} \log(1 + \|\mathbf{x} - \mathbf{y}\|)^{0.5}$ | 2 | $1.13 \pm 0.37$ | 5.82 | 1.0 |
| | 4 | $1.02 \pm 0.06$ | 1.34 | 1.0 |
| | 10 | $1.00 \pm 0.02$ | 1.28 | 1.0 |
| $\|\mathbf{x} - \mathbf{y}\|^{0.1} \log(1 + \|\mathbf{x} - \mathbf{y}\|)^{0.9}$ | 2 | $1.16 \pm 0.45$ | 6.21 | 1.0 |
| | 4 | $1.02 \pm 0.07$ | 1.75 | 1.0 |
| | 10 | $1.00 \pm 0.04$ | 1.47 | 1.0 |
| $1 - \frac{1}{(1 + \|\mathbf{x} - \mathbf{y}\|^{0.5})}$ | 2 | $1.53 \pm 1.25$ | 14.20 | 1.0 |
| | 4 | $1.10 \pm 0.38$ | 5.81 | 1.0 |
| | 10 | $1.01 \pm 0.06$ | 1.63 | 1.0 |
| $1 - \exp \frac{-(\|\mathbf{x} - \mathbf{y}\| - 1)}{\log(\|\mathbf{x} - \mathbf{y}\|)}$ | 2 | $1.51 \pm 1.18$ | 13.55 | 1.0 |
| | 4 | $1.11 \pm 0.41$ | 8.92 | 1.0 |
| | 10 | $1.01 \pm 0.05$ | 1.31 | 1.0 |
| $1 - \frac{1}{1 + \|\mathbf{x} - \mathbf{y}\|^{0.2} + \|\mathbf{x} - \mathbf{y}\|^{0.5}}$ | 2 | $1.60 \pm 1.42$ | 17.07 | 1.0 |
| | 4 | $1.13 \pm 0.45$ | 4.44 | 1.0 |
| | 10 | $1.00 \pm 0.05$ | 1.44 | 1.0 |
| | *Minkowski* | | | |
| **Metric** | **Embedding Dim** | **Distortion (average $\pm$ stand dev)** | **Max Distortion** | **Directionality** |
| $\|\mathbf{x} - \mathbf{y}\|^{0.5} \log(1 + \|\mathbf{x} - \mathbf{y}\|)^{0.5}$ | 2 | $2.86 \pm 5.22$ | 72.66 | 0.99 |
| | 4 | $1.70 \pm 2.77$ | 71.09 | 0.99 |
| | 10 | $1.21 \pm 1.33$ | 35.58 | 0.99 |
| $\|\mathbf{x} - \mathbf{y}\|^{0.1} \log(1 + \|\mathbf{x} - \mathbf{y}\|)^{0.9}$ | 2 | $6.77 \pm 133.68$ | 1669.83 | 0.99 |
| | 4 | $1.70 \pm 5.21$ | 77.03 | 0.99 |
| | 10 | $1.19 \pm 1.09$ | 25.18 | 0.99 |
| $1 - \frac{1}{(1 + \|\mathbf{x} - \mathbf{y}\|^{0.5})}$ | 2 | $21.89 \pm 759.05$ | 18753.57 | 0.99 |
| | 4 | $2.87 \pm 20.17$ | 643.71 | 0.99 |
| | 10 | $1.17 \pm 1.26$ | 37.42 | 0.99 |
| $1 - \exp \frac{-(\|\mathbf{x} - \mathbf{y}\| - 1)}{\log(\|\mathbf{x} - \mathbf{y}\|)}$ | 2 | $11.37 \pm 114.98$ | 1876.54 | 0.98 |
| | 4 | $2.49 \pm 8.72$ | 198.04 | 0.98 |
| | 10 | $1.18 \pm 2.49$ | 82.67 | 0.99 |
| $1 - \frac{1}{1 + \|\mathbf{x} - \mathbf{y}\|^{0.2} + \|\mathbf{x} - \mathbf{y}\|^{0.5}}$ | 2 | $20.65 \pm 292.90$ | 5471.07 | 0.96 |
| | 4 | $2.83 \pm 9.54$ | 153.91 | 0.98 |
| | 10 | $1.13 \pm 0.91$ | 28.45 | 0.99 |
| | *De Sitter* | | | |
| **Metric** | **Embedding Dim** | **Distortion (average $\pm$ stand dev)** | **Max Distortion** | **Directionality** |
| $\|\mathbf{x} - \mathbf{y}\|^{0.5} \log(1 + \|\mathbf{x} - \mathbf{y}\|)^{0.5}$ | 2 | $\infty \pm$ | $\infty$ | 0.99 |
| | 4 | $-4.33 \pm 816.47$ | 10235.71 | 0.99 |
| | 10 | $288.17 \pm 9794.97$ | 324027.5 | 0.99 |
| $\|\mathbf{x} - \mathbf{y}\|^{0.1} \log(1 + \|\mathbf{x} - \mathbf{y}\|)^{0.9}$ | 2 | $9.40 \pm 2226.84$ | 63968.21 | 0.99 |
| | 4 | $174.69 \pm 3637.32$ | 115851.88 | 0.99 |
| | 10 | $-10.66 \pm 739.71$ | 8997.62 | 0.99 |
| $1 - \frac{1}{(1 + \|\mathbf{x} - \mathbf{y}\|^{0.5})}$ | 2 | $-63.56 \pm 4438.15$ | 43424.54 | 0.94 |
| | 4 | $\infty \pm$ | $\infty$ | 0.94 |
| | 10 | $-333.23 \pm 8701.18$ | 130111.80 | 0.94 |
| $1 - \exp \frac{-(\|\mathbf{x} - \mathbf{y}\| - 1)}{\log(\|\mathbf{x} - \mathbf{y}\|)}$ | 2 | $-183.71 \pm 9600.71$ | 39648.66 | 0.94 |
| | 4 | $83.04 \pm 4313.82$ | 97524.21 | 0.94 |
| | 10 | $418.26 \pm 6599.80$ | 150543.73 | 0.94 |
| $1 - \frac{1}{1 + \|\mathbf{x} - \mathbf{y}\|^{0.2} + \|\mathbf{x} - \mathbf{y}\|^{0.5}}$ | 2 | $-542.39 \pm 22058.97$ | 114491.45 | 0.94 |
| | 4 | $\infty \pm$ | $\infty$ | 0.94 |
| | 10 | $\infty \pm$ | $\infty$ | 0.94 |

Table 10: Embedding results for real-world web page hyperlink graph datasets and gene regulatory networks.

| | *Neural Spacetime* | | | |
|---|---|---|---|---|
| **Dataset** | **Embedding Dim** | **Distortion (avg. $\pm$ sdev.)** | **Max Distortion** | **Directionality** |
| Cornell | 2 | $1.00 \pm 0.07$ | 1.31 | 0.93 |
| | 4 | $1.00 \pm 0.04$ | 1.08 | 0.94 |
| | 10 | $1.00 \pm 0.04$ | 1.08 | 0.94 |
| Texas | 2 | $1.01 \pm 0.10$ | 2.27 | 0.89 |
| | 4 | $1.00 \pm 0.01$ | 1.05 | 0.90 |
| | 10 | $1.00 \pm 0.00$ | 1.00 | 0.90 |
| Wisconsin | 2 | $1.00 \pm 0.10$ | 1.67 | 0.89 |
| | 4 | $1.00 \pm 0.04$ | 1.16 | 0.89 |
| | 10 | $1.00 \pm 0.04$ | 1.20 | 0.89 |
| In silico | 2 | $1.06 \pm 0.47$ | 18.54 | 1.00 |
| | 4 | $1.00 \pm 0.09$ | 1.73 | 1.00 |
| | 10 | $1.00 \pm 0.05$ | 1.32 | 1.00 |
| Escherichia coli | 2 | $1.02 \pm 0.45$ | 15.37 | 1.00 |
| | 4 | $1.00 \pm 0.06$ | 2.62 | 1.00 |
| | 10 | $1.00 \pm 0.05$ | 1.17 | 1.00 |
| Saccharomyces cerevisiae | 2 | $1.05 \pm 0.34$ | 10.18 | 1.00 |
| | 4 | $1.00 \pm 0.07$ | 1.63 | 1.00 |
| | 10 | $1.00 \pm 0.05$ | 1.57 | 1.00 |

| | *Minkowski* | | | |
|---|---|---|---|---|
| **Dataset** | **Embedding Dim** | **Distortion (avg. $\pm$ sdev.)** | **Max Distortion** | **Directionality** |
| Cornell | 2 | $1.07 \pm 0.70$ | 9.43 | 0.94 |
| | 4 | $1.00 \pm 0.00$ | 1.01 | 0.94 |
| | 10 | $1.00 \pm 0.00$ | 1.00 | 0.94 |
| Texas | 2 | $1.12 \pm 1.73$ | 31.27 | 0.90 |
| | 4 | $1.00 \pm 0.00$ | 1.00 | 0.90 |
| | 10 | $1.01 \pm 0.01$ | 1.05 | 0.90 |
| Wisconsin | 2 | $5.07 \pm 65.99$ | 1410.03 | 0.90 |
| | 4 | $1.00 \pm 0.04$ | 1.19 | 0.90 |
| | 10 | $1.13 \pm 0.69$ | 16.28 | 0.90 |
| In silico | 2 | $105.42 \pm 4671.85$ | 209248.72 | 0.94 |
| | 4 | $0.25 \pm 54.57$ | 1315.76 | 0.95 |
| | 10 | $1.00 \pm 0.05$ | 3.69 | 0.99 |
| Escherichia coli | 2 | $-4.25 \pm 149.61$ | 438.34 | 0.97 |
| | 4 | $1.00 \pm 0.01$ | 1.08 | 0.98 |
| | 10 | $1.00 \pm 0.01$ | 1.01 | 0.99 |
| Saccharomyces cerevisiae | 2 | $-2.38 \pm 173.57$ | 151.43 | 0.91 |
| | 4 | $1.04 \pm 2.25$ | 63.17 | 0.98 |
| | 10 | $1.01 \pm 0.02$ | 1.39 | 0.99 |

| | *De Sitter* | | | |
|---|---|---|---|---|
| **Dataset** | **Embedding Dim** | **Distortion (avg. $\pm$ sdev.)** | **Max Distortion** | **Directionality** |
| Cornell | 2 | $-55.83 \pm 890.45$ | 3950.88 | 0.92 |
| | 4 | $-20.60 \pm 249.49$ | 403.46 | 0.94 |
| | 10 | $0.80 \pm 126.26$ | 1543.07 | 0.93 |
| Texas | 2 | $-0.29 \pm 84.42$ | 818.10 | 0.90 |
| | 4 | $42.03 \pm 795.51$ | 13939.25 | 0.90 |
| | 10 | $2.60 \pm 70.33$ | 1107.60 | 0.90 |
| Wisconsin | 2 | $2.06 \pm 63.46$ | 1291.31 | 0.89 |
| | 4 | $-0.78 \pm 27.91$ | 114.24 | 0.90 |
| | 10 | $0.04 \pm 215.94$ | 2862.19 | 0.89 |
| In silico | 2 | $-63.59 \pm 1866.69$ | 56626.97 | 0.92 |
| | 4 | $-468.81 \pm 33021.14$ | 65289.22 | 0.92 |
| | 10 | $-129.13 \pm 9623.30$ | 261531.59 | 0.93 |
| Escherichia coli | 2 | $34.65 \pm 2637.50$ | 119047.23 | 0.91 |
| | 4 | $-2.00 \pm 3294.81$ | 130509.59 | 0.91 |
| | 10 | $-8.26 \pm 94.57$ | 652.96 | 0.91 |
| Saccharomyces cerevisiae | 2 | $55.36 \pm 3960.09$ | 160278.39 | 0.90 |
| | 4 | $-28.60 \pm 1175.67$ | 63086.54 | 0.90 |
| | 10 | $-121.17 \pm 7550.16$ | 84724.25 | 0.91 |

### D.5 FURTHER CLARIFICATIONS ON EXPERIMENTS USING NEURAL SPACETIME EMBEDDINGS COMPARED TO FIXED-GEOMETRY BASELINES

In this section, we aim to provide additional intuition to the reader on how NSTs compare to the fixed-geometry embedding baselines and why we expect them to perform better.

For all experiments, we use a feature encoder that maps the graph node features to the relevant manifold. We assume that the output of the feature encoder is in Euclidean space in all cases. We then proceed according to the specific geometry. This approach aligns with previous works such as neural snowflakes (Borde & Kratsios, 2023) and other studies on product manifold embeddings (Borde et al., 2023b).

Intuitively, one can understand the problem as a two-step process, which in practice is learned jointly via end-to-end gradient descent. In the first step, the encoder learns to position the graph nodes in space—that is, it learns to map the original node features to coordinates in the manifold. In the second step, the distance between points on the manifold is evaluated, either based on a given metric when the geometry is fixed or by learning the geometry (the metric itself) in a data-driven fashion.

In the cases of Minkowski and De Sitter space, the geometry is not learned but given. More specifically, in Minkowski space, we use the feature outputs of the Euclidean feature encoder, apply a change in the metric tensor, and recognize that one coordinate has a different signature. In this particular case, we do not require an exponential map since the space is flat. For De Sitter space, the situation is similar: only the positioning of points in the manifold is optimized via gradient descent, while the geometry remains fixed. In terms of implementation and metric calculation, De Sitter space is a curved manifold, making computations more complicated than for Minkowski space. We use an embedding approach that avoids exponential maps: we map points into a De Sitter hyperboloid in a higher-dimensional Minkowski space and compute geodesic distances there. The geometric operations we utilize include normalizing spatial components, computing the De Sitter inner product, and handling both timelike and spacelike separations. Additionally, we would like to highlight that, although the authors in (Law & Lucas, 2023) also use Minkowski and De Sitter spaces, our baselines are more powerful. This is because we employ a neural network feature encoder to optimize the placement of node embeddings within the manifold

NSTs are intrinsically different in that their geometry is not fixed but rather random at initialization and parametrized by a neural network that always outputs a pseudo-quasi-metric by construction. The specific pseudo-quasi-metric it generates is learned during optimization based on the input graph. Hence, in this case, both the position of the points on the manifold and the geometry of the manifold itself are learned jointly via gradient descent. For NSTs, we do not need exponential maps either since the construction is based on warping a norm. During optimization, the inputs are the graph node features, and the targets are both the distances between nodes (induced by the graph geometry) and the causality structure (given by the edge directions). Since neural spacetimes are able to reshape the geometry of the embedding space as a function of the data, they are inherently more flexible and they are able to embed DAGs with less distortion (the best possible value for distortion is 1).

### D.6 DOWNSTREAM APPLICATION TESTING: NODE CLASSIFICAITION ON EMBEDDED HETEROPHILIC GRAPHS

Lastly, we evaluate the downstream performance of our encoding approach in transductive node classification for heterophilic graphs, specifically using the Cornell, Texas, and Wisconsin datasets. It is important to note that this is not a standard application of NSTs, as their original design aims to embed DAGs with minimal distortion. NSTs typically use the feature encoder network as an initial intermediate mapping to a latent space, which is then utilized by the learnable quasi-metric and learnable partial order to encode distance and directionality respectively. While the neural quasi-metric takes two node feature vectors as input and outputs a scalar distance value (making it unsuitable as direct input for a downstream node classification task), the neural partial order operates pointwise and returns a feature vector. Therefore, we adopt the following approach. First, we train the complete NST model to encode the original graph into the continuous geometry, simultaneously optimizing all three networks composing the NST, $\mathcal{E}$, $\mathcal{D}$, and $\mathcal{T}$. Once trained, we use the frozen feature encoder and neural partial order as pretrained featurizers to encode the original node features from the dataset at hand. These transformed features are then fed into a downstream network that is trained as a node-wise classifier. That is, the new node features for the graphs become:

$$z_v = x_v \parallel (\hat{x}_v)_{1:D} \parallel t_v, \quad \text{where} \quad \hat{x}_v = \mathcal{E}(x_v), \ t_v = \mathcal{T}(\hat{x}_v), \tag{43}$$

where we have augmented the original features with NST-based features optimized according to the graph topology. In the equation above $x_v$ corresponds to the original features for node $v$, $\mathcal{E}$ and $\mathcal{T}$ are the frozen feature encoder and partial order, which were originally optimized alongside $\mathcal{D}$. As in the main text, the subscript in $(\hat{x}_v)_{1:D}$ means that we utilize the first $D$ spatial dimensions from the feature vector, and we use $\parallel$ for concatenation. We use pre-trained checkpoints with $D = T = 10$.

We display the downstream results in Table 11 below. We experiment with different downstream classifiers, including MLPs, GCNs (Kipf & Welling, 2017), GATs (Veličković et al., 2018), CPGNN-MLPs (Zhu et al., 2021), and CPGNN-Cheby (Zhu et al., 2021; Defferrard et al., 2016). Note that CPGNN is not just a network but also a framework for training on heterophilic graphs. It employs an estimator network, such as an MLP or a Chebyshev polynomial-based model, for prediction. The CPGNN method incorporates estimator pretraining and utilizes a compatibility matrix, which is initialized using the training node labels and the training mask. This matrix is further refined using the Sinkhorn-Knopp algorithm for initialization. Furthermore, the compatibility matrix is regularized during training through an additional loss term on top of the cross-entropy loss. We reimplement all baselines and test them on the Cornell, Texas, and Wisconsin datasets using 10-fold splits, based on the masks provided in PyTorch Geometric. Following (Zhu et al., 2021) we use weight decay of $5 \times 10^{-4}$ for all networks and pretrain the estimators in CPGNN-MLP and CPGNN-Cheby for 100 steps. All our downstream classifiers use two layers with a ReLU activation function and are trained with a learning rate of 0.01 for 400 steps using AdamW. We experiment with varying hidden dimensions—10, 20, 30, and 64—the last of which is the hidden dimension used by the baselines in (Zhu et al., 2021). In our experiments, we study the effect of adding geometric node features based on the NST encoding to downstream classifiers. We find that the MLP classifier achieves the best downstream performance when augmented with NST features, particularly for the Texas and Wisconsin datasets. For Cornell, the best downstream performance is achieved by the MLP without NST features, which attains $71\%$ accuracy with a hidden dimension of 64. The counterpart augmented with NST features achieves $70\%$ accuracy, making the difference not particularly significant. Overall, in Cornell, we observe that adding NST features is especially beneficial for smaller classifiers with fewer hidden dimensions.

These experiments confirm that the NST has learned meaningful features that can be used by downstream classifiers. However, we invite future research in this direction, as this paper primarily focuses on embeddings, and this appendix is only an initial exploration of the applicability of NSTs to downstream tasks.

Table 11: Node classification results for heterophilic graphs, including the test accuracy mean and standard deviation. ✓ denotes that the input features to the downstream classifier consist of the original graph input features plus the NST features $z_v = x_v \parallel (\hat{x}_v)_{1:D} \parallel t_v$. In contrast, ✗ indicates that only $x_v$ is passed into the model.

| Dataset | Hidden Dim | NST Features | MLP | GCN | GAT | CPGNN-MLP | CPGNN-ChebNet |
|---|---|---|---|---|---|---|---|
| Cornell | 10 | ✓ | $0.55 \pm 0.09$ | $0.43 \pm 0.09$ | $0.48 \pm 0.08$ | $0.54 \pm 0.05$ | $0.51 \pm 0.05$ |
| | 10 | ✗ | $0.40 \pm 0.12$ | $0.43 \pm 0.06$ | $0.48 \pm 0.07$ | $0.47 \pm 0.10$ | $0.46 \pm 0.11$ |
| | 20 | ✓ | $0.61 \pm 0.09$ | $0.44 \pm 0.09$ | $0.49 \pm 0.08$ | $0.51 \pm 0.05$ | $0.50 \pm 0.10$ |
| | 20 | ✗ | $0.56 \pm 0.11$ | $0.43 \pm 0.06$ | $0.49 \pm 0.09$ | $0.43 \pm 0.11$ | $0.41 \pm 0.07$ |
| | 30 | ✓ | $0.70 \pm 0.06$ | $0.44 \pm 0.09$ | $0.47 \pm 0.09$ | $0.47 \pm 0.12$ | $0.52 \pm 0.07$ |
| | 30 | ✗ | $0.70 \pm 0.05$ | $0.43 \pm 0.07$ | $0.50 \pm 0.10$ | $0.42 \pm 0.11$ | $0.41 \pm 0.11$ |
| | 64 | ✓ | $0.70 \pm 0.07$ | $0.43 \pm 0.09$ | $0.50 \pm 0.06$ | $0.50 \pm 0.08$ | $0.52 \pm 0.07$ |
| | 64 | ✗ | $\mathbf{0.71 \pm 0.07}$ | $0.43 \pm 0.07$ | $0.50 \pm 0.08$ | $0.40 \pm 0.10$ | $0.40 \pm 0.11$ |
| Texas | 10 | ✓ | $0.58 \pm 0.14$ | $0.58 \pm 0.08$ | $0.56 \pm 0.09$ | $0.61 \pm 0.05$ | $0.62 \pm 0.08$ |
| | 10 | ✗ | $0.41 \pm 0.22$ | $0.49 \pm 0.08$ | $0.50 \pm 0.07$ | $0.63 \pm 0.10$ | $0.62 \pm 0.10$ |
| | 20 | ✓ | $0.64 \pm 0.07$ | $0.56 \pm 0.07$ | $0.54 \pm 0.08$ | $0.61 \pm 0.06$ | $0.63 \pm 0.08$ |
| | 20 | ✗ | $0.57 \pm 0.10$ | $0.49 \pm 0.07$ | $0.50 \pm 0.10$ | $0.61 \pm 0.10$ | $0.61 \pm 0.10$ |
| | 30 | ✓ | $0.72 \pm 0.08$ | $0.58 \pm 0.08$ | $0.52 \pm 0.09$ | $0.61 \pm 0.06$ | $0.64 \pm 0.06$ |
| | 30 | ✗ | $0.67 \pm 0.09$ | $0.49 \pm 0.07$ | $0.52 \pm 0.07$ | $0.61 \pm 0.10$ | $0.62 \pm 0.10$ |
| | 64 | ✓ | $\mathbf{0.74 \pm 0.05}$ | $0.55 \pm 0.07$ | $0.54 \pm 0.06$ | $0.61 \pm 0.06$ | $0.68 \pm 0.07$ |
| | 64 | ✗ | $0.68 \pm 0.05$ | $0.48 \pm 0.07$ | $0.47 \pm 0.07$ | $0.57 \pm 0.10$ | $0.55 \pm 0.11$ |
| Wisconsin | 10 | ✓ | $0.62 \pm 0.13$ | $0.49 \pm 0.05$ | $0.48 \pm 0.07$ | $0.48 \pm 0.05$ | $0.51 \pm 0.08$ |
| | 10 | ✗ | $0.54 \pm 0.16$ | $0.45 \pm 0.05$ | $0.46 \pm 0.05$ | $0.61 \pm 0.03$ | $0.58 \pm 0.05$ |
| | 20 | ✓ | $0.76 \pm 0.10$ | $0.50 \pm 0.06$ | $0.49 \pm 0.08$ | $0.50 \pm 0.10$ | $0.52 \pm 0.11$ |
| | 20 | ✗ | $0.70 \pm 0.09$ | $0.44 \pm 0.06$ | $0.48 \pm 0.05$ | $0.60 \pm 0.06$ | $0.58 \pm 0.05$ |
| | 30 | ✓ | $0.78 \pm 0.06$ | $0.49 \pm 0.06$ | $0.47 \pm 0.09$ | $0.50 \pm 0.08$ | $0.53 \pm 0.11$ |
| | 30 | ✗ | $0.72 \pm 0.05$ | $0.44 \pm 0.07$ | $0.49 \pm 0.08$ | $0.61 \pm 0.03$ | $0.57 \pm 0.04$ |
| | 64 | ✓ | $\mathbf{0.81 \pm 0.05}$ | $0.49 \pm 0.06$ | $0.51 \pm 0.10$ | $0.51 \pm 0.12$ | $0.53 \pm 0.13$ |
| | 64 | ✗ | $0.76 \pm 0.08$ | $0.43 \pm 0.07$ | $0.49 \pm 0.06$ | $0.57 \pm 0.04$ | $0.57 \pm 0.07$ |

