# OpenReview forum: "Neural Spacetimes for DAG Representation Learning"
_ICLR.cc/2025/Conference — ICLR 2025 Poster_

### Official Review · Reviewer_spGj · 2024-10-31

**Soundness:** 3
**Presentation:** 2
**Contribution:** 3
**Rating:** 6
**Confidence:** 2

**Summary:**

The authors introduce a novel architecture, Neural SpaceTimes (NSTs), which represents nodes in Directed Acyclic Graphs (DAGs) as events in a trainable spacetime manifold. This framework innovatively unifies spatial distances and causal directionality into a single representation space, designed to improve upon fixed spacetime geometries typically used in graph embedding. The authors implement this system through three neural networks: an embedding network, a neural quasi-metric, and a neural partial order network.

**Strengths:**

S1. NSTs' combination of quasi-metric spaces for spatial information and partial orders for causal information presents an elegant and flexible solution to representing DAGs.

S2. The authors provide a robust theoretical foundation.

S3. The experimental results on synthetic and real-world DAG datasets indicate the NST’s capability to achieve lower embedding distortions.

**Weaknesses:**

W1. Could authors provide more explanations about the experimental results? How is NST compared to closed-form spacetimes such as the Minkowski? Why the NST is better than the closed-from spacetimes? Besides, comparison with more baselines can be more convincing.

W2. It would be good to see a time complexity analysis for the proposed NST.

**Questions:**

Q1. In line 215 about the metric embedding, what is $\rho(x_1,x_2)$? Since $\rho$ is defined on $\mathcal{Y}$, how can $\rho$ compute on $\mathcal{X}$?

---

> ### Author Response · Authors · 2024-11-19
>
> # Weakness 1:
>
> We discuss experiments extensively in Appendix D and provide additional details about the computational implementation of Neural Spacetimes in Appendix C. However, we agree that additional discussion on the computational implementation of learnable geometries (such as Neural Spacetimes) versus fixed geometries (such as Minkowski and de Sitter space) would be helpful for readers (we have added appendix D.5).
>
> For all experiments, we use a feature encoder that maps the graph node features to the relevant manifold. We assume that the output of the feature encoder is in Euclidean space in all cases. We then proceed according to the specific geometry. This approach aligns with previous works such as neural snowflakes and other studies on product manifold embeddings cited in the main paper.
> Intuitively, one can understand the problem as a two-step process, which in practice is learned jointly via end-to-end gradient descent. In the first step, the encoder learns to position the graph nodes in space—that is, it learns to map the original node features to coordinates in the manifold. In the second step, the distance between points on the manifold is evaluated, either based on a given metric when the geometry is fixed or by learning the geometry (the metric itself) in a data-driven fashion.
> In the cases of Minkowski and de Sitter space, the geometry is not learned but given. More specifically, in Minkowski space, we use the feature outputs of the Euclidean feature encoder, apply a change in the metric tensor, and recognize that one coordinate has a different signature. In this particular case, we do not require an exponential map since the space is flat. For de Sitter space, the situation is similar: only the positioning of points in the manifold is optimized via gradient descent, while the geometry remains fixed. In terms of implementation and metric calculation, de Sitter space is a curved manifold, making computations more complicated than for Minkowski space. We use an embedding approach that avoids exponential maps: we map points into a de Sitter hyperboloid in a higher-dimensional Minkowski space and compute geodesic distances there. The geometric operations we utilize include normalizing spatial components, computing the de Sitter inner product, and handling both timelike and spacelike separations.
> Neural spacetimes are intrinsically different in that their geometry is not fixed but rather random at initialization and parametrized by a neural network that always outputs a pseudo-quasi-metric by construction. However, the specific pseudo-quasi-metric it generates is learned during optimization based on the input graph. Hence, in this case, both the position of the points on the manifold and the geometry of the manifold itself are learned jointly via gradient descent. For Neural Spacetimes, we don't need exponential maps either since the construction is based on warping a norm.
> During optimization, the inputs are the graph node features, and the targets are both the distances between nodes (induced by the graph geometry) and the causality structure (given by the edge directions). Since Neural Spacetimes are able to reshape the geometry of the embedding space as a function of the data, they are inherently more flexible and they are able to embed DAGs with less distortion (the best possible value for distortion is 1).
>
> # Weakness 2:
>
> Please refer to Appendix C.2 for algorithmic descriptions of the components of the NST model. The feature encoder is an MLP, and both the neural partial order and the neural quasi-metric are based on linear projections and pointwise non-linearities. Consequently, the computational complexity is comparable to that of MLPs. We do not use any computationally expensive operations, such as attention.
>
> For an MLP with $L$ layers, the total time complexity sum across layers (total number of layers L) is $O\left(n \sum_{i=1}^{L} d_{\text{in}}^{(i)} \times d_{\text{out}}^{(i)}\right)$ where $d_{\text{in}}^{(i)}$ and $d_{\text{out}}^{(i)}$ are the input and output dimensions of the linear layer $i$, and $n$ is the batch size. The same would apply in our case.
>
> # Question 1:
>
> Thank you, this was a typo and also pointed out by Reviewer Crpy. The correct formula (which we have used in its correct form throughout the manuscript of course) should read:
> $c\,\rho(f(x_1),f(x_2)) \le d(x_1,x_2) \le D \,c\rho(f(x_1),f(x_2)).$
> We have made the correction; thank you for noticing this typo.

---

### Official Review · Reviewer_icVJ · 2024-11-01

**Soundness:** 3
**Presentation:** 3
**Contribution:** 3
**Rating:** 6
**Confidence:** 3

**Summary:**

This paper proposes a framework of quasimetric embeddings using spacetime as the embedding manifold.

**Strengths:**

1. The method seems well-motivated to me. In particular, the decomposition makes sense as a space for DAG-like data, and the paper proposes a nice way of doing so with neural networks.
2. The empirical results seem promising, as the method seems to be able near perfectly embed many common DAG graph datasets.

**Weaknesses:**

While the paper seems to be mostly complete to me, several concerning aspects about the experiments did not sit right with me.

1. The empirical results have some concerning aspects about them. In particular, the Minkowski embedding is able to largely get to near the same final distortion on both the real world graph datasets. Since the proposed method uses neural networks as well as node features (at least for the real-world data) while the previous methods do not (as is my understanding from skimming the Law and Lucas paper), this suggests that the difference provided by the proposed method is not very large with respect to the required additional effort or that the test tasks are too simple.
2. While the distortion is near perfect, the downstream effects of using these embeddings is never studied. In particular, some GNN-type experiments would be greatly appreciated.

**Questions:**

N/A

---

> ### Author Response · Authors · 2024-11-19
>
> Dear reviewer,
>
> Thank you for your comments.
>
> # Weakness 1:
>
> We should perhaps make this more explicit in the text: all our baselines are more powerful than those presented in Law and Lucas, where no neural networks are used. We have added a more detail explanation in Appendix D.5.
>
> In our experiments, a feature encoder is employed across all cases, including those where the geometry of the space is fixed. Specifically, for the Minkowski and De Sitter metrics, we use a neural network to map the original node features to the geometry, even when the metric itself is not learned. This approach ensures a fair comparison between using a fixed, closed-form metric and learning the geometry directly with NSTs.
>
> It is important to note that the best possible distortion is 1, which corresponds exactly to an isometric embedding.  Indeed, if the distortion=1, then Definition 2 (ii) becomes an equality.  We understand that this may not be clear to researchers not used to this kind of evaluation measure with metric embedding problems, and we will add a remark in our paper.
> We disagree that Minkowski embeddings achieve similar levels of final distortion in real-world datasets. For instance, they perform particularly poorly in In silico, E. coli, and S. cerevisiae (see Table 2), these are real-world gene regulatory networks. Additionally, in line with neural snowflakes, we would like to emphasize that NSTs are especially effective at preserving low distortion in lower dimensions: when the embedding dimension is reduced to 2, the performance gap between NSTs and the fixed metrics becomes notably more pronounced.
>
> # Weakness 2:
>
> We have included downstream experiments in Appendix D.6 using NST features to augment the input features to downstream classifiers such as MLPs, GCNs, GATs, CPGNN-MLPs, and CPGNN-Chebys for node classification. We observe that NST features help downstream accuracy. Please refer to the appendix for further details. We also discuss this in the response to reviewer vRyg.

---

### Official Review · Reviewer_uYUp · 2024-11-02

**Soundness:** 4
**Presentation:** 3
**Contribution:** 3
**Rating:** 8
**Confidence:** 3

**Summary:**

The paper proposes Neural SpaceTimes (NSTs), a framework for embedding directed acyclic graphs (DGAs) in continuous geometries which can encode both edge weights and directionality. Unlike previous methods that use fixed spacetime manifolds, NSTs learn the geometry itself through three neural networks -- an embedding network for mapping nodes to coordinates, a neural quasi-metric network for encoding distances in space dimensions, and a neural partial order network for encoding causality in time dimensions. The authors provide theoretical guarantees showing that NSTs can universally embed any k-point DAG with low distortion while preserving its causal structure. Experiments on both synthetic and real-world directed graphs demonstrate that NSTs achieve lower embedding distortions compared to baseline methods using fixed geometries like Minkowski or de Sitter spaces.

**Strengths:**

* The paper is well-organized with clear problem statement , goal (Section 1), and the proposed solution (Section 3).
* The theoretical framework in Section 3 decouples spatial and temporal components while maintaining their interaction through the embedding network.
* The theoretical development in Section 3.1 establishes embedding guarantees through Theorem 1, showing NSTs can universally embed k-point DAGs.
* The experimental validation in Section 4 demonstrates clear improvements over baselines, particularly for low-dimensional embeddings where NSTs maintain average distortion near 1.0 while other methods show much higher distortion.
* The practical applicability is well explained through the analysis of real-world networks in Table 2.

**Weaknesses:**

* In Section 3.1, while the paper proves existence of neural sapcetime embeddings with low distortion, it provides limited practical guidence on selecting the number of temporal dimension. This could be important for complex DAGs where the minimal sufficient dimension is not immediately clear.
* The optimization approach in Section 3.2 focuses on preserving local structure during training. While the authors provide theoretical guarantees about global structure preservation, the empirical validation concentrates mainly on local metrics.

**Questions:**

1. How does the framework scale to very large graphs? The current larges experiment uses ~200 k nodes, but many real-world networks could be larger.
2. The local vs. global geometry optimization tradeoff warrants deeper analysis -- could hierarchical or multi-scale approaches help bridge this gap while maintaining computational tractability?
3. While the paper shows NST can encode both local distances and causality, how well does it preserve higher-order DAG properties like path lengths between distant nodes or branching structures? This could be particularly relevant for applications like computational workflow graphs or neural architecture search where these properties are important.
4. For DAGs with hierarchical structure (like computational graphs or dependency trees), how does the learned geometry reflect this hierarchy? Additional analysis of how NSTs capture hierarchical relationships would strengthen the practical applications.

---

> ### Author Response · Authors · 2024-11-19
>
> Dear reviewer,
>
> Thank you for the suggestions. See the responses below.
>
> # Weakness 1:
>
> To be able to fully capture the causality structure we must have the number of time dimensions be equal to the number of antichains. Please see Appendix C.5 for additional suggestions on this matter.
>
> # Weakness 2:
>
> The NST algorithm is not limited to optimizing for local geometry alone. However, from a computational standpoint, the problem appears intractable. We discuss the global optimization setup in more detail in Appendix C.4. The main challenge lies in generating data to train the algorithm—specifically, generating ground-truth global distances and inferring all global causal relationships within a graph.
>
> Nevertheless, in Appendix D.1, we conduct global embedding experiments on unweighted, undirected trees where all local distances are set to 1. In this particular setup, the problem is simplified, allowing us to focus on evaluating the global spatial embedding capabilities of NST using only the learnable quasi-metric and the feature encoder.
>
> # Question 1:
>
> Following the suggestion of reviewer vRyg we have run experiments for ogbn-arxiv, see Appendix D.3 'Ogbn-arxiv Large Citation Graph Embedding'. We observe that in this case NSTs are also superior to the baseline geometries.
>
> # Question 2:
>
> This is an interesting question, but we do not have a straightforward answer at the moment, this would be an interesting extension of our work.
>
> # Question 3:
>
> Please refer to Section 3.2: ‘Local geometry optimization and global geometry implications’. The model is limited by optimizing the local geometry instead of global geodesic distances over the DAG. Although preserving global distances is not guaranteed, the local transitivity of causal connectivity will implicitly hold globally, even when performing local optimization only.
>
> # Question 4:
>
> Similarly to Question 3, the hierarchical causal structure will be preserved globally thanks to the transitivity property even when optimizing locally. The main concern is guaranteeing lack of causality between anti-chains: this will naturally happen with high probability as the number of time dimensions are increased.

---

> > ### Comment · Reviewer_uYUp · 2024-11-29
> >
> > Dear Authors,
> >
> > Thank you for answering to my questions. It addresses most of my concerns.

---

### Official Review · Reviewer_Crpy · 2024-11-02

**Soundness:** 4
**Presentation:** 3
**Contribution:** 4
**Rating:** 8
**Confidence:** 4

**Summary:**

The paper sets forth a new neural network architecture capable of learning the embedding of a DAG into a pseudo-Riemannian manifold, which encodes both the spatial and temporal distance between nodes of the DAG in a continuous way. Some conditions (along with proof) are given which guarantee that the embedding is faithful.

The network itself is composed of three separate networks, namely, an initial latent space embedding $\mathcal{E}: \mathbb{R}^N \rightarrow \mathbb{R}^{D+T}$, a modified neural snowflake $\mathcal{D}: \mathbb{R}^{D+T} \times \mathbb{R}^{D+T} \rightarrow [0, \infty)$ which is the quasi-metric on the manifold, and a partial order $\mathcal{T}: \mathbb{R}^{D+T} \rightarrow \mathbb{R}^T$ which encodes the causal relationships in the DAG. These latter two networks are trained parallel to each other and are optimized with respect to an aggregated loss function $\mathcal{L} = \mathcal{L}^{\mathcal{D}} + \mathcal{L}^{\mathcal{T}}$.

Empirical results are given over several datasets, some synthetic with a complex pseudo-metric and others real-world webpage hyperlink DAGs. The results are favorable of the new architecture compared to those of Minkowski and De Sitter space embeddings, showing that the new architecture shows little distortion in the learned quasi-metric and almost always preserves the causal relationships among nodes in the DAGs.

**Strengths:**

The paper offers originality within its domain by constructing a novel architecture capable of embedding a DAG into a pseudo-Riemannian manifold which has several spatial and temporal components. In particular, adding the capability of embedding a given DAG into more than one temporal component is a novel contribution and allows the network to represent and preserve more complex causal structure from the DAG. Furthermore, the authors modify the neural snowflake model for use as a learnable quasi-metric.

The quality of the paper is also very good given that the authors cited several relevant related works and gave ample justification in the form of mathematical proof for their many propositions and theorems, thus adding to the already existing clarity of the paper.

I believe this paper makes a significant contribution to the field by adding the ability to represent discrete structures in a continuous fashion with very little approximation error, thus presenting a new and powerful tool to various application domains.

**Weaknesses:**

The paper would benefit from further experimentation, e.g., showcasing the results of the network to various distributions of DAGs and being explicit about any potential weaknesses. It would be nice to know where this architecture breaks down or how. robust it is to perturbations in the assumptions.

The language in lines 420-422 becomes imprecise and can lead to confusion. I would suggest sticking to a single convention here when using variables in a description. Similarly when the switch is made from using $T$ to using $W$, it's not clear why this is done and can lead to confusion.

Also, there seems to be a type on line 215 that would make a significant difference. The elements $x_1, x_2 \in \mathcal{X}$ cannot be compared using the metric from $\mathcal{Y}$. I assume you meant to say $\rho(f(x_1), f(x_2))$ not $\rho(x_1, x_2)$.

**Questions:**

1) In the latent space embedding, is there any guarantee that $\mathcal{E}$ maps feature vectors in such a way that the resulting embedding can be partitioned into spatial and temporal components? E.g., you embed features from $\mathbb{R}^N$ into $\mathbb{R}^{D+T}$ and later separate the embedding by slicing the results ($1:D$ and $D+1:D+T$) for use in the recursions of $\mathcal{D}$ and $\mathcal{T}$. Therefore, my question is, how do you guarantee that spatial information ends up only in the first $D$ dimensions, and temporal in the last $T$ dimensions?

---

> ### Author Response · Authors · 2024-11-19
>
> Dear reviewer,
>
> Thank you for your comments, see responses below.
>
> # Weakness 1: Further experiments
>
> We have included the experiments addressing your concern regarding different distributions of graphs, see Appendix D.2.1 in which we vary the graph connectivity. We find that given the same number of nodes, a less connected graph is easier to embed in space and harder to embed in time according to our experiments.
>
> # Weakness 2.1: Language in lines 420-422.
> Thank you for pointing out the inconsistency in notation. We have fixed it for the final version of the manuscript. We have rewritten everything in terms of $\hat{x}_u,\hat{x}_v,\hat{x}_w$.
>
> # Weakness 2.2: T and W variables
> Thank you for pointing out a possible confusion for readers.
> We chose to use W in that theorem statement, as we are emphasizing that for a graph with width W one may take T=W; i.e.\ W is given before T is determined to be T:=W.  Nevertheless, if the paper would be clearer by adding some details on that then we would be more than happy to.
>
> # Weakness 2.3: Typo in line 215
> Indeed, you are right; thanks for noticing that typo.  We have corrected it.
>
> # Question: Space and Time dimensions in the latent space embedding.
>
> Both the features passed to $\mathcal{D}$ and $\mathcal{T}$ are functions of the original node features. The original node features lie in $\mathbb{R}^N$ and are mapped to $\mathbb{R}^D$ and $\mathbb{R}^T$. Manually partitioning these features in $\mathbb{R}^N$ into spatial and temporal components is impractical, as the original graph lacks a defined notion of spatial and temporal features. Thus, we use a neural network encoder to map the features, learning an optimal transformation of the data via gradient descent in an end-to-end manner. Alternatively, the feature encoder network can be seen as creating two parallel projections of the original input features: one for space and one for time. In summary, the NST model learns the optimal projection of the original input features for space and time by propagating gradients through $\mathcal{D}$, $\mathcal{T}$, and $\mathcal{E}$ simultaneously. The choices of $D$ and $T$ are left as hyperparameters, which may be determined experimentally. Alternatively, domain knowledge may guide these choices; for example, if the number of antichains in the underlying DAG is known a priori, this information can help set the time dimension accordingly. In Appendix C.5, we also discuss potential classical algorithms from the literature to assist in tuning the $D$ and $T$ hyperparameters.

---

> > ### Comment · Reviewer_Crpy · 2024-11-26
> >
> > Authors,
> >
> > Thank you for answering my questions and correcting the weaknesses that I mentioned.
> >
> > Your answer to my question about space and time dimensions, in the latent space of the encoder embedding, still leaves me somewhat skeptical though. I'll try and clarify my original question: The node features in $\mathbb{R}^N$ are mapped into the encoder's latent space via a MLP. This MLP necessarily combines node features and produces a combination of them in the latent space, which is why I wonder how they can later be separated into distinct spaces. Even after training, unless corresponding weights are exactly $0$ in the MLP, they have overlap. Therefore, how do you account for this and do you think this has something to do with the approximation errors you see later on?

---

> > > ### Author Response · Authors · 2024-11-28
> > >
> > > Dear Reviewer,
> > >
> > > We would like to clarify our response further.
> > >
> > > **The neural spacetime can be understood as a single manifold.**
> > > We believe your concern relates to feature mixing. The mixing of information in the latent representations (which we use as an intermediate embedding without direct physical meaning) is both expected and acceptable.
> > >
> > > Our goal is to find an effective mapping from the nodes of the original discrete graph structure to points on the product manifold—the neural spacetime, which is the Cartesian product of a spatial and a temporal manifold. However, this product manifold can also be viewed as a single unified manifold composed of two simpler substructures. Similarly, $\mathbb{E}^2$ (2D Euclidean space) can be regarded as the Cartesian product of $\mathbb{E}^1$ and $\mathbb{E}^1$: in such a case if we wanted to embed a graph it would be intuitive to make all coordinates a function of all the input features, the same applies in our case. Viewing the neural spacetime (NST) as a manifold in its own right makes it more intuitive to understand why using all input features to map from the graph to the manifold is acceptable. One can think of different coordinates on the neural spacetime space as being represented as coordinates in other substructures, but ultimate all coordinates are needed to represent the point as a point on the manifold.
> > >
> > > The internal partitioning within the neural spacetime architecture serves as an inductive bias or a form of multi-task learning. Ultimately, the model's goal is to optimize the mapping from one space to another. Consequently, all dimensions of the node features can contribute to both the spatial and temporal coordinates of the embedding.
> > >
> > > **Node features do not necessarily have a physical meaning that classifies them as temporal or spatial.**
> > > The input node features on the graph are not inherently classified as temporal or spatial. Additionally, our "temporal coordinates" are used to model causality (i.e., edge directionality) in a static graph. This directionality may depend on features that are not strictly "temporal" but could instead reflect static attributes.
> > >
> > > We do agree that rather than using a single MLP encoder for both space and time we could use two MLPs, in such a case we would still use all features in the input to learn the space and time representations but we would decouple the network weigths further upstream in the representation.
> > >
> > > In the current implementation of the MLP encoder, only the weights in the final linear projection are "decoupled." However, note that the neural quasi-metric and the neural partial order are decoupled projections of the features. Deciding whether to decouple weights earlier or later in the representation process could be considered part of an architectural search. Ultimately, though, all representations remain dependent on all feature dimensions of the input. The error could possibly be further optimized by fiddling with the architecture design, but the core idea would remain the same.

---

> > > > ### Comment · Reviewer_Crpy · 2024-12-02
> > > >
> > > > Thank you for your thorough response. You've satisfactorily answered my questions and given further justification for my current score.

---

### Official Review · Reviewer_vRyg · 2024-11-04

**Soundness:** 2
**Presentation:** 3
**Contribution:** 2
**Rating:** 6
**Confidence:** 4

**Summary:**

This paper presents Neural SpaceTimes (NSTs), a class of trainable geometries that can encode nodes of weighted Directed Acyclic Graphs (DAGs) into a spacetime manifold. NSTs are designed to capture both spatial and causal (temporal) information by using a product manifold combining spatial quasi-metrics with temporal partial orders to represent graph weights and directionality. Besides, theoretical analysis of the universal embedding theorem shows that NSTs can learn low-distortion embeddings for any DAG. Some experiments are also conducted to validate the effectiveness of NSTs.

**Strengths:**

1. The problem tackled in this paper, i.e., representation learning in DAGs, is challenging and significant in the graph machine learning community.
2. Theoretical guarantees of the proposed approach are provided in the manuscript.

**Weaknesses:**

Despite the merits of the paper, I have the following concerns.
1. Most results presented in this paper pertain to distortion and directionality, which demonstrate that NSTs perform robustly when evaluated by these two measures. However, there are no concrete results showing the performance of NSTs in real-world learning tasks, e.g., (semi-supervised/unsupervised) node classification, link prediction, and graph classification in synthetic DAGs, real-world DAGs (e.g., web page hyperlink graphs and citation graphs), and spatiotemporal graphs (e.g., encrypted transaction graphs). Authors are suggested to conduct some previously mentioned learning tasks and investigate how NSTs perform.
2. Some test datasets used in the experiments are always considered by those graph neural networks tackling graph heterophily (e.g., CPGNN). Compared with these graph neural networks tackling graph heterophily, how do NSTs perform in real-world tasks?
3. Most test datasets considered in this paper are also used to test conventional graph neural networks, e.g., GCN, GAT, GraphSAGE, and Graph Transformer. The authors should compare NSTs with conventional graph neural networks in real-world learning tasks (as mentioned in point 1).
4. The test datasets used in this paper are relatively small. How do NSTs perform in larger datasets? The authors can consider using datasets from OGBN, and other large datasets.

**Questions:**

1. How do NSTs perform in real-world learning tasks evaluated by well-established metrics (e.g., accuracy, NMI, and F1, ROC-AUC)?
2. How do NSTs perform when compared with conventional GNNs in real-world learning tasks?
3. Do NSTs still perform robustly in massive graphs?
4. How do NSTs perform when compared with GNNs tackling heterophily?

**Details Of Ethics Concerns:**

This reviewer has no critical ethical concerns.

---

> ### Author Response · Authors · 2024-11-19
> **Downstream tasks and clarifications**
>
> Dear reviewer,
>
> Thank you for your comments. We have updated the manuscript to address your concerns. Note that changes are highlighted in red.
>
> # Downstream experiments
>
> We understand the concern regarding downstream tasks. First, we would like to emphasize that the primary objective of this paper is focused on embeddings—specifically, the ability to faithfully represent discrete graph structures within a manifold representation. This focus forms the foundation of both our theoretical guarantees and computational framework. Accordingly, the main experiments in the text concentrate on this objective, encompassing both synthetic DAG embedding experiments and real-world applications such as hyperlink and gene regulatory networks.
>
> Nevertheless, we agree that exploring the downstream performance of features derived from these embeddings is an interesting direction for future research. In response to comments raised in **Weakness Points 1, 2, and 3**, we have included Appendix D.6 at the end of the paper. This appendix proposes a method for leveraging NST-based features in downstream tasks and evaluates them using MLPs, GCNs, GATs, CPGNN-MLPs, and CPGNN-Chebys on the Cornell, Texas, and Wisconsin datasets. We hope this addition addresses:
> - Node classification experiments (**Point 1**),
> - Benchmarking with CPGNN and heterophilic graphs (**Point 2**, as Cornell, Texas, and Wisconsin are heterophilic), and
> - Benchmarking with GCN and GAT on real-world learning tasks (**Point 3**).
> We have reproduced all the baselines, including the full CPGNN framework (pretraining, initialization, and regularization).
>
> Hyperparameters and results are extensively discussed in the appendix. We also reiterate that this was not the original focus of the paper and invite the community to further explore downstream applications of NSTs.
>
> Finally, we would like to clarify that our framework does not address spatiotemporal graphs. We only consider static graphs with directed edges, not dynamic graphs that evolve over time. In our case, the "time" manifold captures causality (i.e., directionality within the static graph). Spatiotemporal graph learning and DAG learning are distinct problems, and our work focuses on the latter.
>
> # Large Graph Embeddings
>
> We also address **point 4 in the weaknesses** section by including an additional subsection called 'Ogbn-arxiv Large Citation Graph Embedding' inside Appendix D.3. In which we embed the arxiv dataset from OGBN as suggested by the reviewer, which is two orders of magnitude bigger than the other graph datasets we considered before. We consistently observe the superiority of NSTs as compared to fixed geometries.
>
> # Questions
>
> - **Question 1**: The reviewer asks how do NSTs perform in real-world tasks using well-established metrics such as accuracy. Please refer to the newly included Appendix D.6. We can see that including NST based features boost the performance in terms of accuracy for most model configurations and ablations.  We would also like to highlight that our distortion metric used to evaluate embedding is a standard 'well-established metric' in the graph embedding literature. Please refer for reference such as 'Cuts, Trees and `1-Embeddings of Graphs' and also ' Embedding Tree Metrics into Low-Dimensional Euclidean Spaces' by Gupta. For the 'directionality' embedding we do use accuracy to quantify the proportion of edges the NST represents correctly.
>
> - **Question 2**: We have addressed with in Appendix D.6 too. Rather than how NSTs compare to GNNs, we use NST features to enhance downstream classifiers, which may include GNNs. Interestingly, we actually find that MLPs enhanced with NSTs work best at least for Texas, Cornell, and Wisconsin.
>
> - **Question 3**: See Appendix D.3. We still show superiority of NSTs over Minkowski and De Sitter space. As expected, increasing the graph makes the task harder for all models.
>
> - **Question 4**: In appendix D.6 we focus on heterophilic graphs and use the baselines suggested by the reviewer. NST features do help in general for this task according to our initial experimentation.

---

> > ### Comment · Reviewer_vRyg · 2024-11-22
> >
> > Dear Authors, thanks very much for the detailed responses to my comments, which address most of my concerns. I will raise the score of this paper to 6 to show my support to this paper. Good luck!
> >
> > In addition, I have some further suggestions for the paper. Although the current paper mainly focuses on the theoretical perspectives of DAG representation learning, it would be great if the authors could provide more discussions (mainly in the appendix, I guess) that can connect the method proposed in this paper to some concrete applications or learning tasks. For example, what data/learning tasks is the proposed method good (or not good) at dealing with? How can we better adapt the proposed method to diverse real-world applications? Are there generic rules one can follow to implement the proposed method to solve real-world learning tasks?

---

> > > ### Author Response · Authors · 2024-11-28
> > >
> > > Dear Reviewer, thank you for raising our score. We will expand the appendix with your suggestions in the camera-ready version.

---

### Author Response · Authors · 2024-11-19
**Paper updates during revision period**

We would like to thank the reviewers for their comments.

All changes to the manuscript are highlighted in red. Most of the changes are in the appendix, where we have added additional sections and experiments. Please scroll down to the end to review them.

# Typos

- As pointed out by the reviewers, there was a typo in Section 2.3, Definition 2(ii), which we have now corrected. We would like to emphasize that this typo was present only in the definition within the main text and does not affect any of our derivations.

- Likewise, we have made the notation in Section 3.2 more consistent, see the paragraph on 'Local geometry optimization and global geometry implications'.

# Additional Experiments

- We have included additional experiments in Appendix D.2.1. in which we vary the graph topology of the DAGs.
- We have included additional real-world large graph embedding experiments in Appendix D.3, with the graph being 2 orders of magnitude larger than in the original experiments.
- We have included downstream node classification experiments using NST based features in Appendix D.6 in which we benchmark MLPs, GCNs, GATs, CPGNN-MLPs, and CPGNN-Cheby models.

# Additional Clarifications on Framework and Comparison between NSTs and Minkowski and De Sitter experiments

- We have included Appendix D.5 clarifying the experimental difference between NSTs and baseline geometries since this was asked by the reviewers.

---

### Meta-Review · Area_Chair_Cqq7 · 2024-12-21

**Metareview:**

This paper introduces Neural SpaceTimes (NSTs), a novel framework for embedding directed acyclic graphs (DAGs) into continuous geometries that capture both edge weights and directionality. Unlike prior methods relying on fixed spacetime manifolds, NSTs dynamically learn the geometry through three neural networks: an embedding network that maps nodes to coordinates, a neural quasi-metric network for encoding spatial distances, and a neural partial order network for capturing causality in temporal dimensions.

The authors provide theoretical guarantees, demonstrating that NSTs can universally embed any \(k\)-point DAG with low distortion while preserving its causal structure. In Section 3.1, Theorem 1 rigorously establishes these embedding guarantees, showcasing the robustness of the framework.

This work represents a significant advancement by enabling the representation of discrete structures in a continuous space with minimal approximation error, offering a versatile and powerful tool for a wide range of application domains.

**Additional Comments On Reviewer Discussion:**

The reviewers have raised some questions with more results on real-world applications. The authors have addressed those concerns during rebuttal. The reviewers satisfied with the updates.

---

### Decision · Program_Chairs · 2025-01-22

Accept (Poster)